# A FtsZ *cis* disassembly element acts in Z-ring assembly during bacterial cell division

Huijia Yin[1,2,4], Yang Liu[1,4], Ying Zhao[1,4], Pengyue Chen[1] & Zengyi Chang [1,3] ✉

Bacterial cell division hinges on the Z-ring, an architecture built from the dynamical assembly and disassembly of FtsZ proteins. This delicate balance ensures not only apparent stability, but also continuous remodeling, both of which are required for Z-ring functioning. However, the molecular nature of such subcellular structures remains elusive. Here, by identifying all amino acid residues participating in FtsZ self-assembly in *Escherichia coli*, we show that the extreme N-terminal intrinsically disordered region (N-IDR) of FtsZ acts as a *cis* disassembly element that contacts and disrupts the longitudinal interface, tipping the balance more toward polymer disassembly. This previously unappreciated structural characteristic is indispensable for promoting Z-ring architecture condensation at midcell (rather than elsewhere) upon modulation by certain *trans*-acting factors (such as the *E. coli* MinC protein).

Cell division is a key process in both eukaryotes and prokaryotes. Prokaryotic cells divide via a binary fission process involving DNA replication and cytokinesis[1,2]. For bacterial cells, the cytoskeletal protein FtsZ assembles into a ring-like structure, called the Z-ring, at midcell, which in turn recruits other proteins to form a divisome that causes constriction and scission of the cell envelope, ultimately allowing a mother cell to divide into two daughter cells[3–5]. As the divisome scaffold, the Z-ring must be assembled at a precise time and place.

Super-resolution fluorescence microscopy studies have shown that Z-ring architecture exhibits high heterogeneity in cells[4,6–8]. The FtsZ protein, a homolog of eukaryotic tubulin, longitudinally assembles into protofilaments[9–11], which are dynamic entities and key components of Z-ring assembly[12]. To shift the FtsZ polymer concentration toward midcell and away from the poles, extrinsic positioning *trans* factors are used by many bacteria. For example, the *E. coli* MinC protein, which exhibits pole-to-pole oscillations in cells, is believed to disassemble FtsZ protofilaments beyond the midcell position[13]. Furthermore, FtsZ subunits are reported to display a treadmill behavior, which may facilitate FtsZ protofilaments to condense into a Z-ring architecture[14–16].

In vitro studies have suggested that FtsZ assembly occurs upon guanosine triphosphate (GTP) binding, and disassembly is facilitated by GTP hydrolysis[11,17,18]. However, the importance of GTPase activity for FtsZ functioning remains unclear. For example, the FtsZ84 variant (FtsZ^{G105S}), which has ~10% of the GTPase activity of wild-type FtsZ and significant defects in FtsZ treadmilling, and also slower recovery in fluorescence-recovery-after-photobleaching (FRAP), was shown to support cell division[19]. The molecular mechanisms whereby a functional Z-ring architecture is formed in cells via FtsZ dynamic assembly and disassembly remain elusive.

To decipher this, a critical aspect is to characterize all FtsZ self-assembly interfaces in cells, particularly those unobservable in structures resolved under in vitro conditions.

## Results

### The FtsZ N-terminal intrinsically disordered region (N-IDR) participates in longitudinal assembly

In searching for elusive FtsZ self-assembly interfaces involved in assembling the dynamic Z-ring in bacterial cells, we performed in vivo protein photocrosslinking for each of the 383 FtsZ residues, except the first Met residue, as mediated by the unnatural amino acid Bpa (*p*-benzoyl-L-phenylalanine) (Fig. 1a, Step 1)[20–22]. To this end, we first analyzed the crosslinked products of all 382 Bpa variants by resolving UV-exposed *Escherichia coli* cell samples on a single high-throughput

[1]State Key Laboratory of Gene Function and Modulation Research, School of Life Sciences, Peking University, Beijing 100871, P.R. China. [2]Institute of Geriatrics, National Clinical Research Center for Geriatric Diseases, Second Medical Center of Chinese PLA General Hospital, Beijing 100853, P.R. China. [3]Center for Protein Science, Peking University, Beijing 100871, P.R. China. [4]These authors contributed equally: Huijia Yin, Yang Liu, Ying Zhao. ✉e-mail: changzy@pku.edu.cn

SDS-PAGE gel, a methodology previously reported by us[22], and then probed the gel with anti-FtsZ antibodies.

Remarkably, residue positions showing crosslinking products were located across the whole polypeptide chain, not only in the previously defined N-domain (residues 11–176), C-domain (residues 201–316)[23,24] and the C-terminal sequence (residues 317–383), but also in the extreme N-terminus (residues 1–10), which was previously under-characterized (Fig. 1b). More importantly, many residues were never previously reported to mediate protein-protein interactions (Supplementary Fig. 1a). We then retrieved the *E. coli* FtsZ structure from the AlphaFold Protein Structure Database (AF-P0A9A6-F1-v4, https://alphafold.ebi.ac.uk/entry/P0A9A6) and also used Protein Dis-Order prediction System (PrDOS)[25] software to predict its structure. Interestingly, AlphaFold failed to predict a structure for the ten residues from the extreme N-terminus, while PrDOS, with high confidence, predicted eight residues from this terminus as an intrinsically disordered region. Accordingly, we defined these 1–10 residues as an N-intrinsically disordered region (N-IDR). Further sequence alignment analysis of FtsZ proteins from different bacteria showed that they all possessed N-IDR regions, with various lengths and low amino acid sequence similarities[26].

We then assessed whether any of these residues were involved in FtsZ self-assembly interactions. For this, we verified photocrosslinking results involving the N-IDR and N-domain residues in the *ftsZ-Avi* strain (the FtsZ-Avi protein is expressed from genomic DNA)[21], and analyzed crosslinked products using regular (rather than high-throughput) sodium dodecyl sulfate-polyacrylamide gel electrophoresis (SDS-PAGE). It followed that the self-assembled dimer could be visualized as doublet bands (FtsZ[Bpa] +FtsZ-Avi and FtsZ[Bpa] +FtsZ[Bpa]) upon detecting FtsZ but as a single band (FtsZ[Bpa]+FtsZ-Avi) upon detecting the Avi tag (Fig. 1a, Step 2). These results (Fig. 1c and Supplementary Fig. 1b) showed that almost all these verified residues were involved in FtsZ self-interactions. Strikingly, many of these self-assembly residues located in the N-IDR and N-domain (colored bold red) were revealed for the first time. These revelations compelled us to further explore the molecular mechanisms and biological significance of these self-assembly interactions in Z-ring formation.

We started by identifying counterpart surfaces with which the N-IDR and N-domain interacted when FtsZ self-assembled. To achieve this, we characterized FtsZ peptides and residues that were crosslinked to Bpa residues introduced into these regions by performing tandem mass spectrometry on purified FtsZ crosslinking dimers (Fig. 1a, Step 3).

Surprisingly, we observed that Bpa residues introduced to the N-IDR (e.g., at T8 or D10) crosslinked with residues in the N-domain (Fig. 1d and Supplementary Fig. 2a, and Supplementary Table 1), while Bpa residues introduced to the N-domain (e.g., at F40, K51, Q56, or S62) crosslinked with residues in the N-IDR (Fig. 1e, Supplementary Fig. 2b and Supplementary Table 1). These crosslinking results strongly suggested that the N-IDR of one subunit interacted with the N-domain of a neighboring subunit and vice versa when FtsZ self-assembled in cells (Fig. 1f).

To further clarify whether these interactions participated in longitudinal self-assembly, we performed photocrosslinking studies (under in vitro conditions) using purified Bpa variants of FtsZ at low concentrations, such that variants would assemble into single individual protofilaments[27]. The results (Fig. 1g) showed that T8 (in the N-IDR) and K51 (in the N-domain) residues both appeared to be intimately linked to longitudinal rather than lateral FtsZ assembly. This was indicated by the fact that Bpa variants of both residues formed a photocrosslinked product band ladder (lanes 4–6 and 10–12), which was greatly increased in the presence of GTP. This observation contrasted to when Bpa was introduced at residue A87 (lanes 16–18), which failed to form a band ladder, and low dimer levels remained unchanged in the presence of GTP. Residue A87 (its position is shown in Fig. 1f)

was assumed to participate in lateral interactions based on the fact that it was located between R85 and R89, both of which were reported to mediate lateral interactions with FtsZ protofilaments[21]. A87 was selected here because 87[Bpa] crosslinking products were detectable in our longitudinal interaction crosslinking analysis (Supplementary Fig. 1a).

Of note, interactions involving the N-IDR were largely missed in previous studies[28–31]. Furthermore, we noted that the N-IDR interacted with residues (e.g., F40, Q56, and S62) peripheral to those previously defined at the longitudinal interface[30]. Therefore, this posed a series of questions: did this interaction have a role in the longitudinal assembly of FtsZ, Z-ring formation, and cell division?

## The FtsZ N-IDR is indispensable for assembling a functional Z-ring

We next adopted a genetic approach to assess the biological significance of the interaction between the FtsZ N-IDR and the N-domain in bacterial cell division. To this end, we disrupted this interaction by removing the N-IDR (i.e., residues 1–10, but retained the first Met for protein expression purposes) and constructed an FtsZ[ΔNIDR] variant. Our functional complementation analyses showed that this variant failed to support cell proliferation (Fig. 2a; second row from the top), indicating that the N-IDR was indeed essential for FtsZ functioning. In particular, functional complementation analyses were performed using the *ftsZ* knockout strain, where pJSB100 was used to conditionally express wild-type FtsZ (in the presence of arabinose) to rescue its survival. The FtsZ[ΔNIDR] was constitutively expressed from the pTac plasmid in this strain to test its function.

To further assess the phenotype defect leading to failed cell proliferation, residual cells from the Δ*ftsZ* strain expressing FtsZ[ΔNIDR] were viewed under brightfield light microscopy. The micrographs (Supplementary Fig. 3a) revealed that residual cells appeared as either long filaments or cell debris, indicating that cells had grown but failed to divide. These observations implied that the N-IDR plays an indispensable role for FtsZ functioning.

We subsequently assessed the role of this self-interaction at the subcellular level. To achieve this, we first sought to directly visualize the Z-ring formed by the FtsZ[ΔNIDR] variant in live-cell imaging. We examined wild-type bacterial cells expressing both FtsZ[ΔNIDR]-mNeon-Green (in a rhamnose-inducible manner) and FtsZ[ΔNIDR] (constitutively expressed from the pTac plasmid and driven by the P[T-23105], a promoter modified from BBa_J23105), such that abnormalities in not only cell division but also Z-ring architecture could be observed upon FtsZ[ΔNIDR]-mNeonGreen induction, as previously performed by us[21,32].

Live-cell imaging revealed that the FtsZ[ΔNIDR] variant generated multiple highly distorted fluorescent FtsZ entities (subcellular assemblies) dispersed throughout the cell (Fig. 2b, middle panel), in contrast to wild-type cells expressing FtsZ[WT]-mNeonGreen (in a rhamnose-inducible manner) and FtsZ[WT] (constitutively expressed), where a normal Z-ring architecture was formed at midcell (Fig. 2b, left panel). It was worth noting that the FtsZ[ΔNIDR] variant was expressed at levels highly comparable to the FtsZ[WT], excluding the possibility that the abnormal phenotype was due to protein overexpression (Supplementary Fig. 3b).

These observations indicated that although the N-IDR interacted with sites peripheral to the previously defined longitudinal interface, it was indispensable for cell division. Given that N-IDR removal as a whole produced a lethal phenotype, we then constructed more N-IDR variants to verify and characterize more about N-IDR action mechanisms.

To this end, we adopted a screening approach by replacing residues T8 and D10 in the FtsZ N-IDR with each of the other 19 natural amino acids, and then subjected each variant to functional complementation analysis that only detected severe phenotype defects. These two residues were selected for mutational analysis as they

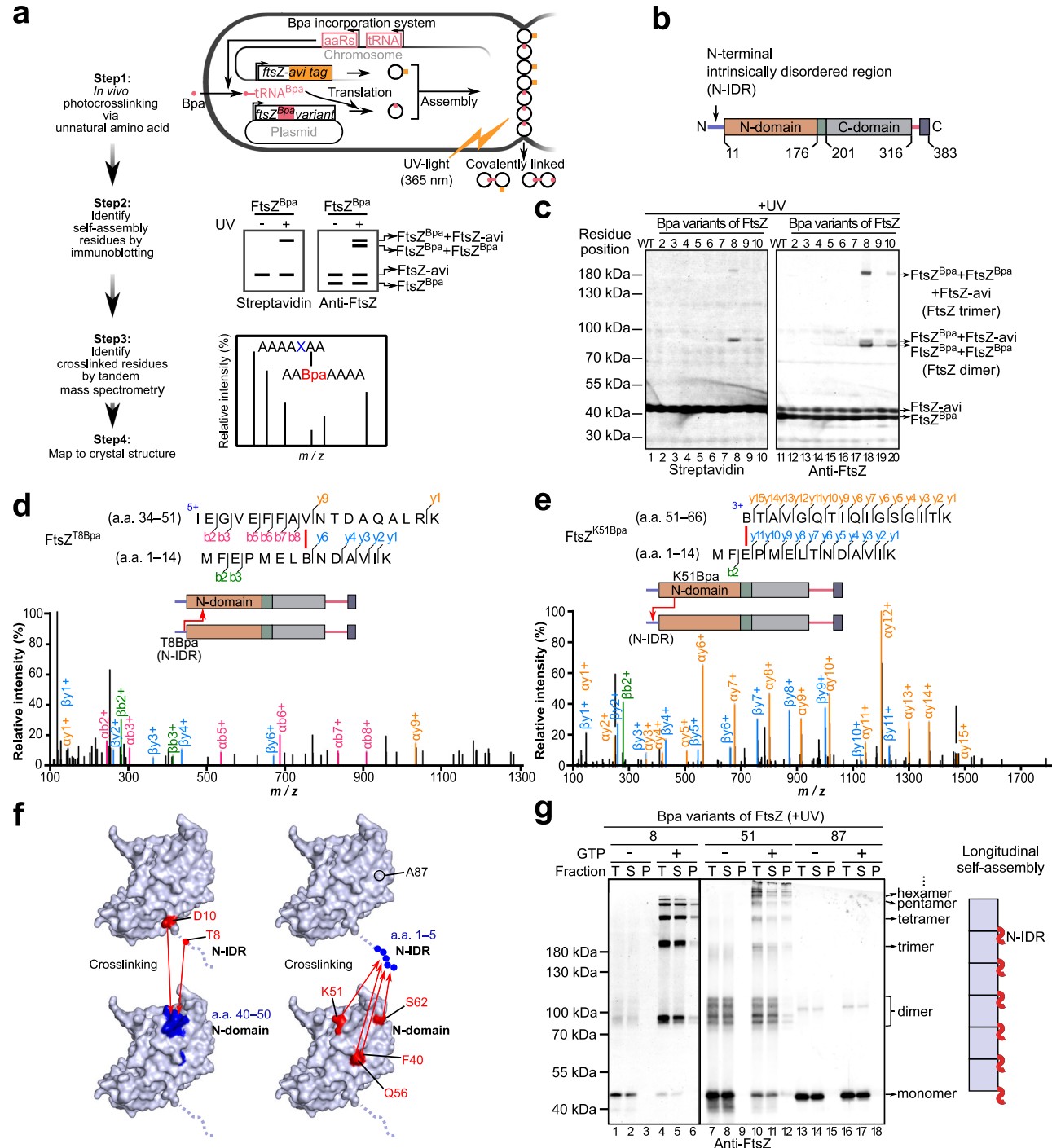

**Fig. 1 | The FtsZ N-IDR participates in longitudinal assembly. a** Schematic showing whether and how a particular residue is involved in FtsZ self-assembly using Bpa-mediated in vivo protein photocrosslinking and tandem mass spectrometry. **b** FtsZ structural domains. **c** Immunoblots showing the photocrosslinked products of indicated Bpa variants of the FtsZ N-IDR, probed with streptavidin (left) and polyclonal antibodies against FtsZ (right). Molecular weight marker positions and FtsZ forms are indicated on the left and right, respectively. **d, e** MS/MS spectra and schematics showing the two peptide fragments crosslinked via T8Bpa (**d**) and K51Bpa (**e**), respectively; B, Bpa. **f** Schematics showing crosslinked surfaces on two neighboring FtsZ subunits; based on the *E. coli* FtsZ structure for amino acids 9–316 (PDB:6UNX) and hand-drawn for amino acids 1–8. **g** Immunoblotting results (left) show photocrosslinked products formed with indicated purified Bpa variants (~1 μM), probed with polyclonal antibodies against FtsZ. Of note, FtsZ^WT assembled into single individual protofilaments at 1 μM (as indicated by TEM analysis, left panel Fig. **4**d); T total, S supernatant, P pellet, scheme (right) shows N-IDR participation in longitudinal self-assembly. Source data of **c**–**e** and **g** are provided in the Source Data file.

showed the most pronounced photocrosslinking bands (Fig. 1c and Supplementary Fig. 1a). Interestingly, when T8 was replaced by the other 19 amino acids, none appeared to interfere with Δ*ftsZ* strain proliferation. However, among the 19 D10 variants, approximately half showed significant defects in cell proliferation (Fig. 2a, right panel for

FtsZ^D10F and Supplementary Fig. 4a for the rest), and three examined variants (FtsZ^D10F, FtsZ^D10Q, and FtsZ^D10Y) showed Z-ring architecture defects (Supplementary Fig. 4b). Further super-resolution imaging analysis, as performed for FtsZ^ΔNIDR (Fig. 2b, middle panel), demonstrated that FtsZ^D10F caused the formation of a somewhat more

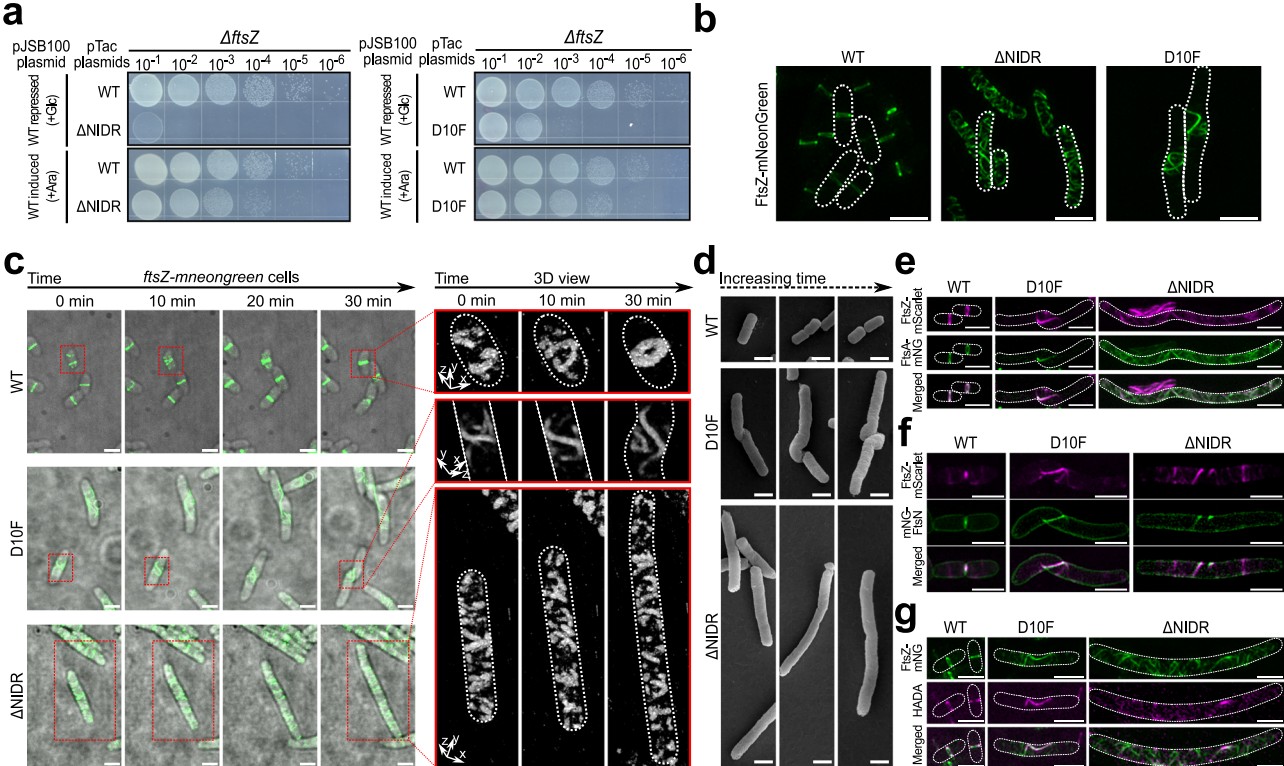

**Fig. 2 | The FtsZ N-IDR is indispensable for assembling a functional Z-ring.**
**a** Complementation results for FtsZ^ΔNIDR and FtsZ^D10F supporting Δ*ftsZ* strain growth (contains the rescue plasmid pJSB100) where FtsZ^ΔNIDR, FtsZ^D10F, or FtsZ^WT (positive control) were constitutively expressed from a pTac plasmid. **b** Structure illuminated microscopy graphs showing cells expressing indicated FtsZ forms (each fused to mNeonGreen); scale bar = 2 μm. **c** Timelapse live-cell fluorescence micrographs merged with brightfield micrographs showing cells expressing indicated FtsZ forms (each fused to mNeonGreen) which were taken every 10 min and show three-dimensional (3D) FtsZ entities at correlated time points; scale bar = 2 μm; each axis of 3D bars represents 1 μm. **d** Scanning electron micrographs

showing cells expressing indicated FtsZ forms; the three micrographs, taken for each cell form, putatively represent different moments in cell division; scale bar = 1 μm. **e**, **f** Fluorescence micrographs showing cells co-expressing indicated FtsZ forms (each fused to mScarlet) and FtsA-mNeonGreen (**e**) or mNeonGreen-FtsN (**f**); scale bar = 2 μm. **g** Fluorescence micrographs showing cells expressing indicated FtsZ forms (each fused to mNeonGreen) and incubated with HADA (a blue fluorescent D-Ala analog which labels newly synthesized peptidoglycans) for <10 s; scale bar = 2 μm. Overall cell shapes were drawn according to brightfield micrographs and are indicated by dotted white lines (**b**, **e**–**g**).

expanded and looser Z-ring architecture at midcell (Fig. 2b, right panel), in comparison to the Z-ring architecture formed by FtsZ^WT (Fig. 2b, left panel), but far less expanded and less loose than the Z-ring architecture formed by FtsZ^ΔNIDR (Fig. 2b, middle panel). Of note, FtsZ^D10F was generated at levels largely comparable to the wild-type protein (Supplementary Fig. 4c).

To identify key N-IDR roles in Z-ring formation and cell division, we performed super-resolution timelapse microscopy in three-dimensional (3D) mode to analyze FtsZ^D10F and FtsZ^ΔNIDR variants and how they facilitated abnormal Z-ring architecture. Micrographs (Fig. 2c) showed that, remarkably, if we defined the Z-ring assembling process of FtsZ^WT as dispersed (e.g., at 0 min), semi-condensed (e.g., at 10 min), and condensed (e.g., at 30 min) stages, abnormal Z-ring architectures formed by FtsZ^D10F and FtsZ^ΔNIDR could only proceed to semi-condensed and dispersed stages, respectively. In line with this, the malformed Z-ring architecture assembled by FtsZ^D10F appeared to be partially functional, such that a twisted morphology was formed at midcell, while the FtsZ^ΔNIDR assembly was non-functional and lacked any cell division morphology as observed in scanning electron micrographs (Fig. 2d).

Of note, FtsZ^D10F and FtsZ^ΔNIDR phenotypes were different when compared to common mutants which disrupted lateral interactions; namely, while the former showed strong but dispersed fluorescent entities, lateral interaction mutants commonly caused reduced, weakened, or even absent FtsZ fluorescent entities in cells[21]. These observations suggested that neither longitudinal interactions nor

lateral interactions were disrupted for FtsZ^D10F and FtsZ^ΔNIDR assembly in cells.

Surprisingly, we observed that these defective Z-ring architectures, that were mainly assembled by FtsZ^D10F and FtsZ^ΔNIDR, still recruited FtsA (Fig. 2e) and also FtsN (Fig. 2f). FtsA is a key membrane anchoring protein for FtsZ[33,34], and FtsN is the last essential protein recruited to the divisome[5,35]. Furthermore, such abnormal architectures initiated cell wall biogenesis (Fig. 2g). These observations implied that the recruitment of cell division proteins occurred even at the dispersed Z-ring formation stage.

## The FtsZ N-IDR endows the Z-ring with a proper high-level dynamicity

In view of our observations showing that these two variants were able to assemble, but did not allow assembled entities to further condense into a final Z-ring architecture, we then assessed subunit-exchange and treadmilling features of these entities in cells. Thus, subunit-exchange features were examined using fluorescence-recovery-after-photobleaching (FRAP) analysis. While entities assembled by FtsZ^WT exhibited a fluorescence recovery half-time of 4.48 s after photobleaching, those for FtsZ^D10F and FtsZ^ΔNIDR took 7.54 and 13.39 s, respectively (Fig. 3a, b).

The treadmilling features of assembled entities were assessed by recording time-dependent spatial positions using total internal reflection fluorescence microscopy (i.e., kymographs). Moving velocities, based on kymographs recorded for entities formed by FtsZ^D10F

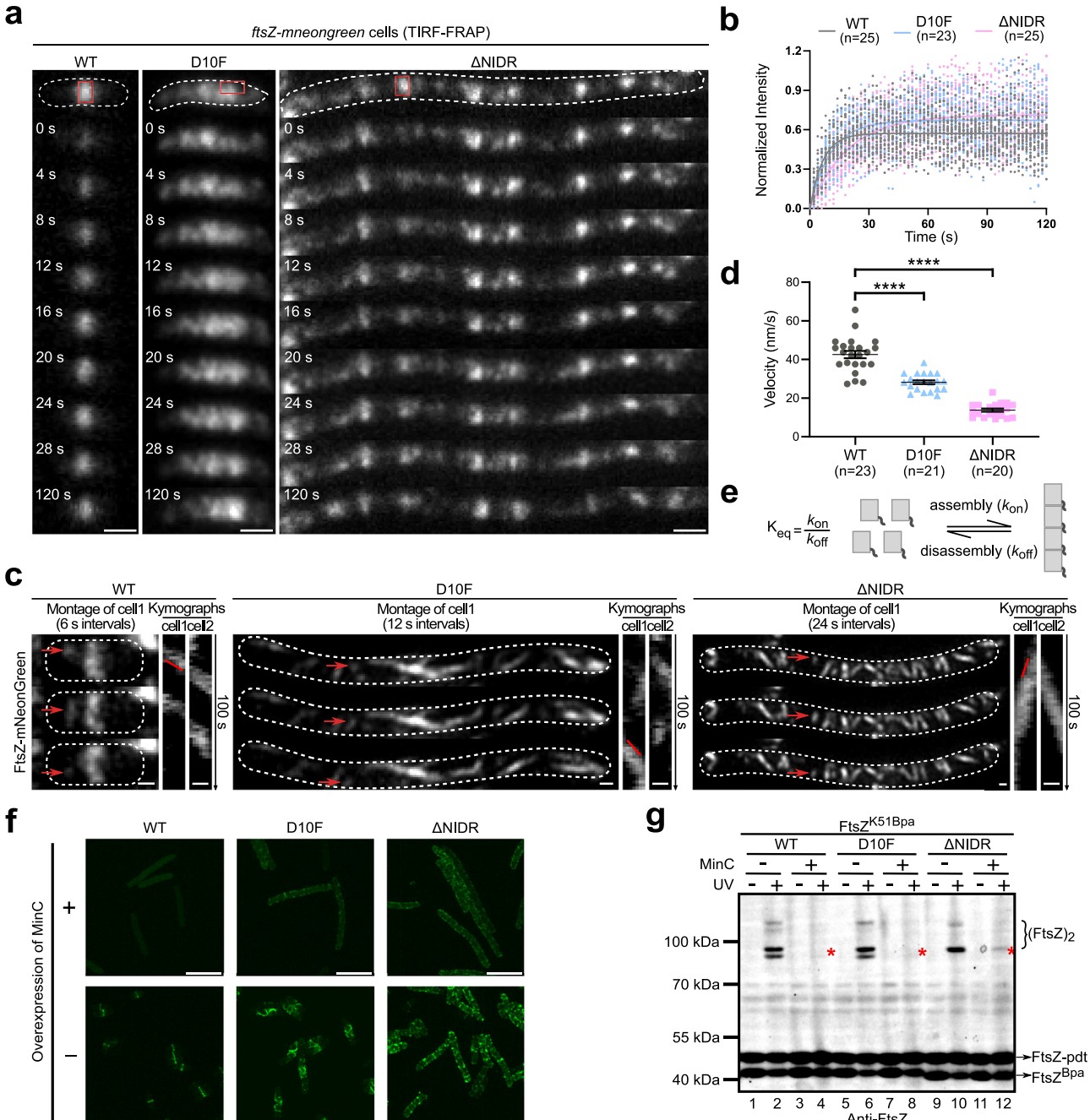

**Fig. 3 | The FtsZ N-IDR endows the Z-ring with a proper high-level dynamicity.**
**a** Timelapse imaging showing cells expressing indicated FtsZ forms (each fused to mNeonGreen) taken at indicated time points after indicated sites (red boxes) were photobleached; scale bar = 1 μm. **b** Statistical time-dependent fluorescence recovery curves; the recovery half-times for FtsZ[WT], FtsZ[D10F], and FtsZ[ΔNIDR] are 4.48, 7.54, and 13.39 s, respectively. **c** Live-cell images show the directional motion of indicated FtsZ entities, and kymographs show entities in two cells recorded every 2 s (total = 100 s). Tracked FtsZ condensates are indicated by white arrows. Overall cell shapes were drawn according to brightfield micrographs and are indicated by dotted white lines; scale bar = 0.5 μm. **d** Statistical treadmill velocities of indicated

FtsZ entities are calculated using recorded kymographs and show treadmill velocities for FtsZ[WT], FtsZ[D10F], and FtsZ[ΔNIDR] at 42.53 ± 1.886, 28.19 ± 0.9758, and 13.77 ± 0.8098 nm/s, respectively; mean ± SEM; parametric unpaired two-tailed Student's $t$-test, ****$P < 0.0001$; $n$, entities in individual cells. **e** Relationship between extent of assembly ($K_{eq}$) and rates of assembly ($k_{on}$) and disassembly ($k_{off}$) for FtsZ protofilaments. **f** Fluorescence images showing cells expressing indicated FtsZ fused forms and overexpressing MinC; scale bar = 2 μm. **g** Immunoblot showing photocrosslinked products of indicated FtsZ[K51Bpa] forms expressed in cells overexpressing MinC; red asterisks indicate residual crosslinked dimers. Source data of **b**, **d**, **g** are provided in the Source Data file.

and FtsZ[ΔNIDR] were 28.19 ± 0.98 and 13.77 ± 0.81 nm/s, respectively, and were significantly slower than the 42.53 ± 1.89 nm/s recorded for FtsZ[WT] (Fig. 3c, d). Together, these observations showed considerably reduced subunit-exchange and treadmilling rates, and revealed a significantly decreased dynamicity for the abnormal Z-ring architectures by both variants. Theoretically, a reduced dynamicity might be generated from a rate decrease in either assembly ($k_{on}$) or disassembly

($k_{off}$), which could lead to a change in the extent of assembly ($K_{eq}$) (Fig. 3e).

We subsequently assessed whether the assembly extent of the abnormal Z-ring architecture differed to FtsZ[WT]. To this end, we examined whether the entities assembled by FtsZ variants could be effectively disassembled upon MinC overexpression in cells, similar to FtsZ[WT] [36]. Live-cell fluorescence microscopy imaging data (Fig. 3f)

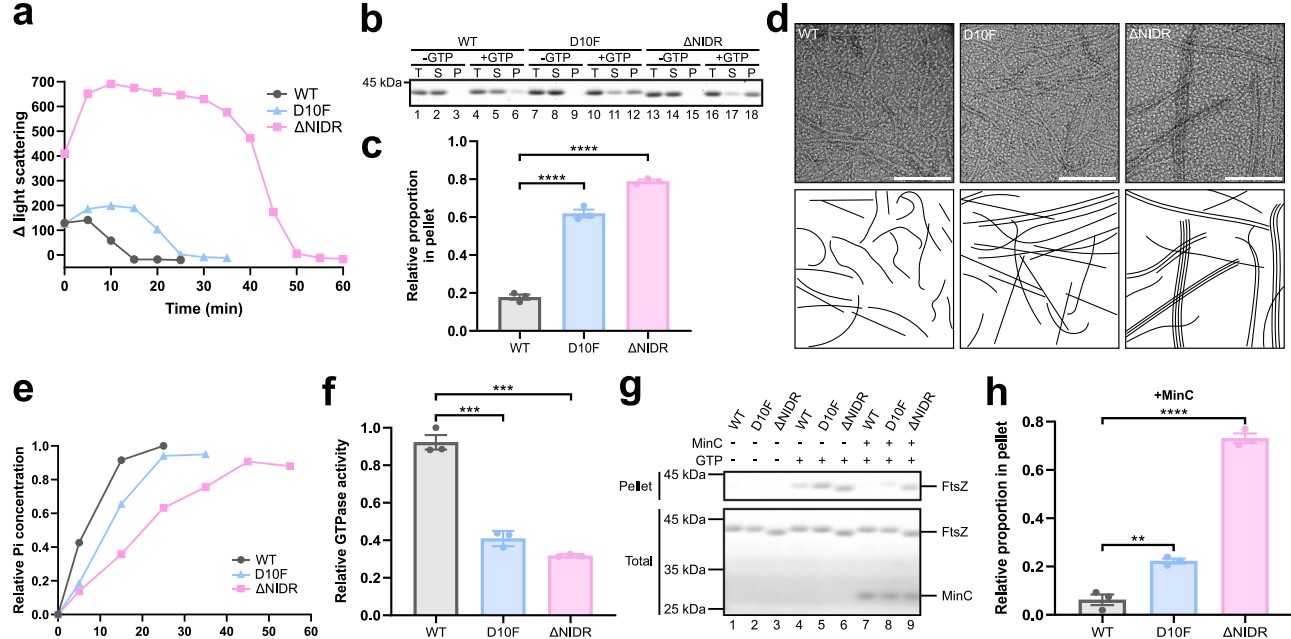

**Fig. 4 | The N-IDR ensures an adequate degree of disassembly for FtsZ protofilaments. a** Representative light scattering curves show time-dependent polymerization processes for indicated FtsZ forms, each with a starting monomeric concentration of 15 μM, initiated by 1 mM GTP, and monitored by 90° angle light scattering. **b** Coomassie brilliant blue staining shows the sedimentation analysis of polymerized FtsZ forms, with a starting monomeric concentration of 1 μM and 2 mM of GTP; T total, S supernatant, P pellet. **c** Statistically relative FtsZ proportions in pellets analyzed in (**b**); $n = 3$; mean ± SEM; parametric unpaired two-tailed Student's $t$-test, ****$P < 0.0001$. **d** Negatively-stained electron micrographs and handdrawn diagrams (beneath) of indicated FtsZ forms with a starting monomeric concentration of 1 μM; scale bar = 100 nm. **e** These curves reflect time-dependent relative Pi concentrations released from polymerization samples in (**a**). **f** Relative GTPase activity for indicated FtsZ forms with a starting monomeric concentration of 15 μM and addition of 1 mM GTP; $n = 3$, mean ± SEM; parametric unpaired two-tailed Student's $t$-test, ***$P < 0.001$; $P$ values for the different parameters are: FtsZ$^{WT}$-FtsZ$^{D10F}$ = 0.0003; FtsZ$^{WT}$-FtsZ$^{ΔNIDR}$ = 0.0001. **g** Sedimentation analysis of indicated FtsZ forms with a starting monomeric concentration of 5 μM in the presence of 5 μM MinC. **h** Statistically relative FtsZ proportions in pellets in the presence of MinC, as analyzed in (**g**); $n = 3$, mean ± SEM; parametric unpaired two-tailed Student's $t$-test, **$P < 0.01$, ****$P < 0.0001$; $P$ value for FtsZ$^{WT}$-FtsZ$^{D10F}$ = 0.0027. Source data of **a**–**c**, **e**–**h** are provided in the Source Data file.

revealed that abnormal Z-ring architectures were disassembled in a significantly less effective manner, such that residual entities were visible for FtsZ$^{D10F}$ (middle panel) and more so for FtsZ$^{ΔNIDR}$ (right panel).

In line with this, residual photocrosslinked dimer levels for FtsZ$^{ΔNIDR}$ were significantly higher than those for FtsZ$^{WT}$ or FtsZ$^{D10F}$ in MinC-overexpressed cells (Fig. 3g, lanes 4, 8, and 12). Of note, cells used for imaging analysis (Fig. 3c), but not those for crosslinking analysis (Fig. 3g), contained a *ftsZ-mneongreen* allele in the genomic DNA. Here, the FtsZ$^{K51Bpa}$ variant of FtsZ$^{ΔNIDR}$ formed crosslinked products between each other (barely visible on the gel) and with FtsZ-pdt (clearly visible on the gel; this pdt-tagged FtsZ was expressed from genomic DNA). The level of crosslinked products with FtsZ-pdt partially reflected the assembly status of FtsZ$^{ΔNIDR}$-containing protofilaments. Therefore, our crosslinking analysis partially reflected live-cell imaging results.

These observations strongly suggested that the assembly extent of entities formed by FtsZ variants was significantly increased, and hinted at decreased disassembly rates. This partially explained why FtsZ$^{ΔNIDR}$ and FtsZ$^{D10F}$ variant entities were respectively stalled at the dispersed and semi-condensed stages, given the key role of MinC in forming a condensed Z-ring architecture at midcell by promoting FtsZ disassembly at other sites in the cell[13].

Collectively, these in vivo data appeared to indicate that the N-IDR had important roles endowing Z-ring architecture with a proper high-level dynamicity, and also an appropriate extent of assembly, allowing MinC to disassemble FtsZ protofilaments at sites beyond midcell.

## The N-IDR ensures an adequate degree of disassembly for FtsZ protofilaments

To ascertain that the N-IDR exerted its effects on Z-ring assembling by playing direct rather than indirect roles, we performed in vitro studies comparing the assembly/disassembly features among FtsZ$^{D10F}$, FtsZ$^{ΔNIDR}$, and FtsZ$^{WT}$ using purified proteins. We demonstrated the following under in vitro conditions. Firstly, FtsZ variants exhibited significant assembly differences when roughly monitored by 90° angle light scattering analyses (Fig. 4a). Sedimentation analysis at a low concentration (~1 μM) of monomeric FtsZ protein revealed that ~78.9% of FtsZ$^{ΔNIDR}$ and 61.2% of FtsZ$^{D10F}$ were present in the pellet fraction, significantly higher than the 17.8% of FtsZ$^{WT}$ in the pellet fraction (Fig. 4b, c), consistent with the light scattering findings presented in Fig. 4a. Negative staining electron microscopy showed that protofilaments formed by FtsZ$^{D10F}$ were slightly longer (Fig. 4d, middle panel), and those formed by FtsZ$^{ΔNIDR}$ were even longer (Fig. 4d, right panel). Furthermore, FtsZ$^{ΔNIDR}$ protofilaments formed bundles under these in vitro conditions (Fig. 4d, right panel), which possibly occurred due to enhanced longitudinal interactions, which in turn would promote lateral protofilament interactions, consistent with previous observations[21]. These may explain significant increases in light scattering results. Secondly, variants with an altered N-IDR showed prolonged assembly duration times (Fig. 4a) and reduced GTPase activity (Fig. 4e, f), indicating a defect in FtsZ disassembly. Additionally, while showing similar response mechanisms to wild-type FtsZ, mutant FtsZ assemblies buffered against MinC-mediated disassembly, apparently due to their higher extent of assembly. This was indicated by the fact that the level of assembly was decreased for all three forms upon the addition of MinC, but to a different degree (Fig. 4g, h).

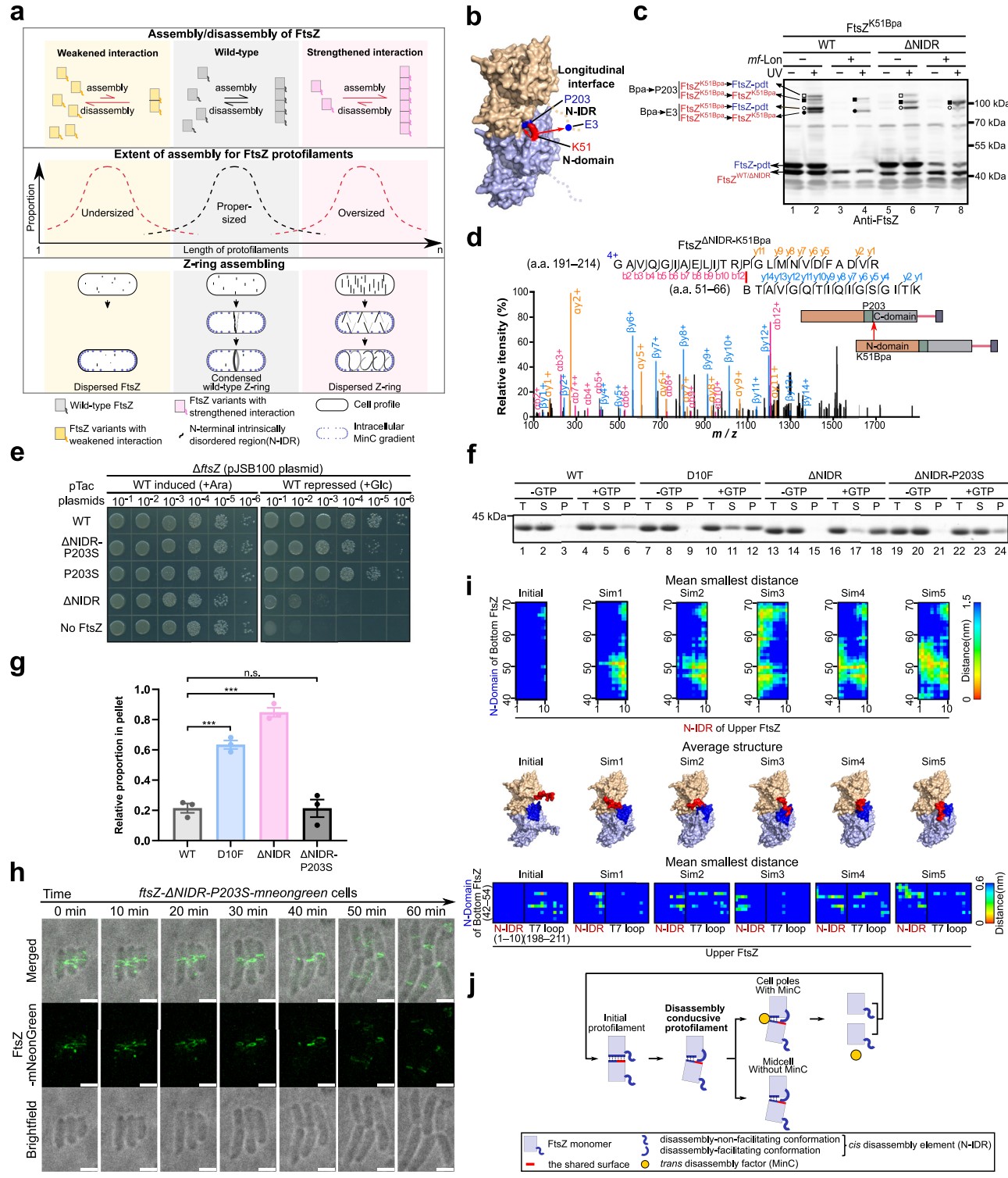

## The FtsZ N-IDR acts via competing interactions

Taken together, our in vivo and in vitro data strongly suggested that the N-IDR acted as a *cis* element endowing an adequate degree of disassembly to FtsZ protofilaments, allowing extrinsic disassembly factors such as MinC to effectively depolymerize the protofilaments at sites away from midcell (Fig. 5a) (middle and right columns; for behavior comparisons of FtsZ variants with weakened self-assembly interactions, see the left column[21,37–40]).

We next deciphered how the N-IDR acted as a *cis* disassembly element. In this regard, we observed that G20, N24, and K51 residues in the N-domain apparently participated in interactions with two, rather

than one, different surfaces (Fig. 5b); one involving the N-IDR and the other involving the longitudinal interface (including P203), as demonstrated by in vivo photocrosslinking (Fig. 5c and Supplementary Fig. 1b) and tandem mass spectrometry analyses (Supplementary Tables 1, 2). These observations were further confirmed by the fact that in FtsZ$^{\Delta NIDR}$, the interaction between K51 and P203 was strengthened by in vivo photocrosslinking (Fig. 5c, lanes 4 and 8; top bands indicated by a solid square) and tandem mass spectrometry (Fig. 5d and Supplementary Table 3). Of note, while wild-type FtsZ encoded by genomic DNA was essential for the proliferation of cells expressing FtsZ$^{\Delta NIDR}$, its presence interfered with crosslinked product formation specific to

**Fig. 5 | The FtsZ N-IDR acts via competing interactions. a** FtsZ may polymerize into proper-sized (middle column, wild-type), undersized (left, weakened assembly), or oversized (right, strengthened assembly) entities at protein (top row), protofilament (middle), and subcellular (bottom) levels. **b** A structure model showing that residue K51 in one subunit interacts with E3 or P203 in a neighboring subunit as suggested by a mass spectrometry analysis of in vivo photocrosslinked products as shown in Supplementary Tables 1, 2; the structure model was based on threading the *E. coli* FtsZ sequence through one of the *S. aureus* crystal structures in the tense conformation for amino acids 11–315 (PDB:3VOA) and hand-drawn for amino acids 1–10. **c** Immunoblot showing photocrosslinked FtsZ$^{K51Bpa}$ products in cells expressing FtsZ-pdt, which was degraded when proteinase *mf*-Lon was induced; molecular weight marker positions and FtsZ forms are indicated on the right and left, respectively. **d** MS/MS spectrum and schematics showing the identification of peptide fragments crosslinked with K51Bpa in FtsZ$^{\Delta NIDR}$; B, Bpa. **e** Functional complementation analysis results of FtsZ$^{\Delta NIDR-P203S}$ and FtsZ$^{P203S}$. **f** Coomassie brilliant blue staining shows sedimentation fractions containing polymerized FtsZ forms with a starting monomeric concentration of 3 μM and

2 mM of GTP; T total, S supernatant, P pellet. **g** Statistically relative FtsZ proportions in pellets, as analyzed in (**f**); $n = 3$; mean ± SEM; parametric unpaired two-tailed Student's *t*-test, ***$P < 0.001$, n.s.no significance; $P$ values for the different parameters are: FtsZ$^{WT}$-FtsZ$^{D10F}$ = 0.0005; FtsZ$^{WT}$-FtsZ$^{\Delta NIDR}$ = 0.0001. **h** Timelapse images (brightfield, fluorescence, and merged) showing *ftsZ-ΔNIDR-P203S-mneongreen* cells; scale bar = 2 μm. **i** Five heat maps depicting the mean smallest distance between the N-IDR of one FtsZ subunit and the N-domain of its neighboring subunit, each representing the last 50 ns from an independent 150 ns molecular dynamics simulation (top row). Five average conformations of the FtsZ dimer structure based on the last 50 ns of each simulation are displayed (middle row). Five heat maps depicting the mean smallest distance between the N-IDR (residues 1–10) or the T7 loop (residues 198–211) of one FtsZ subunit, and the N-domain (residues 42–54) of its neighboring subunit during the last 50 ns of each simulation are displayed (bottom row). For comparison, the initial conformation (prior to energy minimization and subsequent steps) is also presented. **j** Schematic showing the interaction-modulating mechanism for Z-ring assembling. Source data of **c**, **d**, **f**, **g**, **i** are provided in the Source Data file.

FtsZ$^{\Delta NIDR}$ (see bands in lanes 2 and 6 in Fig. 5c). To circumvent this interference, we introduced a pdt-tag into the wild-type FtsZ protein. Inducible *mf*-Lon protease expression then degraded FtsZ-pdt[41], effectively eliminating or reducing the interference (see bands in lanes 4 and 8 in Fig. 5c).

In light of these observations, we examined whether a P203 substitution by any other natural amino acid (to weaken its interaction with residue K51) could reduce the self-assembly interaction of FtsZ$^{\Delta NIDR}$, thus allowing it to resume its support of Z-ring condensation and cell division. Our functional complementation analysis demonstrated that the FtsZ$^{\Delta NIDR}$ lethal phenotype was most alleviated when P203 was replaced by L, E or S (Fig. 5e for the P203S variant, and Supplementary Fig. 6a for the others). Accordingly, we also showed that such a revertant, especially FtsZ$^{\Delta NIDR-P203S}$, allowed cells to resume a largely normal state in terms of cell morphology (Supplementary Fig. 6b), in vitro extent of assembly (Fig. 5f, g), and in vivo Z-ring assembling (Fig. 5h). Of note, we also observed that the P203S substitution exhibited only marginal effects on FtsZ$^{WT}$ properties, under either in vivo (Fig. 5e and Supplementary Fig. 6c) or in vitro (Supplementary Fig. 6d, e) conditions. This was in line with the fact that residue K51, that was in close contact with residue E3, had a weak contact with P203, and thus replacing it with Ser, would exert little effect on wild-type FtsZ assembly (Fig. 5c, lanes 2 and 4), but became unnecessarily strong in entities assembled by FtsZ$^{\Delta NIDR}$.

Intriguingly, GTPase activity levels in FtsZ$^{\Delta NIDR-P203S}$ remained comparable with FtsZ$^{\Delta NIDR}$, rather than FtsZ$^{WT}$ (Supplementary Fig. 6f). This was consistent with previous data showing that an FtsZ mutant, with ~10% of the GTPase activity of the wild-type protein, remained largely functional[19]. In light of this, the GTPase activity of the FtsZ protein requires further characterization.

To learn more about N-IDR actions in FtsZ protofilaments, we performed molecular dynamics (MD) simulation studies. In particular, we performed five independent 150 ns MD simulations based on an FtsZ dimer to examine how N-IDR contacted its neighboring subunit. To assess the adequacy of simulation timescales, we analyzed root mean square deviation (RMSD) and radius of gyration (Rg) values of the N-IDR over five independent 150 ns trajectories. As shown (Supplementary Fig. 7), both metrics plateaued after ~100 ns with fluctuations (standard deviation) within 0.03 nm (RMSD) and 0.02 nm (Rg), indicating that conformational equilibrium was reached within the simulation timescale. Consistently elevated values in both root mean square fluctuation (RMSF) profiles and Rg distributions across all replicates (except Sim4) indicated that the N-IDR retained high structural plasticity, and preferentially adopted extended conformations (Supplementary Fig. 7). The final 50 ns mean smallest distance frequency plot (Fig. 5i, top row) and average structures from the final 50 ns for each simulation (Fig. 5i, middle row) revealed the following.

(1) Starting from an initial extended conformation, the N-IDR transitioned to a state that contacted various surfaces on the monomer itself and on a neighboring FtsZ subunit. While exhibiting consistent trends, each simulation displayed distinct N-IDR conformations and contacting locations. This suggested that the N-IDR was somehow flexible and existed in multiple possible states. (2) The N-IDR interacted with several residues within the N-domain of a neighboring FtsZ subunit. Notably, residue pair contacts between the N-IDR and N-domain were clearly observed in simulation analysis (Fig. 5i, top row), which agreed with our in vivo photocrosslinking analysis and mass spectrometry data (summarized in Fig. 1f).

Further, mean smallest distance analysis between the N-domain of one FtsZ subunit and the N-IDR or T7 loop of a neighboring subunit during the final 50 ns of each simulation revealed the following contact patterns. Simulation results indicated that, in the initial state, the N-IDR and N-domain had no direct contact, while the T7 loop maintained close direct contact with the N-domain. However, in the last 50 ns of simulations, when the system approached stabilization, the N-IDR contacted the N-domain to varying degrees and at different sites, whereas contacts between the T7 loop and N-domain weakened to different extents, particularly for Sim1 and Sim3. These interaction changes are consistent with our in vivo photocrosslinking data (Fig. 5c).

Collectively, these data suggested that the N-IDR acted as a *cis* disassembly element by partially disrupting longitudinal interactions by occupying one side of the interface.

### The FtsZ N-IDR in other bacteria also functions as a *cis* disassembly element

We next examined whether the FtsZ N-IDR in other bacteria also acted similarly. To this end, we removed corresponding FtsZ sequences from *Bacillus subtilis* and *Staphylococcus aureus* (both are Gram-positive bacteria that are distantly related to *E. coli* and whose FtsZ has a low amino acid sequence identity with *E. coli* FtsZ (Supplementary Fig. 8a)) and assessed assembly properties under in vitro conditions. These deletions significantly increased assembly levels similar to *E. coli* FtsZ, as shown by 90° angle light scattering (Supplementary Fig. 8b) and sedimentation analyses (Supplementary Fig. 8c, d).

### Discussion

We conducted this study to acquire and decipher information potentially missing from previous structural studies[28–31], but important for FtsZ to assemble into a dynamic Z-ring architecture required for cyclic cell division processes. To achieve this, we performed multiscale analyses, including in vivo protein photocrosslinking, genetic mutation analyses, live-cell imaging, in vitro assembly, and electron microscopy. One piece of information that was practically missing

from prior studies was that upon cooperating with *trans* (extrinsic) factors such as MinC, the FtsZ N-IDR acted as a *cis* (intrinsic) disassembly element critical for Z-ring architecture condensation.

It is noteworthy that our revelation showing the N-IDR role in FtsZ assembly was supported by previous observations. For example, it was shown that the FtsZ N-IDR deletion in cyanobacteria reduced GTPase activity and promoted the formation of more stable protofilaments[42]; additionally, cryo-EM studies indicated that in protofilaments formed by FtsZ bound to a specific monobody, the N-IDR existed as an α-helix that partially contacted the monobody as well as the N-domain of the neighbouring subunit[43]. Furthermore, it was shown that the N-IDR was more protected from ClpP protease degradation when the *B. subtilis* FtsZ formed a protofilament, hinting at possible participation in certain interactions[44].

In light of our observations, we propose an interaction-modulating mechanism to explain how the N-IDR acts as a *cis* element to facilitate effective FtsZ protofilament disassembly at cell poles, and Z-ring architecture condensation at midcell (schematically represented in Fig. 5j). Specifically, in the initial FtsZ protofilament that is likely in its T (tense) state[23,30], the N-IDR is in a disassembly-non-facilitating conformation. Following the formation of the protofilament, the N-IDR switches to a disassembly-facilitating conformation and thus contacts the shared surface (containing residue K51) and partially disrupts the longitudinal interface, transforming the FtsZ protofilament to a state conducive to disassembly, roughly corresponding to its R (relax) state[23,30]. The presence of the *trans* disassembly factor MinC at cell poles further drives FtsZ protofilaments to disassemble, while a lack of MinC at midcell allows FtsZ to assemble into a Z-ring architecture there. Therefore, these events may help explain what happens for Z-ring assembling at both pre- and post-division stages.

Importantly, we identified an intrinsically disordered rather than an ordered structure element that has uniquely evolved to play this disassembly role, without interfering with the assembly process per se. It would not be unexpected if similar intrinsically disordered *cis* disassembly elements were also present in other proteins that assemble into dynamic multi-protein complexes in cells[45,46].

In light of these observations, we hypothesize that the N-IDR may additionally modulate FtsZ GTPase activity in promoting FtsZ protofilament disassembly. FtsZ protofilament formation, which is triggered by GTP binding[47], simultaneously allows the N-IDR to engage with the shared surface around residue K51. This, in turn, weakens interactions between the two surfaces, respectively containing K51 and P203. As the surface containing P203 is located in the T7 loop critical for FtsZ GTPase activity[48], such a conformational readjustment may promote FtsZ GTPase activity. Subsequent GTP hydrolysis[49,50] would generate a conformational change disrupting longitudinal interactions and making FtsZ protofilaments more susceptible to disassembly. In support of this hypothesis, we observed that N-IDR removal decreased GTPase activity (Fig. 4e, f) and also impaired FtsZ protofilament disassembly (Fig. 4a, g, h). The fact that FtsZ$^{\Delta NIDR-P203S}$ GTPase activity remained low but largely functional in supporting cell proliferation indicated that disrupting longitudinal interactions appeared to be more important than modulating the GTPase activity for N-IDR to act as a *cis* disassembly element.

This hypothesis helps explain why substituting P203 with another residue, like serine, largely reverted the lethal phenotype of the FtsZ$^{\Delta NIDR}$ variant. Specifically, the functional restoration of this variant may have resulted from the fact that a single P203S substitution caused enough disruption to the longitudinal interface such that FtsZ protofilaments resumed their disassembly susceptibility to levels comparable with the FtsZ$^{WT}$, even without N-IDR participation (Fig. 5e–h).

Interestingly, a recent cryo-EM study revealed that ZapA, a protein that binds FtsZ straight protofilaments, sequestered the N-IDR away from the longitudinal interaction surface of FtsZ[51]. This in vitro observation suggested that the N-IDR may act as a target for proteins like ZapA to stabilize FtsZ protofilaments. It follows that when the N-IDR is released from ZapA, FtsZ protofilament disassembly is facilitated. Furthermore, a recent study showed that during natural interbacterial antagonism, *Yersinia pseudotuberculosis* secreted a protein effector, CccR, which covalently modified residue D8 with an adenosine monophosphate (AMP) group within the FtsZ N-IDR in *Enterobacteriaceae*, inducing cell filamentation[52]. Together, these observations implied that the N-IDR acted as an effective and general target for bacteria to modulate their cell proliferation or that of other species.

In summary, the extreme N-terminus of FtsZ acts as a *cis* disassembly element, which disrupts longitudinal assembly by contacting a peripheral site and providing a stumbling block (forced interaction that does not generate an interaction force) near the interface, tipping the balance more toward polymer disassembly. This, in turn, enables Z-ring formation only at midcell (rather than elsewhere) upon modulation by certain *trans*-acting factors (e.g., the *E. coli* MinC protein).

## Methods

### Bacterial strains, plasmids, and cell culturing

All *E. coli* strains are described (Supplementary Table 4). All strains were derived from the *E. coli* strain BW25113, which has the following genotype: F⁻, DE(*araD-araB*)567, *lacZ*4787(del)::*rrnB-3*, LAM⁻, *rph-1*, DE(*rhaD-rhaB*)568, *hsdR*514[53]. Genome modifications were performed using the λ-red genomic recombination system[54]. All plasmids are listed in Supplementary Table 5, and the primers required for plasmid construction are listed in Supplementary Data 1. All genome modifications and newly generated plasmids were confirmed by DNA sequencing. Bacterial cells were cultured at 37°C in Lurai Bertani (LB) medium (10 g/L tryptone, 5 g/L yeast extract, and 5 g/L NaCl) along with corresponding antibiotics for plasmid selection (e.g., 50 μg/mL kanamycin, 34 μg/mL chloramphenicol, 100 μg/mL ampicillin, or 13 μg/mL gentamycin) and shaken at 260 rpm for indicated durations.

### In vivo and in vitro protein photocrosslinking analysis

All in vivo protein photocrosslinking analyses were performed in strains derived from the LY928 *E. coli* strain, the genome of which encodes the orthogonal tRNA (with an anticodon that recognizes the UAG codon introduced at the Bpa incorporation site) and aminoacyl-tRNA synthetase, which incorporates a photoactivatable unnatural amino acid *p*-benzoyl-L-phenylalanine (Bpa, MREDA, M058191-5g), as previously described[21,22].

For in vivo whole-residue-site photocrosslinking FtsZ analyses, an Avi tag was fused to the C-terminus of the wild-type FtsZ (expressed from genomic DNA), while each of the 382 FtsZ$^{Bpa}$ variants were expressed from plasmids at levels comparable to endogenous FtsZ. Specifically, a plasmid containing the Tet promoter was used to express Bpa variants of the FtsZ 2–230 residue region; here, the anhydrotetracycline inducer was not added, thus, the protein was produced via leaky expression.

Another pTac plasmid containing the P$_{T-23105}$ promoter was used to generate FtsZ variants where residues 231–383 were individually replaced by Bpa, such that variant protein expression only occurred in the presence of TetR and anhydrotetracycline. This stringent control strategy prevented the production of potentially toxic truncated FtsZ forms (which might be produced when the introduced TAG codon was read as a stop codon rather than one encoding Bpa) that would cause lethal phenotypes before inducer addition.

The P$_{T-23105}$ promoter was modified from the BBa_J23105 promoter (selected from the Anderson promoter collection (parts.igem.org/Promoters/Catalog/E.coli/Constitutive)). The middle sequence between −10 and −35 sequences (bold) in the P$_{23105}$ promoter was then replaced with the *tetO* operator sequence (underlined), and the λ t1 transcriptional terminator sequence (italics) was added upstream of

the $P_{23105}$ promoter. This allowed the promotor (*CGCAAAAAAC CCCGCTTCGGCGGGGTTTTTTCGC*TAAGGGATTTTGGT**TTTACT**CCC-TATCAGTGATAGA**TACTAT**) to function constitutively and express FtsZ (or variants) at levels comparable with endogenous FtsZ in a strain lacking TetR (a transcription repressor binding the *tetO* sequence), but function in a regulatable manner in a strain expressing TetR (from pYR5C-tetR plasmid).

All cells were cultured at 37 °C in 96-deep-well plates containing LB medium plus 200 µM Bpa. Cells expressing FtsZ variants with Bpa introduced in the 231–383 residue region were grown to an $OD_{600\,nm}$ of 0.5–0.6 before variant protein expression was induced by anhydrotetracycline (0.2 µg/mL) and incubating cells for an additional 3 h. Cells expressing Bpa variants from the leaky plasmid were also cultured for the same period.

Cells (strain: LY928-*ftsZ-pdt*) expressing K51Bpa variants of FtsZ$^{WT}$, FtsZ$^{D10F}$, and FtsZ$^{\Delta NIDR}$ from the pTac plasmid were cultured to an $OD_{600\,nm}$ of 1–1.2, diluted two-fold before adding anhydrotetracycline (0.2 µg/mL) to induce MinC expression for 2 h.

Cells (strain: LY928-*ftsZ-pdt*) expressing K51Bpa variants of FtsZ$^{WT}$ and FtsZ$^{\Delta NIDR}$ were cultured to an $OD_{600\,nm}$ of 0.5–0.6 before adding anhydrotetracycline (0.2 µg/mL) to induce the *mf*-Lon protease[41] for an additional 2 h to degrade endogenous FtsZ-pdt.

All cells aforementioned were then irradiated with UV-light (365 nm) for 10 min at room temperature using a Hoefer UVC 500 Crosslinker, then collected by centrifugation at 7000×*g* before immunoblotting.

In vitro protein photocrosslinking with purified Bpa variants of FtsZ at low concentrations such that they would assemble into single individual protofilaments was performed as follows. Purified FtsZ proteins were first incubated at 30 °C for 5 min in polymerization buffer (100 mM NaMES, 5 mM MgAc, and 1 mM egtazic acid [EGTA], pH 6.5) after adding 1 mM GTP. The reaction solution was then irradiated with UV point-light (365 nm, 6000 mW/cm$^2$) for 1 min at room temperature, ultra-centrifuged (450,000×*g*, 4 °C, 15 min), and immunoblotted.

## Immunoblotting

Cell samples were resolved using high-throughput SDS-PAGE that we previously developed[22]. Briefly, for each Bpa variant, cells collected from 200 µL of bacterial culture by centrifugation were suspended in 50 µL of sample buffer and heated for 15 min at 95 °C in 96-well plates. Lysed extracts of all 382 cell samples (plus one without Bpa incorporation in FtsZ; i.e., a negative control) were loaded in a single horizontal gel. For high-resolution analysis, the same cell samples were resolved by vertical SDS-PAGE (using Bis-Tris-MOPs as the buffer system and loading only 24 samples on each gel).

After electrophoresis, the gel was blotted and probed with anti-FtsZ antibodies (Rabbit antiserum against purified EcFtsZ protein. Dilution: 1:10,000) and IRDye®800CW Goat anti-Rabbit IgG (H + L) (Bioss, bs-40295G-IRDye8. Dilution: 1:10,000) to detect FtsZ, or with Streptavidin-Alexa Fluor®680 (Bioss, bs-0437P-AF680. Dilution: 1:10,000) to detect the Avi tag (linked only to endogenous wild-type FtsZ). Protein bands on polyvinylidene fluoride membranes were then visualized, scanned by Typhoon NIR (Amersham, USA), and processed using GNU image manipulation 2.10 software.

For high-throughput SDS-PAGE, loaded protein samples were not normalized. Cells were cultured in four 96-well plates (due to large sample numbers and low protein quantities). For regular SDS-PAGE (except those shown in Fig.1c and Supplementary Fig. 1b), protein samples were normalized by measuring the optical density of the cells.

## Protein purification

To obtain crosslinked products for mass spectrometry analysis, a His tag was fused to the C-terminal end of Bpa variants of FtsZ that were expressed from plasmids. Cells were grown to an $OD_{600\,nm}$ of 0.5–0.6 before the expression of Bpa variants of FtsZ was initiated by adding

anhydrotetracycline (0.2 µg/mL) and culturing cells for another 3 h. This generated a FtsZ$^{Bpa}$-His variant at ~20-fold higher than the constant leaky levels of the FtsZ$^{Bpa}$ variant (used for in vivo photocrosslinking), which caused minimal changes in photocrosslinked product patterns. These cells were then poured into a tray, irradiated with UV-light (365 nm) for 15 min at room temperature to generate photocrosslinked products, and then collected by centrifugation at 10,000×*g*. FtsZ$^{Bpa}$-His variant photocrosslinking products were purified by nickel affinity chromatography before mass spectrometry was performed to identify crosslinked peptides and residues. Briefly, cells collected by centrifugation were resuspended in binding buffer (20 mM phosphate buffer, 500 mM NaCl, and 30 mM imidazole, pH 7.4) and lysed using a French press at 1000 MPa. Crude cell extracts were then centrifuged at 13,000×*g*, and supernatants applied to a 1 mL HisTrap 6FF resin column (Yeason, China) on an ÄKTA Pure L chromatography system (Cytiva, USA) for affinity purification. Fractions were collected at a flow rate of 1 mL/min. The resin was subsequently washed in binding buffer before FtsZ monomer and crosslinking products were eluted in elution buffer (20 mM phosphate buffer, 500 mM NaCl, and 500 mM imidazole, pH 7.4). The samples were then mixed with loading buffer and resolved by vertical SDS-PAGE.

All *E. coli* FtsZ protein forms that were subjected to in vitro polymerization and protein photocrosslinking analyses were purified using previous methods[55]. Briefly, *E. coli* BL21(DE3) cells transformed with plasmid pET28a derivatives expressing FtsZ variants were grown at 37 °C in 1 L of kanamycin-containing LB medium to an $OD_{600\,nm}$ of 0.5–0.6. For FtsZ$^{Bpa}$ variants, an additional pBAD-BpaRS-tRNA$^{Bpa}$ plasmid, expressing the orthogonal tRNA and aminoacyl-tRNA synthetase to incorporate Bpa, was induced by adding 0.05% arabinose for another 1 h. Next, isopropyl-1-thio-$\beta$-D-galactopyranoside (IPTG) was added to a final concentration of 1 mM, and cells were grown for an additional 3 h. Cells were then harvested, washed twice, and resuspended in 20 mL of PEM buffer (50 mM PIPES, 5 mM MgCl$_2$, and 1 mM ethylenediaminetetraacetic acid [EDTA], pH 6.5). After cells were lysed in a French press at 1000 MPa, FtsZ-containing supernatants were collected by centrifugation at 100,000×*g* for 2 h at 4 °C to remove membrane fractions. FtsZ components were then isolated as the pellet fraction using two precipitation rounds induced by the addition of 1 mM GTP and 20 mM CaCl$_2$. The final FtsZ pellet was resuspended in 5 mL of PEM buffer. Protein samples were further purified using 5 mL Hi-Trap Q-Sepharose columns (Yeason) after equilibration in Tris-glycerol buffer (50 mM Tris-HCl, 50 mM KCl, 5 mM MgCl$_2$, 1 mM EDTA, and 10% glycerol, pH 8.0), mainly to remove nucleic acids. FtsZ proteins were then eluted in a 50 mM–1 M KCl gradient at a flow rate of 1 mg/mL.

The MinC protein was purified as follows. BL21(DE3) *E. coli* cells transformed with plasmid pET28a derivatives expressing MinC-His were grown at 37 °C in 1 L of kanamycin-containing LB medium to an $OD_{600\,nm}$ of 0.5–0.6. Then, IPTG was added and cells were grown for an additional 3 h before they were collected by centrifugation, after which MinC-His was purified in the same way as FtsZ$^{Bpa}$-His variants.

Untagged, full-length FtsZ$^{WT}$ and FtsZ$^{\Delta NIDR}$ from *Bacillus subtilis* (BsFtsZ) and *Staphylococcus aureus* (SaFtsZ) were purified based on methods described before[56] but with major modifications. The DNA sequences encoding BsFtsZ$^{WT}$ and SaFtsZ$^{WT}$ were designed based on the protein sequences stored in the protein databank (as Uniprot P17865 and P0A029, respectively), chemically synthesized (with codons being optimized for *E. coli* expression) inserted into the pET28a plasmid and expressed in *E. coli* BL21(DE3) cells. The BsFtsZ$^{\Delta NIDR}$ and SaFtsZ$^{\Delta NIDR}$ were constructed with residues 2–11 deleted. The cells were grown in LB medium containing 100 µg/mL kanamycin at 37 °C to an $OD_{600\,nm}$ of 0.5–0.6, and protein expression was induced with 1 mM IPTG for 2 h. Cells were then harvested, resuspended in 50 mL TKM buffer (50 mM Tris-HCl, 50 mM KCl, 1 mM EDTA, 5 mM MgCl$_2$ and 10% glycerol, pH 8.0), lysed by a French press

at 1000 MPa, centrifuged at 100,000 × g for 2 h at 4 °C to remove membrane fractions. The supernatants were collected, and FtsZ-containing fractions were then precipitated with 35% $(NH_4)_2SO_4$ (20,000×g, 10 min). For SaFtsZ$^{WT/\Delta NIDR}$, resuspended in 50 mL MES-KOH buffer (50 mM MES, 5 mM $MgCl_2$ and 10% glycerol, pH 6.5), transferred to Bs/SaFtsZ polymerization buffer (50 mM NaMES, 2.5 mM MgAc, 50 mM KAc and 1 mM EGTA, pH 6.5)[57]. For BsFtsZ$^{WT/\Delta NIDR}$ purification, protein samples were resuspended in 50 mL MES-KOH buffer, and $(NH_4)_2SO_4$ was added to a concentration of 1.15 M. The protein samples were afterwards subjected to hydrophobic interaction chromatography (Source 15ISO, Amersham) in 50 mM $KH_2PO_4$/ $K_2HPO_4$-KOH, 1 mM EDTA, eluted with 1.15–0 M $(NH_4)_2SO_4$ gradient at 0.6 mL/min. The BsFtsZ$^{WT/\Delta NIDR}$ was further purified with a 5 mL anion-exchange chromatography Hi-Trap Q-Sepharose column (Yeason) in Tris-glycerol buffer (50 mM Tris-HCl, 50 mM KCl, 5 mM $MgCl_2$, 1 mM EDTA, and 10% glycerol, pH 8.0), eluted in a 0–1 M KCl gradient at 1 mL/min.

Before use, proteins were desalted using 5 mL Hi-Trap desalting columns (Cytiva) equilibrated with polymerization buffer.

## Mass spectrometry and crosslinked peptide and site analyses using pLink2 software

Crosslinked protein product bands, resolved by SDS-PAGE and visualized by Coomassie blue staining, were excised and subjected to Orbitrap Fusion Lumos Tribrid liquid chromatography mass spectrometry (Thermo Fisher Scientific, USA) analysis. To increase the probability of selecting crosslinked peptide segments, mother ions with <three positive charges were excluded from secondary mass spectrometry. Raw mass spectrometry data were analyzed using pLink 2.3.9 software[58]. Specifically, the photoactivatable unnatural amino acid Bpa was set as code B in the FASTA file database, with a molecular formula of $C_{16}H_{13}NO_2$. It was assumed that Bpa could crosslink with any of the 20 natural amino acids while maintaining its original molecular weight.

## Functional complementation analyses of FtsZ variants

This was performed using the *ftsZ*-knockout strain Δ*ftsZ*. This strain carried the pJSB100 rescue plasmid[21,39], resulting in cell survival upon wild-type FtsZ induction with 0.05% arabinose, but cell death in the presence of 0.2% glucose (without arabinose). FtsZ variants were tested in this strain (under the latter conditions) by constitutive expression from the pTac plasmid; pTac plasmids expressing and not-expressing wild-type FtsZ were used as positive and negative controls, respectively.

Briefly, cells were cultured overnight at 37 °C in LB medium plus 0.05% arabinose. Then, 5 μL of $10^{-1}$–$10^{-6}$-fold diluted cultured cells were spread onto plates containing either 0.05% arabinose (positive control group) or 0.2% glucose (test group). Any FtsZ variant that allowed Δ*ftsZ* cells to grow in a comparable manner in the presence of glucose or arabinose was considered fully functional. Of note, the LY928 strain used in photocrosslinking analysis was also used here (solely for the purpose of consistency, though not necessary).

## Fluorescence microscopy

To visualize entities assembled by FtsZ$^{WT}$, FtsZ$^{\Delta NIDR}$, FtsZ$^{D10F}$, FtsZ$^{\Delta NIDR-P203S}$ or FtsZ$^{P203S}$, variants were fused to a fluorescent protein, which was expressed from the genomic *rha* operon upon rhamnose induction. For co-localization analyses, FtsA was constitutively expressed, and FtsN was expressed from a plasmid in a rhamnose-induced manner in cells. Cells were cultured overnight and re-cultured in fresh LB medium for an additional 2 h before microscopy. To label newly synthesized peptidoglycans, re-cultured cells were mixed with the fluorescent D-Ala analog HADA (Aladdin, H287563-5 mg) (by brief vortexing) at a final concentration of 80 μM, according to a previous protocol[59].

For live-cell imaging examining the effects of overexpressed MinC on FtsZ variant assembly, *ftsZ-WT-mneongreen*, *ftsZ-D10F-mneongreen*, and *ftsZ-ΔNIDR-mneongreen* cells were cultured overnight in the presence of rhamnose (0.5%). Cells were then re-cultured in fresh medium for an additional 2 h before MinC (from a pYRG plasmid) was induced for 2 h by adding anhydrotetracycline (0.2 μg/mL) prior to imaging. Cells were placed on a glass dish (NEST Biotechnology, USA) and covered with an LB agarose pad before microscopy.

Traditional wide-field fluorescence microscopy was performed using an EVOS™ FL Auto 2 imaging system (Thermo Fisher Scientific, USA) with a 100×/1.4 NA oil-immersion objective lens (Olympus, Japan). Images were processed using Fiji 1.48 software.

Super-resolution structure illuminated microscopy (SIM) imaging was performed using an N-SIM imaging system (Nikon, Japan) in 3D-SIM mode, with a 100×/1.49 NA oil-immersion objective lens (Nikon) and excited by 405, 488, or 561 nm laser beams. Samples were sectioned at 120 nm intervals along the Z-axis and images reconstructed using NIS-Elements AR 5.42.00 (Nikon).

Timelapse live-cell imaging was performed using a CSU W1 rotary confocal microscope (Nikon), with a live SR structured lighting super-resolution module and a 100×/1.4 NA oil-immersion objective lens (Nikon). Three-dimensional images were acquired at 100 nm intervals along the Z-axis and images captured using a sCMOS Prime 95b camera and reconstructed using MetaMorph software. All images were reconstructed using NIS-Elements AR 5.42.06 and were processed using Fiji 1.48 software.

Brightfield microscope images were taken before fluorescence imaging to record cell shapes.

## Scanning electron microscopy

Cells expressing specific FtsZ variants were cultured overnight and re-cultured for 2 h in fresh LB medium before 5 mL of the culture medium was applied to a coverslip that was wrapped in mirror cleaning paper. Coverslips were then incubated with fixative solution (2.5% glutaraldehyde) for 1 h at room temperature and transferred to 4 °C for an overnight incubation. Cells were then washed in 1 M phosphate-buffered saline, dehydrated in an ethyl alcohol gradient, and dried using a critical point dryer (Leica, Germany) for 2.5 h. Coverslips were then attached to a plastic sheet and covered with gold before visualization and photography using a FEI Quanta FEG 450 microscope (FEI, USA) at ×30,000 magnification.

## Fluorescence-recovery after photobleaching (FRAP)

FRAP was performed using a CSU W1 rotary confocal microscope (Nikon) with a live SR structured lighting super-resolution module and a 100×/1.4 NA oil-immersion objective lens (Nikon). We performed FRAP experiments using cells cultured for 6.5 h, the earliest time point at which obvious fluorescence and significant phenotypes were observed, also a time point where most wild-type cells formed well-maintained Z-ring architectures. Cell samples were photographed three times before the subcellular zone (to be bleached) was irradiated with a high-intensity pulsed laser, and images acquired at 2 s intervals for 2 min. Recorded images were processed using the FRAP Profiler v2 plugin (Fiji 1.48 software) to monitor fluorescence intensity recovery in photobleached areas. To measure relative fluorescence intensity in bleached areas during recovery stages, the mean intensity of bleached areas was set to 100% before bleaching, and 0% immediately after bleaching. Converted fluorescence intensity data from multiple bleached areas (in different individual cells) were analyzed in Graph-Pad Prism 10.1 software to generate a time-dependent intensity recovery curve. Variations in fluorescence intensity across photobleached areas at each time point were modeled using one-phase decay nonlinear regression to calculate the time taken to reach half-maximum fluorescence recovery.

## SIM-TIRF microscopy and kymograph analysis

In these assays, overnight-cultured cells were re-cultured on fresh LB agarose pads for 1 h, after which, the movement of a selected FtsZ entity in a particular cell was recorded. Specifically, a small FtsZ condensate was tracked using a CSU W1 rotary confocal microscope (Nikon) containing a live SR structured lighting super-resolution module, using a 488 nm laser and a 100×/1.49 NA total internal reflection fluorescence (TIRF) oil-immersion objective lens (Nikon). During acquisition, images were recorded at 2 s intervals. The Image Stabilizer plugin in Fiji 1.48 software was used for initial image alignments. Three-pixel lines were drawn over a small condensate trace, and kymographs were plotted using the KymographBuilder plugin. Condensate treadmill velocity was determined using the slope of manually drawn lines along motion lines in kymographs.

## Light scattering analysis

FtsZ polymerization under in vitro conditions was monitored using 90° angle light scattering on a Hitachi fluorometer (model F-4500, Japan) with excitation and emission wavelengths set to 350 nm and the slit width set to 1.5 nm, as previously reported[60]. Although initial experiments were conducted with FtsZ at a physiologically relevant concentration of 5 μM, we ultimately adopted a 15 μM concentration to ensure experimental reproducibility and operational feasibility in achieving our specific research objectives. It followed that the purified form of particular *E. coli* FtsZ (EcFtsZ)/BsFtsZ/SaFtsZ was added at a final concentration of 15 μM in respective polymerization buffers, to a fluorometer cuvette with a 1 cm path length. The cuvette was then transferred to the fluorometer, and light scattering readings were recorded three times and used as a baseline. GTP was then added at a concentration of 1 mM, and the cuvette placed into a 30°C incubator before data were collected at 5 min intervals.

## Ultracentrifugation sedimentation analysis

For EcFtsZ, the purified protein was diluted to the indicated concentrations in polymerization buffer in a final reaction volume of 400 μL. After GTP addition at a final concentration of 2 mM, and the sample incubated at 30 °C for 5 min either with MinC (final concentration the same as FtsZ) or without MinC, 300 μL of each reaction mixture was centrifuged at 450,000 × *g* in a TLA120.1 rotor at 4 °C for 15 min in an Optima MAX-XP ultracentrifuge (Beckman, USA). After supernatant removal, the pellet was resuspended in 300 μL polymerization buffer, and the total mixture, supernatant, and pellet were resolved by SDS-PAGE and stained with Coomassie bright blue stain. Protein band intensity was measured with the Fiji 1.48 software.

For BsFtsZ/SaFtsZ proteins, polymer pelleting assays were performed as previously reported[57]. Briefly, polymerization was started with 10 μM purified proteins with or without the addition of 2 mM GTP in 300 μL polymerization buffer and incubated at 25 °C for 15 min. Then, 200 μL each reaction mixture was centrifuged at 450,000 × *g* in a TLA120.1 rotor at 25 °C for 20 min. The total mixture, supernatant, and pellet (resuspended in 200 μL polymerization buffer) were resolved by SDS-PAGE and stained with Coomassie bright blue stain. Protein band intensity was measured with the Fiji 1.48 software.

## GTP hydrolysis assays

GTP hydrolysis was determined by measuring released Pi concentrations using malachite green phosphate assays, as previously reported[61]. Specifically, 50 μL of the reaction mixture was added to an 800 μL solution comprising three volumes of 0.045% malachite green and one volume of 4.2% ammonium molybdate solution. Then, a 100 μL solution of 34% (w/v) sodium citrate was added and tubes incubated on ice for 20 min before absorbance (630 nm) was measured. Relative GTPase activity was calculated using measured Pi concentrations at indicated time after GTP addition to the reaction solution.

## Negative stain electron microscopy

Transmission electron microscopy images of FtsZ protofilaments formed under in vitro conditions were recorded as previously reported[27]. Briefly, carbon-coated copper grids (200 mesh, Electron Microscopy Sciences, China) were glow-discharged for 30 s before use. Each FtsZ variant was diluted to a final concentration of 1 μM in polymerization buffer. Then, 2 mM GTP was added to initiate polymerization. The reaction mixture was incubated at 30 °C for 5 min, after which 10 μL was applied to carbon-coated copper grids and incubated for 1 min before excess liquid was blotted. Grids were immediately rinsed in one drop of ddH$_2$O and three drops of 2% uranyl acetate, blotted, and air-dried. Images were recorded using a JEM-F200 electron microscope (JEOL, Japan) at × 50,000 magnification.

## Molecular dynamics simulations

Five independent 150 ns molecular dynamics (MD) simulations were conducted using the GROMACS 2024.2 package[62].

The *E. coli* FtsZ dimer structure (residues 11–315) was modeled from SaFtsZ (PDB 3VOA)[63] by using the SWISS-MODEL server[64]. The N-IDR (residues 1–10) of FtsZ was constructed as an extended conformation by using the Pymol software[65], minimizing its interaction with the rest of the dimer. The GDP molecules coordinates were replaced with that of GTP. Topology and force field parameters for the GTP molecule were generated using the ACPYPE tool[66]. The FtsZ dimer was positioned at the center of a rectangular periodic boundary box filled with water molecules. Sodium ions were added to neutralize negative charges in the system. Simulation system setup is summarized (Supplementary Table 6). FtsZ protein and sodium ions were modeled using the AMBER ff99SB-ILDN force field[67], whereas GTP parameters were derived from the standard AMBER force field[68], and water molecules were defined using the SPC/E model[69]. All residues maintained their canonical protonation states at pH 7.0, as parameterized in the AMBER ff99SB-ILDN force field. No explicit protonation state adjustments were required for the simulated system.

The simulation methodology followed conventional practices for all-atom MD of protein-nucleotide systems. AMBER ff99SB-ILDN force field and AMBER parameters were adopted for their balance of accuracy and computational efficiency in modeling folded and disordered protein regions. The SPC/E water model was selected for its compatibility with the AMBER force field and its realistic dielectric properties. While model limitations (e.g., fixed-charge approximations) are acknowledged, this setup enabled the probing of ms-scale conformational changes that were inaccessible to polarizable models. These choices provided sufficient accuracy to resolve residue-level interactions and conformational changes.

The system was energy-minimized using the steepest descent algorithm (50,000 steps) with a convergence threshold of 1000 kJ mol$^{-1}$ nm$^{-1}$ for the maximum force. A 0.1 ns equilibration at 300 K was performed under the NVT ensemble with positional restraints applied to the protein. Temperature coupling was achieved using the V-rescale thermostat algorithm. The system was further equilibrated for 0.1 ns under the NPT ensemble (1 bar, 300 K) using the Parrinello-Rahman barostat method, with positional restraints maintained. The system underwent 150 ns of unrestrained MD simulation under constant temperature and pressure (NPT) conditions, with a 2 fs integration interval. Hydrogen bonds were constrained using the LINCS algorithm[70], and non-bonded interactions were computed using a 1.0 nm cutoff for both Coulombic and van der Waals terms, with long-range electrostatics handled using the Particle-Mesh Ewald method[71]. Trajectories were saved every 10 ps for analysis. Verlet cutoff scheme and periodic boundary conditions were applied throughout. Randomly assigned velocities were initialized from a Maxwell– Boltzmann distribution at 300 K.

The mean smallest distance between every pair of residues within the FtsZ dimer during the last 50 ns simulation was calculated using

the Gromacs mdstat tool with a cutoff of 1.5 nm and visualized as a heat map. The average structure of the FtsZ dimer during the last 50 ns of each simulation was generated using the Gromacs rmsf tool, and the visualization was performed using the PyMol software (2.5). The mean smallest distance between the N-IDR (residues 1–10), the T7 loop (residues 198–211) of one FtsZ monomer, and the N-domain (residues 42–54) of the other during the last 50 ns was calculated with a cutoff of 0.6 nm to reveal potential contacts. RMSD, RMSF, and Rg values of the N-IDR, T7 loop, and N-domain within the dimer interface were calculated from 150 ns trajectories using GROMACS tools (rms for RMSD, rmsf for RMSF, and gyrate for Rg). The resulting data were visualized using Matplotlib (v3.6.3).

### Statistical and reproducibility

Unless otherwise mentioned, all experiments were repeated independently at least three times. The numbers of experimental repeats are indicated in the respective figure legends. Statistical analyses were performed in GraphPad Prism 10.1 software, and probability values ($P$ value) and standard error of the mean (SEM) were calculated using parametric unpaired two-tailed Student's $t$-tests; $P$ values <0.05 were considered statistically significant. Biological replicates are indicated in the figure legends.

### Reporting summary

Further information on research design is available in the Nature Portfolio Reporting Summary linked to this article.

## Data availability

Simulation datasets generated in this study, including initial coordinates, input parameter files, and trajectory snapshots saved at 100 ps intervals (full trajectories with 10 ps resolution are available upon request), along with time-averaged coordinate files derived from the conformational ensemble sampled during the final 50 ns, have been deposited in the Zenodo repository (accession code: 15186509; https://zenodo.org/records/15186509). Mass spectrometry data generated in this study have been deposited in the ProteomeXchange Consortium through the PRIDE partner repository with dataset identifier PXD063736. Source data are provided with this paper.

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

## Acknowledgements

We thank Prof. Peter Schultz (The Scripps Research Institute, USA) for providing plasmids that express genes encoding the orthogonal tRNA and orthogonal aminoacyl-tRNA for Bpa incorporation. We thank the staff at the Core Facilities at the School of Life Sciences, Peking University, especially Dr. Chunyan Shan and Dr. Jun Ren, for assistance with the Structure Illuminated Microscopy imaging system and Live SR CSU W1 microscope. We are grateful to Dr. Yingchun Hu and Dr. Yiqun Liu for assistance with sample preparation and electron microscopy. We thank Dr. Wen Zhou at the Mass Spectrometry Facility of the National Center for Protein Sciences at Peking University for assistance with mass spectrometry. This research was supported by funds from the National Natural Science Foundation of China (32150003, 3215000010, 31971189, 31670775, and 31470766), the National Basic Research Program of China (2012cB917300), and the Qidong-SLS Innovation Fund.

## Author contributions

Investigation, H.Y., Y.L., and Y.Z.; methodology, H.Y., Y.L., Y.Z., and P.C.; writing, Z.C., H.Y., Y.L., and Y.Z.; Funding acquisition and supervision, Z.C.

## Competing interests

The authors declare no competing interests.
