## [Transparent Peer Review file · Nature Communications]

A FtsZ cis disassembly element acts in Z-ring assembly during bacterial cell division

Corresponding Author: Professor Zengyi Chang

Version 0:

Reviewer comments:

Reviewer #1

(Remarks to the Author)

The research article by Yin et al presents an exhaustive screen of all possible residues of FtsZ that may interact with other proteins, including other FtsZ subunits. The screen itself may be of significant general interest to the field, although it does suffer from some consistency issues. Nonetheless, through this screen, the authors uncovered a previously unappreciated role of the extreme N-terminus of FtsZ (NIDR) in regulating FtsZ polymer dynamics. From the various experiments presented, it appears that the FtsZ NIDR fine-tunes FtsZ polymers by reducing subunit affinity. In a Δ NIDR mutant, FtsZ polymers are stabilized, but this defect can be rescued by introducing the P203S mutation (which presumably destabilizes polymers enough to compensate for the loss of the NIDR?). Overall, the manuscript presents convincing data for a novel role of the NIDR, although the exact mechanism by which the NIDR affects polymerization is not entirely resolved.

The authors favor a model in which longitudinal interactions are directly impacted by the FtsZ NIDR, however the experiments presented do not adequately exclude the possibility of an effect on lateral interactions (and indeed, Fig. 4D appears to indicate such a possibility). Further characterization of the P203S mutant alone may help lend credence to the author's preferred mechanism. The manuscript also did not explore whether this mechanism might exist in other bacteria with other FtsZ proteins, which might also improve the general interest in these findings.

However, in general the authors need to take greater care when interpreting and presenting their data and experiments. There were critical experimental details left undefined, including the exact definition of the NIDR residues or the Δ NIDR mutant. In some cases, it was also unclear what strains were being used in each experiment, with some details available or only implied by the strain and plasmid list presented in the supplemental materials. Additionally, the authors need to exert greater caution in interpreting the presented data to avoid speculating further than the data supports. Overall these issues would be correctable with more careful and complete editing.

Comments

1. "Architecturing" is not a proper English word. Please replace all instances of "architecturing" (including in the Title) with "assembling" or "organizing"
2. Many of the figures include text that is too small to be legibly printed, even by laser printer. Fig 1a and 5C in particular are too small, and the diagram depicting longitudinal self-assembly to the right of panel 1G is difficult to see.
3. Lines 36-38: Need to clarify the scope of this discussion as many bacteria do not have oscillating Min systems (or any Min systems at all).
4. Lines 41-42: This mutant requires greater context as the current writing suggests that GTPase activity is not very important for FtsZ function. The mutant in question is ftsZ84 (FtsZ G105S), which exhibits 10% of the GTPase activity of WT FtsZ. In the cited Sticker 2002 paper, this decreased activity carries distinct changes in FtsZ treadmilling behavior (increased midcell accumulation and slower recovery in FRAP experiments) consistent with significantly decreased GTPase activity.
5. Line 48: What is the evidence for the N-IDR? It is probably true that the extreme N-terminus (residues 1-11 or so) is disordered, but this would be a common trait of the extreme N terminus (or C terminus) of many proteins. What residues do the authors define as the N-IDR? This needs to be couched in background and context instead of defining it without any introduction.
6. The N terminus of FtsZ was previously implicated as an important domain for self-interaction in ref. 23. The highly conserved Ala 11 was previously identified as important for self-interaction (A11 at the bottom of one subunit interacts with

L68, in the N-terminal domain at the top of the next subunit). This is adjacent to the residues 5 through 10 that have crosslinked bands in Fig. S1. Do the authors define the N-IDR only from residue 1 to residue 10?

7. Curiously, A11-Bpa itself does not crosslink, perhaps because A11 is crucial for longitudinal interactions and the relatively bulky Bpa replacing Ala at this position sterically interferes with this. Some of the other interacting residues in the crystal structure do not crosslink with Bpa in the present study, perhaps for the same reason. Could the authors comment on this?

8. Fig. S1A: The figure legend states that blue residues were found before from crystal structures or previous data by this group. This is insufficient information. Which blue residues were implicated by crystal structures (and which crystal structures) and which were previously identified by this group, and in what publication?

9. FtsZ residues that this group has found before (presumably ref. 18) to crosslink such as R78, N79, D82, R85, R89, K155, S231 were claimed to be lateral interactions. Nonetheless, they did not show up in Fig. S1, which theoretically should detect any interaction, longitudinal or lateral; the genetic makeup of the strains seems to be identical. Is it because lateral interactions are much weaker under these conditions compared with longitudinal interactions and thus would be below detection limits?

10. Fig S1A indicates numerous residues that are labeled in red as self-interaction residues. Some of these residues (eg, 5,6,7,9) fail to show confirmatory crosslinking with FtsZ-avi on either the anti-FtsZ blots (as double bands) or the streptavidin blots (as single bands) shown in Fig 1C or Fig S1B. Additionally, there are some residues (such as 58 and 59) that are not indicated on Fig S1A but nonetheless appear to form crosslinks on the confirmatory blots of Fig S1B. What is different about the experiments in Fig S1A versus Fig 1C / Fig S1B that would explain the loss or appearance of apparent crosslinking bands? Can these be called self-interaction residues if they cannot be verified?

Lacking sufficient supporting confirmatory evidence, the authors need to walk back claims of FtsZ self assembly roles for some of these residues (e.g., Fig S1A, lines 66-70). It would be helpful if the authors provided a table or a list to summarize results across the various blots for (for example) which residues formed any crosslinks versus those that formed verified crosslinks with other FtsZ proteins or other proteins, or that were simply untested (as for the 325-329 residues).

11. In Fig S1A, there are bands visible in some lanes that are neither labeled as novel nor previously identified. Residue 31, for instance, which is revealed by another blot to be an interaction with a different protein. Additionally, residues, 86, 90, 95, 103, 142, 237, 239, 323, 330, 343, 344, 355, 357 (and others?) also appear to form bands of some sort. Even if the interaction partners are not identified, these putative crosslinking sites may be useful findings for other researchers interested in FtsZ protein interactions and should also be summarized, blot reliability notwithstanding.

12. Fig 1D and Table S1 indicates that Bpa at T8 and D10 crosslinks residue V42 with the highest or second-highest abundance. However, V42 is fairly deeply buried inside FtsZ, so it is not clear how this interaction would occur.

13. Lines 88-93 and Fig. 1G: here, A87 seems to exhibit lateral interactions (but not the other residues listed in point 5). The evidence is that there is no ladder like there is with T8 and K51. However, the crosslinking for A87 is much weaker than that for T8 and K51, so perhaps a ladder cannot be detected at this exposure. Moreover, these results do not clearly demonstrate that lateral dimers can not also be formed. Lines 88-93 should be re-written to reflect this. Bear in mind that the mass spec results (Fig 1/Table S1) show N-IDR interacting with residues Q56, F40, S62 that are not at the longitudinal interface, but are in close proximity to A87 at the lateral interface. It remains plausible that the N-IDR could also play a role in lateral interactions as well.

14. Ref. 18 is cited for A87-Bpa mediating lateral interactions, but there does not seem to be a mention of A87 in Ref. 18.

15. Lines 89 and 94 make reference to the T8 (N-IDR) and K51 'mediating' longitudinal self-assembly. At this point, there is not sufficient evidence to state that T8 or K51 are required for (or are mediating) longitudinal interactions. Given as the crosslinking interactions occur despite the Bpa substitution at these residues, they are unlikely to directly play an essential role in the interaction as well. Indeed, Fig 2 provides ample evidence that FtsZ can assemble into filaments without the N-IDR.

16. Lines 94-97: It is true that there was no previous report of extensive interactions between the N-IDR (assuming that it is an IDR, see comment #1) and the rest of the N-terminus in refs. 23-24, but as mentioned in comment #5 above, there was a clear mention of interaction between A11 and L68 in ref. 23 as being involved in longitudinal interactions. It therefore depends on how the authors define the N-IDR.

17. The deletion of N-IDR needs to be defined. Is it the deletion of residues 1-10?

18. Lines 102-104 and Fig. 2A: the genetic constructs need to be described more fully in the text and figure legend, not only in the Methods; even in the Methods the strains are not well described. The reader should not have to look up a previous paper to get basic information about a strain used extensively in the current study (also in Fig. 5E). If the cultures in Fig. 2A (and 5E) have an *ftsZ* null allele in the chromosome and pJSB100-FtsZ and pJSB100-FtsZ delta N-IDR plasmids under arabinose control, then why does repressing the plasmid promoter with glucose result in viability for the strain carrying WT FtsZ on the plasmid? The data in 5E imply that WT FtsZ is under arabinose control; so then what controls expression of the FtsZ variants?

19. Also, the strain with the deletion of FtsZ is still partially viable on the plates with arabinose. This contradicts the statement that this variant "failed to support cell growth"

20. In the FtsZ sandwich fusion that is viable, mNeonGreen is inserted between residues 55 and 56 of FtsZ. How could this region be important for FtsZ assembly if it is disrupted by the large mNeonGreen moiety?

21. Line 114: How is FtsZ constitutively expressed in this strain?

22. Lines 125-129: This paragraph is very unclear as written.

23. Line 131-134: This sentence implies that multiple positions within the FtsZ N-IDR were subjected to this mutational analysis ("by replacing residues in the FtsZ N-IDR"), however only mutations for D10 are discussed. Were any residues besides D10 mutated, or was only D10 mutated? This should be clarified.

24. Fig 2 B-G: FtsZ-mNeonGreen (B/C/G) or FtsZ-mScarlet (E/F) appear to be expressed in trans with the native FtsZ. This indicates that the negative effects observed with the mutants must therefore be dominant over native FtsZ. Additionally, native and mutant FtsZ subunits may co-polymerize, so native subunits may be responsible for the recruitment activity observed. This limits the ability to interpret these experiments.

25. Lines 156-161: It is interesting that the aberrant FtsZ structures can recruit FtsA and FtsN, but this has been reported before. For example, aberrant non-ring FtsZ structures are known to recruit FtsA and ZipA to the same structures (PMID: 10601211) and short non-ring polymers of FtsZ can drive cell constriction events (PMID: 28438890). Additionally, the FtsN co-localization is limited to regions where FtsZ is forming a midcell band in Fig 2F, as co-localization is not observed with FtsZ bands formed elsewhere. The cell wall biosynthesis at these structures is also not particularly convincing, as the HADA/FtsZ patterns are mostly not overlapping in the images shown. It would also be much easier to see the HADA fluorescence in Fig 2G if the dark blue pseudocolor were changed to a higher contrast color.

26. What is more surprising is that FtsZ lacking the N-IDR can assemble into polymeric structures. It is not clear how this occurs if the N-IDR is required for FtsZ longitudinal subunit interactions. It looks more like the N-IDR facilitates lateral interactions between FtsZ protofilaments, where lack of these lateral interactions would result in polymers that could not coalesce into a coherent Z ring. This is essentially stated in lines 163-164.

27. Fig. 3A: It is puzzling why photobleaching recovery time for the huge ribbons made by FtsZ D10F is not that much lower than that of WT FtsZ in rings. It is hard to imagine how such ribbons remain that dynamic.

28. Line 173: which FtsZ "entities" were chosen for velocity measurements, and which were not? From the micrographs, it looks like most of the FtsZ structures remained static over the time course.

29. Lines 173-175: There is a notable difference between the FtsZ velocities reported in the text and those represented in Fig 3D. The text gives 39.09, 22.9, and 9.43 nm/s for WT, D10F, and Δ NIDR. The cross bars in the figure appear to represent values of about 42, 28, and 14 nm/s, respectively.

30. Line 184 and following: The conclusion that MinC overproduction perturbs FtsZ structures less efficiently in the D10F and delta N-IDR variants is not that convincing from the images in Fig. 3F. Perhaps showing the fluorescence channel only, without the DIC overlay, would help.

31. Lines 187-189: the conclusion that the residual photocrosslinked FtsZ dimer levels were significantly higher for the delta N-IDR variant of FtsZ after MinC overproduction is not that convincing. Yes, the band is somewhat stronger than for WT FtsZ or FtsZ D10F, but still much weaker than the uninduced controls, and higher order polymers were not detectable. Was this result reproducible? Were cells harvested for imaging in Fig. 3F also used for the crosslinking? There is no mention of how the cells were harvested for the crosslinking.

32. Perhaps expressing MinC at lower levels (or shorter times) would provide a better contrast between crosslinked dimer band intensities. In any case, the data with purified FtsZ and MinC shown in Fig. 4 are more convincing.

33. In Fig. 3G, why is there only 1 crosslinked dimer band (lane 10) vs the doublet bands in lanes 2 and 6?

34. Speaking of MinC overproduction, the Methods state that minC was induced by AHT overnight and then 2 additional hours after dilution of the culture. It would probably be more physiologically relevant to induce MinC only in the exponentially growing culture for a short time prior to imaging.

35. FtsZ-pdt is shown in Fig. 3G but it is not explained. Later in the manuscript it is mentioned that this version of FtsZ is degraded by mf-Lon proteinase with a reference, but this description is insufficient and should be explained in the text and figure legend.

36. Fig. 4, particularly panel D, shows clear evidence for increased FtsZ filament bundling of the delta N-IDR variant compared with WT or the D10F variant. And yet most of the rest of the paper argues that the N-IDR affects longitudinal interactions, not lateral interactions. This is puzzling. How might a polymer disassembly defect result in polymer bundling?

37. Lines 208-209: Comparing Fig 4D and F, it appears that expression of MinC reduced the amount of FtsZ in the pellets of both the WT and D10F mutant by half, whereas the Δ NIDR mutant is essentially unaffected. This suggests the D10F isn't necessarily less sensitive to MinC expression (compared to WT) so much as it instead starts at a higher degree of polymerization.

38. Fig 5B: The figure legend indicates that this model is based on the 6UNX pdb structure. However, the 6UNX structure is E. coli FtsZ crystalized in the R (relaxed) conformation in which K51 and P203 are separated by ~ 17 Å, whereas this model appears to represent FtsZ in the T (tense) conformation. To my knowledge, there is no crystal structure solved with E. coli FtsZ in the tense conformation, and this can only be approximated by threading the E. coli FtsZ sequence through one of the S. aureus crystal structures in the tense conformation (eg, 3VOA). The authors need to clarify the origin of this protein structure.

39. Lines 222-225: It would be helpful to explain the purpose of the pdf tag fusion & expression of the Lon protease in Fig 5C.

40. The P203S variant and its ability to restore the function of the delta N-IDR variant are very interesting and nicely support the hypothesis that the N-IDR is involved in disassembling FtsZ polymers. One prediction of the authors' model is that the delta N-IDR P203S double variant should now have WT or nearly WT levels of GTPase activity. Was this tested?

41. Another prediction of the model is that FtsZ P203S alone (with the N-IDR intact) would be defective in assembly, or perhaps could disassemble better than WT FtsZ. Was this attempted? This would help to shed light on how P203S actually affects FtsZ assembly.

42. Lines 249-253 / Fig 5I: The use of "non-interfering state" or "interfering state" to refer to the N-IDR in FtsZ monomers or polymers (respectively) needlessly complicates the explanation. Likewise, rephrasing "Event 1", "Event 2", etc with "polymerization", "NIDR binding", etc would also make this more clear.

43. Lines 263-272: This discussion would benefit from further consideration of the orientations of the tense and relaxed forms of FtsZ. For instance, most likely the K51 Bpa crosslinking with P203 occurs with the (GTP-bound) tense conformation of FtsZ Δ NIDR, as these residues are too far apart in the relaxed conformation to crosslink. Given as P203S restores the defect in FtsZ Δ NIDR, likely the NIDR is functionally active with tense form of FtsZ. It also seems unlikely that any conformational change as suggested in lines 267-270 (eg, to relaxed form of FtsZ with GTP hydrolysis) would improve FtsZ NIDR interactions, because the FtsZ NIDR will be more distant from residues it interacts with (particularly Q56 & F40) when filaments are in the relaxed conformation.

44. Lines 278-281: This is an interesting idea, and there is prior evidence for the G105S (ftsZ84) allele having a disassembly defect (PMID 11029443). However, I am pretty sure that FtsZ84 protein has not been shown to assemble in vitro (at any temperature), so this point may not be valid for that reason.

45. Line 284: Earlier comments about FtsN recruitment and cell wall biogenesis also apply here.

46. Lines 285-287: this sentence is confusing; "far earlier before" does not make sense. The sentences following this also do not make sense.

47. Lines 282-291: there is already considerable evidence for certain proteins, including FtsZ, as limiting factors for division. Too little FtsZ: no division. Too much: minicells. FtsZ:FtsA ratios are important too. So, it is inaccurate to say that division relying on quantitative measures is a new idea; same with FtsZ disassembly as being important (FtsZ84 with its low GTPase activity has defects in disassembly, similar to the Δ N-IDR mutants). The main novelty of the present work is that it identifies an interesting function of the extreme N terminus of FtsZ that was not previously appreciated, and that this function may be to compete with longitudinal interactions in order to tip the balance more towards polymer disassembly. I suggest removing this paragraph and replacing it with a brief summary of the main findings of the paper without the needless speculation.

48. Are there conserved features in the FtsZ NIDR that might suggest this mechanism plays a role in other FtsZ proteins? A greater exploration of this across diverse bacterial species would increase the relevance and impact of these findings beyond just the *E. coli* cell division field.

Minor comments

1. Line 14: "has to be both steady and constantly changing"

Although the meaning will be clear to experts in the field, authors should more explicitly explain the meaning for the sake of other readers

2. Lines 18-21: "a previously unobserved ..."

The text following the semicolon is a sentence fragment that lacks a subject; it needs to be rewritten.

3. Line 27: Sentence missing a period & has unnecessary comma.

4. Line 87: delete "a"

5. Line 275: replace "resumption" with "rescue" or "restoration"

6. Figure 3F: A different label for the lower row of images would be helpful. The sideways '-' looks like an 'l' here and can be confusing.

7. Fig. 4H: y axis "proportion" is misspelled

Reviewer #2

(Remarks to the Author)

This study explores the formation of the bacterial Z-ring through the self-assembly of the FtsZ protein in *Escherichia coli*. Using protein photocrosslinking, mass spectrometry, and high-resolution microscopy, the researchers identified the role of specific residues of the N-terminal intrinsically disordered region (N-IDR) in FtsZ longitudinal contacts in *E. coli*. Disrupting the N-IDR impairs cell division, underscoring its importance in Z-ring formation. IDR regions have been classically ignored because they cannot be crystallized, so this study follows a necessary new trend of further characterize their unexplored biological functions. Thus, this research article would be interesting in the field of bacterial cell division, providing significant advancements in our knowledge on the FtsZ ring formation. That said, there are some issues that must be addressed before publication:

- In general, the text needs strong edition. While the experiments were explained in a logical order, and the story could be followed, some parts of the work were difficult to understand.

For instance, which specific residues were eliminated in FtsZ Δ NIDR variant?

What is the difference between Supplementary Figures 1A and 1B? Is just a repetition restricting the experiment to a set of positions, or is experimentally different? Authors could include the explanation.

- Bpa photocrosslinking has been shown to underperform when applied to disordered regions (Zhang et al., 2011, *Nat Chem Biol*). Could the authors comment on the motivation of the election of Bpa instead of other available options, for instance, DiZPK, also developed by the authors?

- To support the conclusions of in vivo photocrosslinking (specific amino acid interactions) the authors should validate these interactions with an independent approach, compatible with intrinsically disordered domains, like HDX, NMR, BLI, SPR... Additionally, Molecular Dynamics simulations could help to understand the movements and contacts of the N-IDR, and the data obtained can be used to plot frequency contact maps.

- Was the experiment in Fig.1G performed with the other Bpa positions indicated to participate in longitudinal interactions: residues 10, 20, 24, 56, etc? In my opinion, this experiment was very clever, and it would be nice to see if all those amino acids behave in the same manner as 8 and 51.

Line 94 - The authors claim that "We strongly emphasize that longitudinal self-assembly interactions mediated by the N-IDR and N-domain were revealed here for the first time, and were almost completely missed in previous studies (21–24), evidently due to N-IDR invisibility in resolved crystal structures."

However, they are missing some previous studies that support the role of the N-IDR in protofilament formation: Corrales-Guerrero et al., 2018 (*Front Microbiol*) already showed that the deletion of N-IDR in multicellular cyanobacteria leads to straight filaments that are more stable by ultracentrifugation and present a reduced GTPase activity. Also, Silber et al., 2020 (*mBio*) showed that the N-IDR is more protected from ClpP degradation when *Bacillus subtilis* FtsZ is forming a protofilament, and proposed that the reason might be the interaction within subunits. Fujita et al, 2023 (*Nat Comms*) resolved *Klebsiella pneumoniae* FtsZ by cryo-EM, showing that the N-IDR region participates both in longitudinal and lateral interactions. The authors should change the mentions to "the first time" or "previously unobserved" across the text.

- Related with the previous comment: To broaden the conclusions, the authors could test if the hypothesis of the competing

interactions is applicable to other bacteria.

Line 131 - In the text is written: "To this end, we adopted a screening approach by replacing residues in the FtsZ N-IDR with each of the other 19 natural amino acids", but only mutations from D10 are shown in Supplementary Figure 4. Were the mutations in other positions not producing cell growth defects? The differences in phenotype between dNIDR and D10F shown in Fig. 2D would point to other residues having a role too. I don't think it is necessary to make replacements by every single amino acid, but if it is done already, it would be helpful to show it and discuss it in the text.

Line 208 - The authors claim that "the polymers formed by FtsZ variants exhibited more resistance to MinC protein depolymerization effects". If I understood correctly, they expect that this is an indirect effect derived from the fact that the polymers are stabilized in the variants FtsZ Δ NIDR and FtsZD10F, and MinC binds mainly to Helix 8 and 10, which are only available in the last subunit of the polymer. However, Machado et al., 2024 (biorxiv) NMR experiments on *Bacillus subtilis* FtsZ show that, apart from residues already described in the C-terminal region, FtsZ residues L11, A12 and A26 presented significant differences in the chemical shifts (Figures 3C,E from the preprint) in the presence of MinC. For that reason, the results shown in Figures 3F,G and 4G,H of this work could be explained by both direct and indirect facts, or the addition of both. The authors must clarify if the N-IDR region interacts with MinC in their conditions in order to properly explain the results.

- The authors should demonstrate that the interaction between S203 and K51 is debilitated again in FtsZ Δ NIDR P203S variant in comparison to FtsZ Δ NIDR.
- Figure 5G - What is the effect (by ultracentrifugation) of P203S mutation in the wild type background? The authors need to show that the effect of the mutation is directly related to the interaction with K51 and not only with being in the polymerization surface. The combination of D10F P203S would also shed light into the role of the N-IDR.
- Figure 5I - If the hypothesis is correct, titration of a peptide containing N-IDR to full-length FtsZ (and to FtsZ Δ NIDR) could destabilize filaments, and promote disassembly. This could be assayed by RALS or sedimentation assays. This is only a suggestion, but could reinforce the model.
- It has been demonstrated that FtsZ suffers a conformational change, which is related to the monomeric/polymeric states (Matsui et al., 2014, J Biol Chem; Wagstaff et al., 2017, MBio). This conformational change affects the N-terminal domain, so it could also affect N-IDR movements and contacts. In fact, Fujita et al, 2023 (Nat Comms, already mentioned) analyzed FtsZ conformational changes by cryo-EM, and observed the N-IDR region ordered as an alpha-helix in one of the conformations, participating both in longitudinal and lateral interactions. The authors could comment on that in the discussion.

Minor:

- Fig.5A: Please change color of MinC in the scheme so it is not the same as the one for FtsZ (weakened interaction).
- Line 605 - Methods: I understand that FtsZ concentration was 15 micromolar, not milimolar.
- Why FtsZ Bpa variants were expressed/induced using two different manners? If there is a reason why variants 2-230 couldn't be induced with tetracycline like the others, this should be stated in Materials and Methods.

Reviewer #3

(Remarks to the Author)

In this study by Yin, et al., the authors show using photocrosslinking and tandem mass spec. that residues (T8 and D10) in the FtsZ N-terminal IDR crosslink to residues in the polymerization domain. To investigate this further, they characterize two mutants, D10F and deltaIDR for in vivo Z-ring assembly and in vitro for GTP hydrolysis and polymerization. Both mutants, D10F and dIDR form filament-like structures in a cell and recruit other division proteins, including FtsA and FtsN, but fail to coalesce into condensed rings. Interestingly, the FtsZ mutant proteins appear less dynamic in vivo and in vitro for GTP hydrolysis, and they have a higher tendency to bundle. The results are very interesting, but the authors make several claims that seem to be over-interpreted, or could be verified through additional analysis. For example, many FtsZ mutants with modifications to the filament interface have slow GTP hydrolysis rates and show similar structures in vivo (i.e., Q47K or D212C in Redick, et al., 2005). While this does suggest that N-terminal residues may engage the protofilament interface, the supposition that the N-terminus is intrinsically a destabilizing cis element seems premature and not well supported. Moreover, the analysis of MinC failing to disrupt FtsZ-dIDR polymers could also be due to poor recognition by MinC, which may result from the mutation itself, or from the slower dynamics. Finally, it is not clear that the IDR itself plays a role, but that residues 8 and 10 may be important irrespective of the rest of the IDR. In fact, in the predicted structure of FtsZ (via AlphaFold), residue 10 is practically part of the polymerization domain and not the IDR.

In addition to these comments, the authors are asked to address the following:

1. How conserved are the N-terminal 1-9 residues?
3. Lines 40-42 – This is an oversimplification of FtsZ behavior, because while polymers assemble with GTP, polymers have been observed to contain GDP after hydrolysis by prior to disassembly (i.e., curved polymers). Also, which variant is referenced here, as many variants are described in the literature with altered hydrolysis activities.
2. Lines 78-84. What is the distance in angstroms spanned by the N-terminal IDR assuming an extended random coil? What

is the IDR residue cutoff? Residues 10, and also 8, would be really close to the polymerization domain and not really part of the IDR. Could the D10F substitution artificially create a more stable polymer by stabilizing the interface?

3. Fig. 1G – Where does A87 map on the structure in 1F?

4. If GDP is used in vitro instead of GTP as in Fig. 1G, are dimer interactions also disrupted. If not, then the GDP interface is different from the GTP interface.

5. Lines 78-93. “Single individual protofilaments (line 88)”. This should be corroborated with TEM images. However, Fig. 4D clearly shows that minimally FtsZ-dIDR bundles.

6. Lines 99-104. It is not clear how many residues were removed from deltaN-IDR (nine or 10?). It is assumed that an initiator methionine was inserted, resulting in the sequence ‘M-DAVIKV..’

Also, why are only these two mutants D10F and dIDR characterized? What about T8 substitutions or shorter N-terminal deletions? Does mutation of residue 9 show any phenotypic aberrations? What about the phenotypes and in vitro characterization of K51, S62, F40, and Q56? These are important mutations to analyze.

7. Line 100. Does the N-domain here refer to the polymerization domain? Again, residue numbers in the text would help. It is counterintuitive that the N-terminal IDR is not in the N-terminal domain.

8. Line 105-110. This result implies that the IDR is important for division- the suggestion that this defect tracks with the IDR-N-domain interaction is an overinterpretation.

9. Line 112 – Please clarify “entities”.

10. Line 184 – What is FtsZ^{WT29}. Is this a reference?

11. The experiments with MinC overexpression are difficult to interpret. Failure of MinC to disrupt dimers could simply be due to reduced MinC activity/binding, such as if the N-terminal residues of FtsZ overlap the MinC N-domain or C-domain binding sites. This should be tested directly in vitro.

12. If the model for MinC function is for the N-domain to bind to an open polymer interface to prevent additional FtsZ subunits from attaching, then an FtsZ protein with reduced dynamics should be more resistant to destabilization by MinC. FtsZ(G105S) is known to have slower dynamics because it has a mutation near the GTP binding site, is it also more resistant to MinC than wild type, similar to FtsZ-dIDR?

13. Line 190-194. This is an overstatement.

14. Line 204. Light scatter is non-quantitative and does not distinguish between a few bundles versus longer filaments versus many short filaments. Caution should be taken in comparing and interpreting difference in amplitudes of scatter. The TEM data favors that FtsZ-dIDR has a higher tendency to bundle laterally, and this is what may cause the increase in scatter.

15. Line 207. Clarify or restate “maintenance times”. This language is strange.

16. Fig. 4B – Why is there such a low percentage of wt FtsZ in the GTP pellet (also in 5F)?

17. Fig. 4C – Proportion is misspelled.

18. Fig. 5A. Proportion is misspelled. Architecturing is insufficiently descriptive. Please correct/clarify.

19. A recent BioRxiv paper further suggests that ZapA, which bundles FtsZ filaments, also engages the FtsZ N-terminal region (Fujita, et al., BioRxiv, 2024). This and the idea that other FtsZ-interacting proteins could engage the N-terminus should be discussed further.

Reviewer #4

(Remarks to the Author)

FtsZ is an exceptionally well-studied and essential bacterial cell division protein. In vitro FtsZ assembles head-to-tail to form single stranded polymers that interact laterally to form larger bundled structures. To date research has largely overlooked the short, intrinsically disordered N-terminal domain. Using previously a developed high throughput technique to evaluate the interactions of individual protein residues, Yin and Liu et al. present data supporting a role for the N-terminal IDR as a modulator of the longitudinal FtsZ-FtsZ assembly interface. Using UV crosslinking and mass spectrometry they identify residues on FtsZ's longitudinal interface that appear to interact with the N-terminal IDR. In vivo mutational analysis and microscopy, as well as in vitro biochemical assays further support the author's assertion that the identified residues play an important role in FtsZ assembly, although it is not clear that this is limited exclusively to the longitudinal interface. Genetic experiments indicating that a P203 substitution that weakens its interaction with residue K51) reduces the self-assembly of

FtsZ Δ NIDR were particularly appreciated. Finally, the authors utilize MinC and Lon protease to suggest that the N-terminal IDR plays a role in weakening the longitudinal interface. Combined these data suggest a mechanistically detailed role for the N-terminal IDR in facilitating Z ring localization and disassembly.

While the mass spec experiments support a role for the N-terminal IDR as a mediator of longitudinal interactions between FtsZ polymers, data indicating a critical role for this domain in FtsZ assembly and function in vivo is less compelling lessening the overall strength of the manuscript.

Major comments

Western blot methodology: were any steps taken to ensure sample uniformity between strains, such as total protein stains of the blot or normalizing samples by optical density/protein concentration? If so, please include this in the methodology.

577-578: The methodology for the FRAP experiments is highly unusual. "To reduce interference by FtsZ treadmilling, we used overnight-cultured cells with relatively stable Z-ring architectures." How many hours specifically were cells grown? How long were cells in stationary phase? What percentage of cells had Z rings? We would expect few cells to be forming Z rings and dividing in stationary phase cultures. This calls into question how well these experiments can represent active division.

Authors only mention using bright field in the context of Sup figure 3A, while other figures clearly include brightfield images as well. More care should be taken to specify which experiments included brightfield and when these images were taken during the experiment both in the methodology and figure legends. To this end, it's unclear if brightfield images were taken before and after fluorescence imaging, as is often standard. This is especially important for frap experiments, as the photobleaching is a harsh cellular perturbation.

4A. Δ NIDR appears to assemble far higher than expected for something that isn't bundling. Additionally, the TEMs in 4D may show some bundling as well. Taken together, this casts some doubt on the validity of 1G, where we can no longer be sure that dimers, trimers, etc. are only the result of lateral interactions. This is especially a concern because the "low" concentration of FtsZ used in 1G is not detailed.

Line 605: 15mM seems like an unusually high concentration, particularly compared to the other assays. Is this a typo? The typical concentration for in vitro assays of FtsZ is 5 μ M. Additionally, 5-minute time points is very infrequent- WT is already disassembling by the end of the first 10 minutes.

It would benefit the authors to cite Silber et al. 2020, as their findings may corroborate the assertion that the N-terminal IDR plays a role in FtsZ disassembly. Corrales-Guerrero 2018 also implicates the N-terminal IDR in FtsZ polymerization in cyanobacteria. Additionally, Vaughan et al. 2006 serves as a valuable reference for the conservation of the N-terminal IDR in FtsZ across species.

Other comments:

Generally, architecture is not a verb although it is used as such throughout this manuscript.

Assemble, organize, orchestrate, are more precise terms.

Line 33/34 more extensive citations are recommended.

Line 33 fluorescence microscopy (not fluorescent)

Line 156: This sentence is grammatically confusing. Do the authors mean that the two mutants can support normal divisome assembly and division? I am confused by the word "architecture" here.

Figure 1C. the right-side blot is numbered differently on the top than the bottom- appears to be a typo?

Sup 1C. It would be interesting to see what these other proteins that certain residues appear to be crosslinking.

2A could use a little more detail in the interpretation of the results, or at least more explanation of the experimental set up in the figure legend. The experiment itself does make sense on examination though.

3F. Fluorescence is difficult to see, please include images of individual channels, not just the merges.

Line 156: This sentence is written in a confusing way. Please simplify.

Line 590: The "kymograph analysis" method section might be better named "SIM-TIRF microscopy and kymograph analysis"

2E-F. To make the figures more colorblind accessible, we recommend using green and magenta instead of green and red.

Protein purification methods do not specify if a dialysis step occurred after eluting off the Hi-trap column.

Sedimentation methodology lists 3 concentrations for assays, but doesn't label which experiment uses which concentration.

4A and E. It is not immediately clear if there are replicates of this data. Please add clarifying language.

5H. Fluorescence is difficult to see, please include images of individual channels, not just the merges.

Line 290: While I agree that ring condensation isn't necessarily a prerequisite for divisome recruitment, this seems like a dramatic flattening of a process we still don't fully understand.

Reviewer #5

(Remarks to the Author)

Version 1:

Reviewer comments:

Reviewer #1

(Remarks to the Author)

I have thoroughly reviewed the revised manuscript, the comments posed by the other reviewers, and the responses of the authors to all the comments. Overall I feel that the authors have sufficiently addressed the concerns posed. The revised manuscript is now much stronger than the version originally submitted and should be accepted for publication.

I do have a few minor courtesy comments for the authors to consider before publication. I do not believe these require any further response from the authors:

1. Supplementary Fig 1a: residue "2" is labeled but is not colored according to the five categories added in revision.
2. Lines 358-360: as written it appears that FtsZ N-IDR deletion in cyanobacteria reduced the formation of more stable protofilaments, which is the opposite as reported in ref 42. This sentence should be modified as follows (or equivalent) "For example, it was shown that the FtsZ N-IDR deletion in cyanobacteria reduced GTPase activity and ****promoted**** the formation of more stable protofilaments"
3. Line 360: "while" suggests the following point will contrast with the previous, but instead it builds upon it. I suggest replacing it with "additionally,"

Reviewer #2

(Remarks to the Author)

The revised version of the manuscript shows a significant improvement. The authors have addressed some of the reviewer's concerns, providing a better characterization of the IDR role.

However, I am concerned about the MD experiments depicted in Figure 5:

The MD analysis indeed fits the mass spectrometry results, and suggests the competition between the N-terminal domain and the T7-loop for the N-IDR region. That said, the authors claimed that the "N-IDR transitioned to a folded state" and indicated that the N-IDR was flexible and mobile during the dynamics to finally interact with different residues. However, I am missing some data supporting these statements (perhaps in supplementary material) about the behavior of such regions during the full dynamics (for instance parameters such as RMSF, RG, RMSD..) including secondary structure analysis, and also the contact maps of the full protein from which Fig.5i was extracted. Also, according to Materials and Methods the first set of contact maps was calculated for the full dynamics (150ns) but the second only for the last 50 ns? I would assume that is always better to analyze only the last 50 ns in all cases, after initial stabilization of the system. And, if it was calculated using the full 150 ns how was separated for the initial situation, and how was the latter analyzed?

I have another question regarding the initial structure, which was obtained from a model using *S. aureus* 3VOA crystal structure. I understand that they selected this structure because it is in the open conformation. However, 3VOA was crystallized in a GDP-bound form. While this may not change the result of the MD, I would like to see an independent MD simulation using a model created after an *E. coli* structure. 6UNX corresponds to *E. coli* L178E-GTP bound, it is in the open conformation, and shows a similar conformation to 3VOA, however there are subtle differences in the subunit interface. This structure could be used as a model for the *E. coli* WT protein.

Also, regarding new results shown in Supplementary Figure 5d, the authors could consider if the low GTPase activity presented by FtsZ dNIDR-P203S could be due to low polymerization capacity (compared to WT) caused by the mutation of P203S. In my opinion, the experiments *in vivo* performed by the authors shed some light to the function of P203, but they could be complemented by *in vitro* assays, like TEM or sedimentation, to discern if the polymerization capacity is impaired. Low impairment could not be discarded even when the growth of the *E. coli* strain is not compromised.

In addition, I have a few minor comments:

For Light scattering experiments of BsFtsZ and SaFtsZ: In Materials and methods is written 10 μ M but 15 μ M in the legend figure.

Reviewer #3

(Remarks to the Author)

In this study by Yin, et al., the authors show that residues (T8 and D10) in the FtsZ N-terminal IDR crosslink to residues in the polymerization domain. To investigate this further, they characterize two mutants, D10F and Δ IDR for in vivo Z-ring assembly and in vitro for GTP hydrolysis and polymerization. Both mutants, D10F and Δ IDR form filament-like structures in a cell and recruit other division proteins, including FtsA and FtsN, but fail to coalesce into condensed rings. Interestingly, the FtsZ mutant proteins appear less dynamic in vivo and in vitro for GTP hydrolysis, and they have a higher tendency to bundle. The results are very interesting and suggest that N-terminal residues engage the protofilament interface and act as a destabilizing element. The revised version of the manuscript now includes molecular dynamic simulations that are consistent with their hypothesis and additional clarification of residues implicated in the interaction. The authors have addressed all of my concerns associated with the original version of the manuscript. Overall the work is noteworthy for understanding structural implications for FtsZ stability and is of general interest.

Reviewer #4

(Remarks to the Author)

We appreciate the authors' clarifications of the manuscript, especially regarding the methodology. However, now that the experimental methodology has been clarified, several critical experiments do not conform to standards in the field, making comparison with previous work difficult if not impossible.

Specifically:

"Response: We apologize for this confusion. The term "overnight-cultured" was improperly used because the cells were cultured for 10 h. Thus, we modified this as follows: "To reduce interference by FtsZ treadmilling, we used cells that were cultured for 10 h and had reached late logarithmic phase, a timepoint where most wild-type cells formed well-maintained Z-ring architectures." (lines 808–810 in the revised manuscript)."

Unless the authors can provide evidence otherwise, if these cells were cultured for 10 hours at 37°C in LB, they are almost certainly in stationary phase with division occurring only in a limited population. For these types of imaging experiments, the authors should employ established protocols. Yang et al. 2017 (doi: 10.1126/science.aak9995) provides exemplary methodology that is accepted by the field.

"Response: Due to technical constraints, we were unable to monitor filament formation at intervals shorter than 5 min. We also had to work with three samples at the same time and perform measurements one by one on a single piece of equipment. Our setup required removing samples from a temperature-controlled incubator for light scattering measurements. Such operations limited our ability to perform more frequent recordings on assembly dynamics. While we regret that we did not record initial filament formation stages in a more precise manner, our data demonstrated distinct differences between samples."

1. 15 μ M FtsZ is not a physiologically relevant concentration and is 3x higher than concentrations typically used in these types of experiments. GTP is depleted more quickly at high FtsZ concentration, leading to accumulation of GDP which itself inhibits FtsZ polymerization. This experiment should be repeated at more physiologically relevant FtsZ concentrations similar to those used in prior studies of FtsZ assembly.
2. It is unclear why the authors chose to incubate cuvettes at 30°C, rather than 25°C, which is standard.
3. The authors should use a spectrophotometer capable of continuous reads such that data points are close together (e.g. 10 seconds or less between reads). Closer time points will allow the authors to capture assembly dynamics in full and permit capture of an entire time course for a single sample, instead of collecting data for three samples at individual time points. The expectation is that there will still be substantial difference in light scattering between WT and mutants, and this approach will allow results to be compared to previously published work. (Using a spectrofluorimeter to take continuous reads is considerably easier for the operator too.)
4. If the authors are more concerned about capturing disassembly dynamics than assembly dynamics, it's recommended they use their existing data to estimate an appropriate incubation period before taking readings.
5. The authors should include in the main figure or the supplement data on baseline values before addition scattering begins. Some FtsZ mutants can have unusually high baselines compared to WT, which may be especially important for mutants which are impaired for disassembly.

Reviewer #5

(Remarks to the Author)

Version 2:

Reviewer comments:

Reviewer #2

(Remarks to the Author)

I would like to thank the authors for taking into account all suggestions and implementing them in the manuscript. By addressing the concerns of all reviewers, the manuscript convincingly establishes the impact of the N-terminal region of FtsZ in maintaining longitudinal interactions and increasing dynamicity to the Z-ring, and it is, in my opinion, suitable for publication.

Reviewer #4

(Remarks to the Author)

Respectfully, regarding the methodology, we remain unsatisfied with the authors' explanations. If experiments are not done using standard approaches, it is difficult if not impossible to compare results between studies. In the case of FtsZ, as outlined in our previous review, the methodology used by the overwhelming majority of labs in the field is straightforward and produces reliable, repeatable results. If the authors are unable to adapt standard methods to their needs, they should provide an explanation for their decision to use different approaches in the main text of the manuscript. E.g. In the case of data obtained at 15uM FtsZ please indicate that you also attempted these experiments at the more physiological concentration of 5uM but were unable to obtain reliable results.

With regard to point 5, baseline data should be included in supplementary data not source data.

Reviewer #5

(Remarks to the Author)

Email: chanzy@pku.edu.cn

Point-to-point response for manuscript NCOMMS-24-40635

REVIEWER COMMENTS

Reviewer #1 (Remarks to the Author):

The research article by Yin et al presents an exhaustive screen of all possible residues of FtsZ that may interact with other proteins, including other FtsZ subunits. The screen itself may be of significant general interest to the field, although it does suffer from some consistency issues. Nonetheless, through this screen, the authors uncovered a previously unappreciated role of the extreme N-terminus of FtsZ (NIDR) in regulating FtsZ polymer dynamics. From the various experiments presented, it appears that the FtsZ NIDR fine-tunes FtsZ polymers by reducing subunit affinity. In a Δ NIDR mutant, FtsZ polymers are stabilized, but this defect can be rescued by introducing the P203S mutation (which presumably destabilizes polymers enough to compensate for the loss of the NIDR?). Overall, the manuscript presents convincing data for a novel role of the NIDR, although the exact mechanism by which the NIDR affects polymerization is not entirely resolved.

The authors favor a model in which longitudinal interactions are directly impacted by the FtsZ NIDR, however the experiments presented do not adequately exclude the possibility of an effect on lateral interactions (and indeed, Fig. 4D appears to indicate such a possibility). Further characterization of the P203S mutant alone may help lend credence to the author's preferred mechanism. The manuscript also did not explore whether this mechanism might exist in other bacteria with other FtsZ proteins, which might also improve the general interest in these findings.

However, in general the authors need to take greater care when interpreting and presenting their data and experiments. There were critical experimental details left undefined, including the exact definition of the NIDR residues or the Δ NIDR mutant. In some cases, it was also unclear what strains were being used in each experiment, with some details available or only implied by the strain and plasmid list presented in the supplemental materials. Additionally, the authors need to exert greater caution in interpreting the presented data to avoid speculating further than the data supports. Overall these issues would be correctable with more careful and complete editing.

Comments

1. “Architecturing” is not a proper English word. Please replace all instances of “architecturing” (including in the Title) with “assembling” or “organizing”

Response: Thank you for this suggestion. We replaced all instances of "architecturing" with "assembling" throughout the manuscript, including the title, to ensure clarity and accuracy.

2. Many of the figures include text that is too small to be legibly printed, even by laser printer. Fig 1a and 5C in particular are too small, and the diagram depicting longitudinal self-assembly to the right of panel 1G is difficult to see.

Response: Thank you for this comment. We reformatted Figs. 1 and 5 (all panels), as well as Fig. 3f and 3g such that figure size was extended and all font sizes are no smaller than 5 pt.

Fig. 1a

Fig. 1g

Fig. 5c

3. Lines 36-38: Need to clarify the scope of this discussion as many bacteria do not have oscillating Min systems (or any Min systems at all).

Response: We modified these sentences as follows: “To shift the FtsZ polymer concentration toward midcell and away from the poles, extrinsic positioning *trans* factors are used by many bacteria. For example, the *E. coli* MinC protein, which exhibits pole-to-pole oscillations in cells, is believed to disassemble FtsZ protofilaments beyond the midcell position¹³.” (lines 39–43 in the revised manuscript).

4. Lines 41-42: This mutant requires greater context as the current writing suggests that GTPase activity is not very important for FtsZ function. The mutant in question is ftsZ84 (FtsZ G105S), which exhibits 10% of the GTPase activity of WT FtsZ. In the cited Sticker 2002 paper, this decreased activity carries distinct changes in FtsZ treadmilling behavior (increased midcell accumulation and slower recovery in FRAP experiments) consistent with significantly decreased GTPase activity.

Response: We revised this paragraph as follows: “However, the importance of GTPase activity for FtsZ functioning remains unclear. For example, the FtsZ84 variant (FtsZ^{G105S}), which has approximately 10% of the GTPase activity of wild-type FtsZ and significant defects in FtsZ treadmilling, and also slower recovery in fluorescence-recovery-after-photobleaching (FRAP), was shown to support cell division¹⁹.” (lines 48–52 in the revised manuscript).

5. Line 48: What is the evidence for the N-IDR? It is probably true that the extreme N-terminus (residues 1-11 or so) is disordered, but this would be a common trait of the extreme N terminus (or C terminus) of many proteins. What residues do the authors define as the N-IDR? This needs to be couched in background and context instead of defining it without any introduction.

Response: We apologize for this confusion. To define what the N-IDR is, we added the following paragraph to the Results section “Remarkably, residue positions showing crosslinking products were located across the whole polypeptide chain, not only in the previously defined N-domain (residues 11 - 176), C-domain (residues 201-316)^{23,24} and the C-terminal sequence (residues 317-383), but also in the extreme N-terminus (residues 1-10), which was previously under-characterized (Fig. 1b). More importantly, many residues were never previously reported to mediate protein-protein interactions

(Supplementary Fig. 1a). We then retrieved the *E. coli* FtsZ structure from the AlphaFold Protein Structure Database (AF-P0A9A6-F1-v4, <https://alphafold.ebi.ac.uk/entry/P0A9A6>) and also used Protein Disorder prediction System (PrDOS)²⁵ software to predict its structure. Interestingly, AlphaFold failed to predict a structure for the ten residues from the extreme N terminus, while PrDOS, with high confidence, predicted eight residues from this terminus as an intrinsically disordered region. Accordingly, we defined these 1 - 10 residues as an N-intrinsically disordered region (N-IDR). Further sequence alignment analysis of FtsZ proteins from different bacteria showed that they all possessed N-IDR regions, with various lengths and low amino acid sequence similarities²⁶.” (lines 66–80 in the revised manuscript).

6. The N terminus of FtsZ was previously implicated as an important domain for self-interaction in ref. 23. The highly conserved Ala 11 was previously identified as important for self-interaction (A11 at the bottom of one subunit interacts with L68, in the N-terminal domain at the top of the next subunit). This is adjacent to the residues 5 through 10 that have crosslinked bands in Fig. S1. Do the authors define the N-IDR only from residue 1 to residue 10?

Response: Yes, we defined the N-IDR as residues 1–10, as described in a newly added paragraph to the revised manuscript (see the reply to point 5).

7. Curiously, A11-Bpa itself does not crosslink, perhaps because A11 is crucial for longitudinal interactions and the relatively bulky Bpa replacing Ala at this position sterically interferes with this. Some of the other interacting residues in the crystal structure do not crosslink with Bpa in the present study, perhaps for the same reason. Could the authors comment on this?

Response: The failure to detect any photocrosslinking products, when Bpa was introduced at a certain residue position, was possibly due to the improper orientation of the Bpa side chain, based on our extensive experience with Bpa photocrosslinking. Importantly, not all residue-residue contacts could be detected by Bpa photocrosslinking. This may explain the failure to detect A11-L68 contacts, although we cannot absolutely rule out that the relatively bulky Bpa replacement of Ala at this position sterically interfered with longitudinal interactions.

8. Fig. S1A: The figure legend states that blue residues were found before from crystal structures or previous data by this group. This is insufficient information. Which blue residues were implicated by crystal structures (and which crystal structures) and which were previously identified by this group, and in what publication?

Response: Thank you for pointing this out. We made the following revisions. 1) We recategorized the photocrosslinked residues into the indicated five groups shown below (with the sources of publications indicated). 2) The blue residues in Fig. S1a of our original manuscript were roughly assigned. With our current stricter standard many residues have to be assigned as newly identified (i.e., become red colored now). 3) Residues 323-357 were relabeled as “Residues putatively involved in protein-

protein interactions” though they were verified to be self-contact sites, because these data were not included in our manuscript.

We added the following description in the legend of Supplementary Fig. 1a: “The residues that formed crosslinked products are categorized into the indicated five groups; of note, residues 323-357 were verified to be involved in self-contacts though the data are not shown here. “Residues hinted to be involved in FtsZ longitudinal assembly” were defined as interface residues located within 4.5 Å to a neighboring subunit; “Residues hinted to be involved in FtsZ lateral assembly” were defined as residues located near one previously reported as mediating lateral assembly.”

9. FtsZ residues that this group has found before (presumably ref. 18) to crosslink such as R78, N79, D82, R85, R89, K155, S231 were claimed to be lateral interactions. Nonetheless, they did not show up in Fig. S1, which theoretically should detect any interaction, longitudinal or lateral; the genetic makeup of the strains seems to be identical. Is it because lateral interactions are much weaker under these conditions compared with longitudinal interactions and thus would be below detection limits?

Response: Yes, you are correct! The crosslinking bands generated by lateral interactions were in general weaker when compared with longitudinal interactions, and thus more difficult to detect by our high-throughput SDS-PAGE. We added one more group as “Residues hinted to be involved in FtsZ lateral assembly (based on PMID:29889022)” for those crosslinked residues located near a residue previously reported as mediating lateral assembly.

10. Fig S1A indicates numerous residues that are labeled in red as self-interaction residues. Some of these residues (eg, 5,6,7,9) fail to show confirmatory crosslinking with FtsZ-avi on either the anti-FtsZ blots (as double bands) or the streptavidin blots (as single bands) shown in Fig 1C or Fig S1B. Additionally, there are some residues (such as 58 and 59) that are not indicated on Fig S1A but nonetheless appear to form crosslinks on the confirmatory blots of Fig S1B. What is different about the experiments in Fig S1A versus Fig 1C / Fig S1B that would explain the loss or appearance of apparent crosslinking bands? Can these be called self-interaction residues if they cannot be verified?

Lacking sufficient supporting confirmatory evidence, the authors need to walk back claims of FtsZ self assembly roles for some of these residues (e.g., Fig S1A, lines 66-70). It would be helpful if the authors provided a table or a list to summarize results across the various blots for (for example) which residues formed any crosslinks versus those that formed verified crosslinks with other FtsZ proteins or other proteins, or that were simply untested (as for the 325-329 residues).

Response: We apologize for these confusions.

In particular, Fig. S1a was generated using a single high-throughput SDS-PAGE gel (indicated in the figure legend) which was previously developed by our laboratory (Ref. 22); it allows 384 samples to be analyzed on a single gel, but only very small sample amounts (0.5 μ L) can be loaded into lanes, and also separating distances were not long. In contrast, Figs. 1c and S1b were generated using several regular SDS-PAGE gels, which allowed significantly more protein samples to be loaded (3.5 μ L), and also they had much longer separating distances. Therefore, resolution was much higher than in high-throughput gels. In other words, the detection threshold was much lower in Fig. S1b than in Fig S1a, and explained why some residue variants showed weak crosslinking bands (e.g., residues 58 and 59) in Fig. S1a showed clearer crosslinking bands in confirmatory blots in Fig S1b.

The reason why crosslinked product bands between FtsZ-Avi and FtsZ^{Bpa} were barely visible for residues 5, 6, 7, 9 in Fig. 1c was mainly due to relatively low FtsZ-Avi levels in comparison to FtsZ^{Bpa} in cells. This meant that the top band of the double band, when probed with anti-FtsZ antibodies (right gel in Fig. S1b) and the single band when probed with Streptavidin conjugates, was barely visible. Indeed, after adjusting the contrast, we observe these bands (see below).

Furthermore, our mass spectrometry results also showed that residues 1, 2, 3, 4 and 5 interacted with residues in the N-domain (Table 1). Taken together, we consider that these residues participated in self-assembly during FtsZ assembly in cells.

According to reviewer suggestions, we recategorized the photocrosslinked residues into the indicated five groups as described in our response to your comment #8.

“The results (Fig. 1C and Supplementary Fig. 1B) demonstrated that almost all these residues (as well as other residues showing crosslinking product bands in Supplementary Fig. 1A) mediated FtsZ self-assembly interactions. Strikingly, many of these self-assembly residues located in the N-IDR and N-domain (colored red) were revealed for the first time.” (lines 66-70 in the original manuscript). This description was replaced by the following one: “These results (Fig. 1c and Supplementary Fig. 1b) showed that almost all these verified residues were involved in FtsZ self-interactions. Strikingly, many of these self-assembly residues located in the N-IDR and N-domain (colored bold red) were revealed for the first time.” (lines 88-91 in the revised manuscript)

11. In Fig S1A, there are bands visible in some lanes that are neither labeled as novel nor previously identified. Residue 31, for instance, which is revealed by another blot to be an interaction with a different protein. Additionally, residues, 86, 90, 95, 103, 142, 237, 239, 323, 330, 343, 344, 355, 357 (and others?) also appear to form bands of some sort. Even if the interaction partners are not identified, these putative crosslinking sites may be useful findings for other researchers interested in FtsZ protein interactions and should also be summarized, blot reliability notwithstanding.

Response: We agree with these statements and now included these residues in the group of “Residues putatively involved in protein-protein interactions” as described in our response to your comment #8. We did not label these residues based on our relatively high cutoff standard for the results to be considered positive.

12. Fig 1D and Table S1 indicates that Bpa at T8 and D10 crosslinks residue V42 with the highest or second-highest abundance. However, V42 is fairly deeply buried inside FtsZ, so it is not clear how this interaction would occur.

Response: Our *in vivo* photocrosslinking and mass spectrometry results clearly revealed an interaction. The fact that this interaction was forbidden based on the determined structure suggests that other alternative FtsZ protein conformation may occur in cells. Indeed, in the *Klebsiella pneumoniae* (8IBN pdb) FtsZ structure, 44~56 residues tended to be in a flexible loop instead of a helix, thus making it possible for the N-IDR to interact with V42.

This hitherto unexpected result was consistent with our final conclusion that the N-IDR may function as a disassembly element by modulating FtsZ protein conformation.

13. Lines 88-93 and Fig. 1G: here, A87 seems to exhibit lateral interactions (but not the other residues listed in point 5). The evidence is that there is no ladder like there is with T8 and K51. However, the crosslinking for A87 is much weaker than that for T8 and K51, so perhaps a ladder cannot be detected at this exposure. Moreover, these results do not clearly demonstrate that lateral dimers can not also be formed. Lines 88-93 should be re-written to reflect this. Bear in mind that the mass spec results (Fig 1/Table S1) show N-IDR interacting with residues Q56, F40,

S62 that are not at the longitudinal interface, but are in close proximity to A87 at the lateral interface. It remains plausible that the N-IDR could also play a role in lateral interactions as well.

Response: We thank the reviewer for making these critical points. Indeed, we cannot conclude that no lateral interactions occur simply based on crosslinking band levels, given that lateral interactions are much weaker than longitudinal interactions. But the fact that the crosslinked band formed by 87^{Bpa} (e.g., the dimer form) remained unchanged upon GTP addition, unlike the increase in band ladders for 8^{Bpa} and 51^{Bpa}, demonstrates the lack of formation of crosslinked lateral dimer.

Furthermore, the FtsZ concentration was limited to 1 μ M, under which FtsZ assembled into single protofilaments with almost no lateral interactions (see Fig. 4d).

We do not consider residues T8 and K51 are involved in lateral interactions, but peripheral to the conventionally defined longitudinal interface.

To incorporate these points, we modified lines 88–93 in the original manuscript as follows: “The results (Fig. 1g) showed that T8 (in the N-IDR) and K51 (in the N-domain) residues both appeared to be intimately linked to longitudinal rather than lateral FtsZ assembly. This was indicated by the fact that Bpa variants of both residues formed a photocrosslinked product band ladder (lanes 4–6 and 10–12), which was greatly increased in the presence of GTP. This observation contrasted to when Bpa was introduced at residue A87 (lanes 16–18), which failed to form a band ladder, and low dimer levels remained unchanged in the presence of GTP. Residue A87 (its position is shown in Fig. 1f) was assumed to participate in lateral interactions based on the fact that it was located between R85 and R89, both of which were reported to mediate lateral interactions with FtsZ protofilaments²¹. A87 was selected here because 87^{Bpa} crosslinking products were detectable in our longitudinal interaction crosslinking analysis (Supplementary Fig. 1a).

Of note, interactions involving the N-IDR were largely missed in previous studies^{28–31}. Furthermore, we noted that the N-IDR interacted with residues (e.g., F40, Q56, and S62) peripheral to those previously defined at the longitudinal interface³⁰. Therefore, this posed a series of questions; did this interaction have a role in the longitudinal assembly of FtsZ, Z-ring formation, and cell division?” (lines 109–125 in the revised manuscript).

14. Ref. 18 is cited for A87-Bpa mediating lateral interactions, but there does not seem to be a mention of A87 in Ref. 18.

Response: We apologize for this. In Ref. 18, residues R78, N79, D82, R85, R89, K155, and S231 were described as mediating lateral interactions. However, these crosslinking products were too low to be detectable in the same gel when crosslinking products of longitudinal interactions involving T8 or K51 were analyzed (longitudinal interactions are much stronger than lateral interactions). Nevertheless, the crosslinking products of residue A87 were detected as a faint band in our gel analysis (Fig. S1a). Given its closeness to R85 and R89, A87 was hypothesized to mediate lateral interactions.

To incorporate these points, we modified the following lines: “Residue A87 (its position is shown in Fig. 1f) was assumed to participate in lateral interactions based on the fact that it was located between R85 and R89, both of which were reported to mediate lateral interactions with FtsZ protofilaments²¹. A87 was selected here because 87^{Bpa} crosslinking products were detectable in our longitudinal interaction crosslinking analysis (Supplementary Fig. 1a).” (lines 115–120 in the revised manuscript).

15. Lines 89 and 94 make reference to the T8 (N-IDR) and K51 ‘mediating’ longitudinal self-assembly. At this point, there is not sufficient evidence to state that T8 or K51 are required for (or are mediating) longitudinal interactions. Given as the crosslinking interactions occur despite the Bpa substitution at these residues, they are unlikely to directly play an essential role in the interaction as well. Indeed, Fig 2 provides ample evidence that FtsZ can assemble into filaments without the N-IDR.

Response: We agree with these statements and modified them accordingly.

Line 89 in the original manuscript: “The results (Fig. 1G) clearly indicated that residues T8 (in N-IDR) and K51 (in the N-domain) both mediated longitudinal rather than lateral FtsZ assembly.” This has been modified as follows:

“The results (Fig. 1g) showed that T8 (in the N-IDR) and K51 (in the N-domain) residues both appeared to be intimately linked to longitudinal rather than lateral FtsZ assembly.” (lines 109–111 in the revised manuscript).

Line 94 in the original manuscript: “We strongly emphasize that longitudinal self-assembly interactions mediated by the N-IDR and N-domain were revealed here for the first time, and were almost completely missed in previous studies^{21–24}, evidently due to N-IDR invisibility in resolved crystal structures.” This has been modified as follows:

“Of note, interactions involving the N-IDR were largely missed in previous studies^{28–31}. Furthermore, we noted that the N-IDR interacted with residues (e.g., F40, Q56, and S62) peripheral to those previously defined at the longitudinal interface³⁰. Therefore, this posed a series of questions; did this interaction have a role in the longitudinal assembly of FtsZ, Z-ring formation, and cell division?” (lines 121–125 in the revised manuscript).

16. Lines 94-97: It is true that there was no previous report of extensive interactions between the N-IDR (assuming that it is an IDR, see comment #1) and the rest of the N-terminus in refs. 23-24, but as mentioned in comment #5 above, there was a clear mention of interaction between A11 and L68 in ref. 23 as being involved in longitudinal interactions. It therefore depends on how the authors define the N-IDR.

Response: Indeed, as described above, we defined the N-IDR as covering residues 1–10.

17. The deletion of N-IDR needs to be defined. Is it the deletion of residues 1-10?

Response: We apologize for the confusion. We deleted residues 1–10 but retained the first residue Met for protein expression purposes.

We modified the description as follows: “To this end, we disrupted this interaction by removing the N-IDR (i.e., residues 1–10, but retained the first Met for protein expression purposes) and constructed an FtsZ^{ΔNIDR} variant.” (lines 128–131 in the revised manuscript).

18. Lines 102-104 and Fig. 2A: the genetic constructs need to be described more fully in the text and figure legend, not only in the Methods; even in the Methods the strains are not well described. The reader should not have to look up a previous paper to get basic information about a strain used extensively in the current study (also in Fig. 5E). If the cultures in Fig. 2A (and 5E) have an *ftsZ* null allele in the chromosome and pJSB100-FtsZ and pJSB100-FtsZ delta N-IDR plasmids under arabinose control, then why does repressing the plasmid promoter with glucose result in viability for the strain carrying WT FtsZ on the plasmid? The data in 5E imply that WT FtsZ is under arabinose control; so then what controls expression of the FtsZ variants?

Response: We apologize for this lack of clarity. To better understand our thought process, we modified the text in Figs. 2a, 5e, Supplementary Figs. 4a and 5a. Fig. 2a is pasted here. Indeed, we used the pTac plasmid (rather than pJSB100 that was used to conditionally express wild-type FtsZ for rescuing the survival of the FtsZ knockout strain) to constitutively express FtsZ^{ΔNIDR}, FtsZ-T8 mutants, and FtsZ-D10 mutants for testing (as well as the positive control FtsZ-WT).

We added the following description to the text.

“In particular, functional complementation analyses were performed using the *ftsZ* knockout strain, where pJSB100 was used to conditionally express wild-type FtsZ (in the presence of arabinose) to rescue its survival. The FtsZ^{ΔNIDR} was constitutively expressed from the pTac plasmid in this strain to test its function.” (lines 133–137 in the revised manuscript).

We also modified the legend in Fig. 2a: “a Complementation results for FtsZ^{ΔNIDR} and FtsZ^{D10F} supporting Δ *ftsZ* strain growth (contains the rescue plasmid pJSB100) where FtsZ^{ΔNIDR}, FtsZ^{D10F} or FtsZ^{WT} (positive control) were constitutively expressed from a pTac plasmid.” (lines 947–950 in the revised manuscript).

We also modified descriptions in the Methods section: “This was performed using the *ftsZ*-knockout strain LY928- Δ *ftsZ*. This strain carried the pJSB100 rescue plasmid, resulting in cell survival upon wild-type FtsZ induction with 0.05% arabinose, but cell death in the presence of 0.2% glucose (without arabinose). FtsZ variants were tested in this strain (under the latter conditions) by constitutive expression from the pTac plasmid^{21,39}; pTac plasmids expressing and not-expressing wild-type FtsZ were used as positive and negative controls, respectively.

Briefly, cells were cultured overnight at 37°C in LB medium plus 0.05% arabinose. Then, 5 μ L of 10⁻¹–10⁻⁶-fold diluted cultured cells were spread onto plates containing either 0.05% arabinose (positive control group) or 0.2% glucose (test group). Any FtsZ variant that allowed LY928- Δ *ftsZ* cells to grow in a comparable manner in the presence of glucose or arabinose was considered fully functional.

Of note, the LY928 strain used in photocrosslinking analysis was also used here

(solely for the purpose of consistency though not necessary).” (lines 749–761 in the revised manuscript).

19. Also, the strain with the deletion of FtsZ is still partially viable on the plates with arabinose. This contradicts the statement that this variant “failed to support cell growth”

Response: We assume you are referring to the residual cells seen for FtsZ^{ΔNIDR} on these plates (containing glucose not arabinose) in Fig. 2a. Here, we are actually testing cell proliferation (division) rather than cell growth; even defective cells will grow for certain periods and then die due to multiple cell division failures as shown by microscopy (Supplementary Fig. 3a).

We modified the description as follows (replacing “growth” with “proliferation”): “Our functional complementation analyses showed that this variant failed to support cell proliferation (Fig. 2a; second row from the top), indicating that the N-IDR was indeed essential for FtsZ functioning.” (lines 131–133 in the revised manuscript).

20. In the FtsZ sandwich fusion that is viable, mNeonGreen is inserted between residues 55 and 56 of FtsZ. How could this region be important for FtsZ assembly if it is disrupted by the large mNeonGreen moiety?

Response: We explain this as follows. Firstly, FtsZ-mNeonGreen⁵⁵⁻⁵⁶ was constructed by inserting mNeonGreen into that position using a flexible 10-amino-acid linker on both sides. This may reduce the disruptive effects on N-IDR function. Secondly, the N-IDR may participate in interactions without strong specificity, thus it can tolerate the disruption. Thirdly, most modified cells exhibit abnormal Z-ring morphologies (as shown below) as noted by the authors; “In contrast, BW27783 cells expressing mNeonGreen genomically had mostly one Z ring per cell (Fig. 5C). Some were sharp and completely normal (Fig. 5C, upper left corner), but most were somewhat fuzzier, and some were slanted.” (Reference: 10.1128/jb.00553-16). Interestingly, these abnormal phenotypes were somewhat similar to FtsZ^{D10F} phenotypes indicating potential disruption of N-IDR-involved interactions in their cases.

21. Line 114: How is FtsZ constitutively expressed in this strain?

Response: This was performed using a modified BBa_J23105 promoter on the pTac plasmid. We screened the Anderson promoter collection and selected the synthetic constitutive promoter BBa_J23105 that expressed FtsZ (or its variants) at levels comparable with endogenous FtsZ. We then replaced the middle sequence (between -10 and -35 sequences) of the P₂₃₁₀₅ promoter with the *tetO* operator sequence. We further added a λ t1 transcriptional terminator sequence to the front of the P₂₃₁₀₅ promoter allowing it to be a constitutive promoter in a strain lacking TetR (a transcription repressor binding *tetO* sequence). But this became a regulatable promoter in strains expressing TetR.

We modified the description as follows: “... (constitutively expressed from the pTac plasmid and driven by the P_{T-23105}, a promoter modified from BBa_J23105) ...” (lines 147–149 in the revised manuscript).

We also modified promoter descriptions in the Methods section: “Another pTac plasmid containing the P_{T-23105} promoter was used to generate FtsZ variants where residues 231–383 were individually replaced by Bpa, such that variant protein expression only occurred in the presence of TetR and anhydrotetracycline. This stringent control strategy prevented the production of potentially toxic truncated FtsZ forms (which might be produced when the introduced TAG codon was read as a stop codon rather than one encoding Bpa) that would cause lethal phenotypes before inducer addition.

The P_{T-23105} promoter was modified from the BBa_J23105 promoter (selected from the Anderson promoter collection (parts.igem.org/Promoters/Catalog/E.coli/Constitutive)). The middle sequence between -10 and -35 sequences (bold) in the P₂₃₁₀₅ promoter was then replaced with the *tetO* operator sequence (underlined) and the λ t1 transcriptional terminator sequence (italics) was added upstream of the P₂₃₁₀₅ promoter. This allowed the promoter (*CGCAAAAACCCCGCTTCGGCGGGTTTTTCGCTAAGGGATTTTGGTTTTA CTCCCTATCAGTGATAGATACTAT*) to function constitutively and expresses FtsZ (or variants) at levels comparable with endogenous FtsZ in a strain lacking TetR (a transcription repressor binding the *tetO* sequence), but function in a regulatable manner in a strain expressing TetR (from pYR5C-tetR plasmid).” (lines 612–628 in the revised manuscript).

22. Lines 125-129: This paragraph is very unclear as written.

Response: We agree; this paragraph is unclear and has been modified it as follows: “These observations indicated that although the N-IDR interacted with sites peripheral to the previously defined longitudinal interface, it was indispensable for cell division. Given that N-IDR removal as a whole produced a lethal phenotype, we then constructed more N-IDR variants to verify and characterize more about N-IDR action mechanisms.” (lines 160–164 in the revised manuscript).

23. Line 131-134: This sentence implies that multiple positions within the FtsZ N-IDR were subjected to this mutational analysis (“by replacing residues in the FtsZ N-IDR”), however only mutations for D10 are discussed. Were any residues besides D10 mutated, or was only D10 mutated? This should be clarified.

Response: We actually selected residues T8 and D10 for mutational analysis, given that they showed the most pronounced photocrosslinking bands (Fig. 1c and Supplementary Fig. 1a). However, when T8 was replaced by the other 19 amino acids, none appeared to interfere with LY928- Δ ftsZ strain proliferation. We added the functional complementation results of these variants to a new Supplementary Fig. 4a and added the following descriptions.

“These two residues were selected for mutational analysis as they showed the most pronounced photocrosslinking bands (Fig. 1c and Supplementary Fig. 1a). Interestingly, when T8 was replaced by the other 19 amino acids, none appeared to interfere with LY928- Δ ftsZ strain proliferation. However, among the 19 D10 variants, approximately half showed significant defects in cell proliferation (Fig. 2a, right panel for FtsZ^{D10F} and Supplementary Fig. 4a for the rest), and three examined variants (FtsZ^{D10F}, FtsZ^{D10Q}, and FtsZ^{D10Y}) showed Z-ring architecture defects (Supplementary Fig. 4b).” (lines 168–174 in the revised manuscript).

Based on what we observed later in our study, these results may be explained as follows. T8 variant results suggested that N-IDR function did not rely on highly specific interactions (Supplementary Fig. 4a). D10 variant results were likely due to its location at a site where certain replacements disrupted proper N-IDR positioning. Of note, functional complementation assays only showed dramatic defects in cell proliferation, and thus the failure to see T8 variant defects (as well as certain D10 variants) could not completely rule out that some of them may have caused certain Z-ring formation defects.

24. Fig 2 B-G: FtsZ-mNeonGreen (B/C/G) or FtsZ-mScarlet (E/F) appear to be expressed in trans with the native FtsZ. This indicates that the negative effects observed with the mutants must therefore be dominant over native FtsZ. Additionally, native and mutant FtsZ subunits may co-polymerize, so native subunits may be responsible for the recruitment activity observed. This limits the ability to interpret these experiments.

Response: We could not completely exclude the possibility that native subunits were responsible for the observed recruitment activity. But defective structures were mainly formed by mutant proteins. We modified the sentence as follows: “Surprisingly, we observed that these defective Z-ring architectures, that were mainly assembled by FtsZ^{D10F} and FtsZ ^{Δ NIDR}, still recruited FtsA (Fig. 2e) and also FtsN (Fig.

2f). FtsA is a key membrane anchoring protein for FtsZ^{33,34}, and FtsN is the last essential protein recruited to the divisome^{5,35}.” (lines 199–202 in the revised manuscript).

25. Lines 156-161: It is interesting that the aberrant FtsZ structures can recruit FtsA and FtsN, but this has been reported before.

For example, aberrant non-ring FtsZ structures are known to recruit FtsA and ZipA to the same structures (PMID: 10601211) and short non-ring polymers of FtsZ can drive cell constriction events (PMID: 28438890).

Additionally, the FtsN co-localization is limited to regions where FtsZ is forming a midcell band in Fig 2F, as co-localization is not observed with FtsZ bands formed elsewhere. The cell wall biosynthesis at these structures is also not particularly convincing, as the HADA/FtsZ patterns are mostly not overlapping in the images shown. It would also be much easier to see the HADA fluorescence in Fig 2G if the dark blue pseudocolor were changed to a higher contrast color.

Response: Thank you for your comments. A previous study (PMID: 10601211) suggested similar FtsA and ZipA localization patterns on non-ring FtsZ structures using immunostaining in different individual cells. However, we directly visualized the co-localization of these proteins using live-cell fluorescence microscopy in real time.

We are not sure how our observations could be correlated with theirs (PMID: 28438890) given that different approaches were taken. For example, the other group analyzed a Z-ring that was being normally formed, while we analyzed a malformed Z-ring at semi-condensed and dispersed stages. Furthermore, we used live-cell imaging, while they used cryotomography.

Unlike FtsA, FtsN (membrane protein) does not directly interact with FtsZ. As a result, we could only observe co-localization when FtsZ was assembled to a high degree. For HADA incorporation, it is indirectly related to FtsZ assemblies. Accordingly, we would not expect perfect co-localization between FtsN or HADA and FtsZ, but the co-localization signals are significant to us.

The pseudo color was altered to a higher contrast color, and the new Fig. 2g is shown below.

26. What is more surprising is that FtsZ lacking the N-IDR can assemble into polymeric structures. It is not clear how this occurs if the N-IDR is required for FtsZ longitudinal subunit interactions. It looks more like the N-IDR facilitates lateral interactions between FtsZ protofilaments, where lack of these lateral interactions would result in polymers that could not coalesce into a coherent Z ring. This is essentially stated in lines 163-164.

Response: We did not use the correct wording in describing the interactions involving the N-IDR and thus misled you. We apologize for this. We changed the wording accordingly (mainly replacing “mediate” by “involve”) by avoiding the implication that the N-IDR was required for FtsZ longitudinal subunit interactions.

“Of note, interactions involving the N-IDR were largely missed in previous studies^{28–31}. Furthermore, we noted that the N-IDR interacted with residues (e.g., F40, Q56, and S62) peripheral to those previously defined at the longitudinal interface³⁰.” (lines 121–123 in the revised manuscript).

We agree that these observations hinted at the possibility that the N-IDR may have facilitated lateral interactions. However, these are not supported by the following observations.

Firstly, FtsZ N-IDR and D10F mutant phenotypes were different to common mutants which disrupted lateral interactions; while the former mutant displayed strong but dispersed fluorescence entities, lateral interaction mutants commonly showed reduced, weakened, or even absent FtsZ fluorescence in cells. This was shown by our previous data (seen below in merged fluorescence and bright field images) where the K121L mutant represented one such disrupted lateral interaction (<https://doi.org/10.7554/eLife.35578>).

Below are our repeating data from this live-cell imaging analysis showing only the fluorescence channel and much better contrast images (from Dr. Yu Jiayu's PhD thesis).

The fact that $FtsZ^{\Delta N-IDR}$ and $FtsZ^{D10F}$ mutants still assembled into clearly visible entities in cells indicated, intriguingly, that neither longitudinal interactions nor lateral interactions were disrupted.

Secondly, our *in vitro* crosslinking analysis (Fig. 1g) also strongly suggested that the N-IDR was involved in longitudinal rather than lateral interactions.

We added the following sentences summarizing the observation described in the first point: “Of note, $FtsZ^{D10F}$ and $FtsZ^{\Delta N-IDR}$ phenotypes were different when compared to common mutants which disrupted lateral interactions; namely, while the former showed strong but dispersed fluorescent entities, lateral interaction mutants commonly caused reduced, weakened, or even absent FtsZ fluorescent entities in cells²¹. These observations suggested that neither longitudinal interactions nor lateral interactions were disrupted for $FtsZ^{D10F}$ and $FtsZ^{\Delta N-IDR}$ assembly in cells.” (lines 193–198 in the revised manuscript).

Additionally, if the N-IDR facilitated lateral interactions between FtsZ protofilaments, it would be hard to explain how a single P203S mutation on the longitudinal interface could rescue $FtsZ^{\Delta N-IDR}$ function, as described later in the manuscript (Fig. 5e).

27. Fig. 3A: It is puzzling why photobleaching recovery time for the huge ribbons made by FtsZ D10F is not that much lower than that of WT FtsZ in rings. It is hard to imagine how such ribbons remain that dynamic.

Response: We also expected that the $FtsZ^{D10F}$ mutant might have shown a much lower dynamicity. Nevertheless, the difference is significant (about 2-fold) and further similar degree of differences were consistently demonstrated in kymography analysis (Figs. 3c and 3d), *in vitro* assembly properties (Figs. 4a–c), and resistance to MinC disassembly capacity (Figs. 3f and 3g, 4g and 4h).

28. Line 173: which FtsZ “entities” were chosen for velocity measurements, and which were not? From the micrographs, it looks like most of the FtsZ structures remained static over the time course.

Response: We apologize for this confusion, which was mainly due to improper marking with a triangle. We have now used arrows which clearly illustrate the moving entities in micrographs. The modified fig. is shown.

29. Lines 173-175: There is a notable difference between the FtsZ velocities reported in the text and those represented in Fig 3D. The text gives 39.09, 22.9, and 9.43 nm/s for WT, D10F, and ΔNIDR. The cross bars in the figure appear to represent values of about 42, 28, and 14 nm/s, respectively.

Response: We apologize for this error and have corrected this as follows: “**d** Statistical treadmill velocities of indicated FtsZ entities are calculated using recorded kymographs and show treadmill velocities for FtsZ^{WT}, FtsZ^{D10F}, and FtsZ^{ΔNIDR} at 42.53±1.886, 28.19±0.9758, and 13.77±0.8098 nm/s, respectively;” (lines 974–977 in the revised manuscript).

30. Line 184 and following: The conclusion that MinC overproduction perturbs FtsZ structures less efficiently in the D10F and delta N-IDR variants is not that convincing from the images in Fig. 3F. Perhaps showing the fluorescence channel only, without the DIC overlay, would help.

Response: Thank you for this suggestion; we have presented the fluorescence channel only and indeed, they appear much clearer. Modified Fig. 3f is shown below.

31. Lines 187-189: the conclusion that the residual photocrosslinked FtsZ dimer levels were significantly higher for the delta N-IDR variant of FtsZ after MinC overproduction is not that convincing. Yes, the band is somewhat stronger than

for WT FtsZ or FtsZ D10F, but still much weaker than the uninduced controls, and higher order polymers were not detectable. Was this result reproducible? Were cells harvested for imaging in Fig. 3F also used for the crosslinking? There is no mention of how the cells were harvested for the crosslinking.

Response: The cells harvested for imaging in Fig. 3f were different to the cells used for crosslinking in Fig. 3g. In particular, cells used for imaging analysis contained the *ftsZ-mneongreen* allele (*ftsZ^{WT}-mneongreen*, *ftsZ^{D10F}-mneongreen*, *ftsZ^{ΔNIDR}-mneongreen*) in the genomic DNA, while these strains were not used for crosslinking analysis to avoid any interference from these FtsZ forms.

We detected higher crosslinked polymers for FtsZ^{WT} and FtsZ^{D10F}, but not for FtsZ^{ΔNIDR} on the original gel (shown below), which reflected the crosslinking features of the FtsZ^{K51Bpa} variant. In particular, the FtsZ^{K51Bpa} variant of FtsZ^{ΔNIDR} formed crosslinked products between each other (barely visible on the gel) and with FtsZ-pdt (clearly visible on the gel). The levels of crosslinked products with FtsZ-pdt could have partially reflected the assembly status of FtsZ^{ΔNIDR}-containing polymers. These crosslinking experiments were largely reproducible, but MinC expression levels had to be carefully tuned.

Of note, our crosslinking analysis only partially reflected our live-cell imaging analysis. We added the following paragraph to clarify this.

“Of note, cells used for imaging analysis (Fig. 3c), but not those for crosslinking analysis (Fig. 3g), contained a *ftsZ-mneongreen* allele in the genomic DNA. Here, the FtsZ^{K51Bpa} variant of FtsZ^{ΔNIDR} formed crosslinked products between each other (barely visible on the gel) and with FtsZ-pdt (clearly visible on the gel; this pdt-tagged FtsZ was expressed from genomic DNA). The level of crosslinked products with FtsZ-pdt partially reflected the assembly status of FtsZ^{ΔNIDR}-containing protofilaments. Therefore, our crosslinking analysis partially reflected live-cell imaging results.” (lines 233–239 in the revised manuscript).

32. Perhaps expressing MinC at lower levels (or shorter times) would provide a better contrast between crosslinked dimer band intensities. In any case, the data with purified FtsZ and MinC shown in Fig. 4 are more convincing.

Response: We agree that if MinC was expressed at proper levels, the contrast between crosslinked dimer band intensities would be better.

However, as mentioned in the response to comment #31, it was very difficult to tune MinC levels in cells, unlike *in vitro* experiments with purified FtsZ and MinC. Indeed, MinC levels (as shown in Fig. 3g) were generated as a consequence of screening multiple induction conditions, therefore, we only see distinctions in a qualitative rather than a quantitative manner.

33. In Fig. 3G, why is there only 1 crosslinked dimer band (lane 10) vs the doublet bands in lanes 2 and 6?

Response: In lanes 2 and 6, the upper band of the doublet band represents crosslinking between FtsZ^{Bpa} and FtsZ-pdt, and the lower band represents crosslinking between FtsZ^{Bpa} and FtsZ^{Bpa} (Fig. 5c). In lane 10, the lower doublet band, which represents crosslinking between FtsZ^{ΔNIDR-K51Bpa} and FtsZ^{ΔNIDR-K51Bpa} does not exist due to the absence of the N-IDR. See also our response to comment #31.

34. Speaking of MinC overproduction, the Methods state that minC was induced by AHT overnight and then 2 additional hours after dilution of the culture. It would probably be more physiologically relevant to induce MinC only in the exponentially growing culture for a short time prior to imaging.

Response: We apologize for this unclear description. We modified this as follows: "For live-cell imaging examining the effects of overexpressed MinC on FtsZ variant

assembly, *ftsZ-WT-mneongreen*, *ftsZ-D10F-mneongreen*, and *ftsZ-ΔNIDR-mneongreen* cells were cultured overnight in the presence of rhamnose (0.5%). Cells were then re-cultured in fresh medium for an additional 2 h before MinC (from a pYRG plasmid) was induced for 2 h by adding anhydrotetracycline (0.2 μg/mL) prior to imaging.” (lines 771–775 in the revised manuscript).

35. FtsZ-pdt is shown in Fig. 3G but it is not explained. Later in the manuscript it is mentioned that this version of FtsZ is degraded by mf-Lon proteinase with a reference, but this description is insufficient and should be explained in the text and figure legend.

Response: We agree with you and modified the text as follows: “These observations were further confirmed by the fact that in FtsZ^{ΔNIDR}, the interaction between K51 and P203 was strengthened by *in vivo* photocrosslinking (Fig. 5c, lanes 4 and 8; top bands indicated by a solid square) and tandem mass spectrometry (Fig. 5d and Supplementary Table 3). Of note, while wild-type FtsZ encoded by genomic DNA was essential for the proliferation of cells expressing FtsZ^{ΔNIDR}, its presence interfered with crosslinked product formation specific to FtsZ^{ΔNIDR} (see bands in lanes 2 and 6 in Fig. 5c). To circumvent this interference, we introduced a pdt-tag into the wild-type FtsZ protein. Inducible *mf*-Lon protease expression then degraded FtsZ-pdt⁴¹, effectively eliminating or reducing the interference (see bands in lanes 4 and 8 in Fig. 5c).” (285–294 in the revised manuscript).

We also modified the Fig. 5c legend as follows: “**c** Immunoblot showing photocrosslinked FtsZ^{K51Bpa} products in cells expressing FtsZ-pdt, which was degraded when proteinase *mf*-Lon was induced; molecular weight marker positions and FtsZ forms are indicated on the right and left, respectively.” (lines 1010–1013 in the revised manuscript).

36. Fig. 4, particularly panel D, shows clear evidence for increased FtsZ filament bundling of the delta N-IDR variant compared with WT or the D10F variant. And yet most of the rest of the paper argues that the N-IDR affects longitudinal interactions, not lateral interactions. This is puzzling. How might a polymer disassembly defect result in polymer bundling?

Response: This is an initial explanation of the results in Fig. 4d. Our subsequent explanations are based on the following.

Firstly, the protein concentration used for *in vitro* FtsZ assembly was 1 μM, a concentration that was insufficient for lateral FtsZ protein assembly, as observed for WT and D10F. Furthermore, our electron micrographs demonstrated that the ΔNIDR variant formed significantly longer protofilaments when compared with the WT protein.

Secondly, we previously indicated that longitudinal FtsZ assembly was essential and a prerequisite for lateral interactions, such that disrupting longitudinal interactions always led to a loss of lateral interactions (PMID: 29889022). Therefore, increased bundling observed with the ΔNIDR could be explained as a consequence of enhanced longitudinal interactions that promoted lateral interactions of protofilaments.

We modified the descriptions as follows: “Negative staining electron microscopy showed that protofilaments formed by FtsZ^{D10F} were slightly longer (Fig. 4d, middle panel), and those formed by FtsZ^{ΔNIDR} were even longer (Fig. 4d, right panel). Furthermore, FtsZ^{ΔNIDR} protofilaments formed bundles under these *in vitro* conditions (Fig. 4d, right panel), which possibly occurred due to enhanced longitudinal interactions which in turn would promote lateral protofilament interactions, consistent with previous observations²¹.” (lines 260–265 in the revised manuscript).

37. Lines 208-209: Comparing Fig 4D and F, it appears that expression of MinC reduced the amount of FtsZ in the pellets of both the WT and D10F mutant by half, whereas the ΔNIDR mutant is essentially unaffected. This suggests the D10F isn't necessarily less sensitive to MinC expression (compared to WT) so much as it instead starts at a higher degree of polymerization.

Response: We agree with you in principle, but we believe that this explanation applies to both mutants. We modified descriptions as follows to avoid confusion with our explanations described in our original manuscript: “Additionally, while showing similar response mechanisms to wild-type FtsZ, mutant FtsZ assemblies buffered against MinC-mediated disassembly, apparently due to their higher extent of assembly. This was indicated by the fact that the level of assembly was decreased for all these three forms upon the addition of MinC, but to a different degree (Figs. 4g and 4h).” (lines 268–272 in the revised manuscript).

38. Fig 5B: The figure legend indicates that this model is based on the 6UNX pdb structure. However, the 6UNX structure is E. coli FtsZ crystalized in the R (relaxed) conformation in which K51 and P203 are separated by ~ 17 Å, whereas this model appears to represent FtsZ in the T (tense) conformation. To my knowledge, there is no crystal structure solved with E. coli FtsZ in the tense conformation, and this can only be approximated by threading the E. coli FtsZ sequence through one of the S. aureus crystal structures in the tense conformation (eg, 3VOA). The authors need to clarify the origin of this protein structure.

Response: Thank you for this valuable insight. We obtained the tense (T) conformation of the *E. coli* FtsZ assembly by threading the *E. coli* FtsZ sequence through a tense conformation of the *S. aureus* crystal structure (PDB: 3VOA). We replaced the structure model with this approximated model (where residues K51 and P203 are separated by ~ 4 Å) (Fig. 5b).

39. Lines 222-225: It would be helpful to explain the purpose of the pdf tag fusion & expression of the lon protease in Fig 5C.

Response: See comment #35. We modified the description as follows: “Of note, while wild-type FtsZ encoded by genomic DNA was essential for the proliferation of cells expressing FtsZ^{ΔNIDR}, its presence interfered with crosslinked product formation specific to FtsZ^{ΔNIDR} (see bands in lanes 2 and 6 in Fig. 5c). To circumvent this interference, we introduced a pdt-tag into the wild-type FtsZ protein. Inducible *mf*-Lon protease expression then degraded FtsZ-pdt⁴¹, effectively eliminating or reducing the

interference (see bands in lanes 4 and 8 in Fig. 5c).” (lines 289–294 in the revised manuscript).

40. The P203S variant and its ability to restore the function of the delta N-IDR variant are very interesting and nicely support the hypothesis that the N-IDR is involved in disassembling FtsZ polymers. One prediction of the authors’ model is that the delta N-IDR P203S double variant should now have WT or nearly WT levels of GTPase activity. Was this tested?

Response: Thank you for posing this question. We tested this and the data are shown (Supplementary Fig. 5d) in the revised manuscript. We also added the following sentences to describe these data. “Intriguingly, GTPase activity levels in FtsZ^{ΔNIDR-P203S} remained comparable with FtsZ^{ΔNIDR}, rather than FtsZ^{WT} (Supplementary Fig. 5d). This was consistent with previous data showing that an FtsZ mutant, with approximately 10% of the GTPase activity of the wild-type protein, remained largely functional¹⁹. In light of this, the GTPase activity of the FtsZ protein requires further characterization.” (lines 310–314 in the revised manuscript).

41. Another prediction of the model is that FtsZ P203S alone (with the N-IDR intact) would be defective in assembly, or perhaps could disassemble better than WT FtsZ. Was this attempted? This would help to shed light on how P203S actually affects FtsZ assembly.

Response: We did not do this because we believed it would not provide much meaningful data. However, to address your concerns, we performed this study and added the following sentences to describe our data: “We further observed that the P203S mutation barely showed any detectable effects in terms of FtsZ functioning, neither in cell proliferation (Fig. 5e) nor in Z-ring formation (Supplementary Fig. 5c). This was in line with the fact that residue K51, that was in close contact with residue E3, had a weak contact with P203, and thus replacing it with Ser, would exert little effect on wild-type FtsZ assembly (Fig. 5c, lanes 2 and 4), but became unnecessarily strong in entities assembled by FtsZ^{ΔNIDR}.” (lines 304–309 in the revised manuscript).

42. Lines 249-253 / Fig 5I: The use of “non-interfering state” or “interfering state” to refer to the N-IDR in FtsZ monomers or polymers (respectively) needlessly complicates the explanation. Likewise, rephrasing “Event 1”, “Event 2”, etc with “polymerization”, “NIDR binding”, etc would also make this more clear.

Response: We agree that using “non-interfering state” or “interfering state” needlessly complicated the explanation, while “Event” did not help with the description. We modified these descriptions as follows: “In light of our observations, we propose an interaction-modulating mechanism to explain how the N-IDR acts as a *cis* element to facilitate effective FtsZ protofilament disassembly at cell poles, and Z-ring architecture condensation at midcell (schematically represented in Fig. 5j). Specifically, in the initial FtsZ protofilament that is likely in its T (tense) state^{23,30}, the N-IDR is in a disassembly-non-facilitating conformation. Following the formation of protofilament, the N-IDR switches to a disassembly-facilitating conformation and thus contacts the

shared surface (containing residue K51) and partially disrupts the longitudinal interface, transforming the FtsZ protofilament to a state conducive to disassembly, roughly corresponding to its R (relax) state^{23,30}.” (lines 366–374 in the revised manuscript).

43. Lines 263-272: This discussion would benefit from further consideration of the orientations of the tense and relaxed forms of FtsZ. For instance, most likely the K51 Bpa crosslinking with P203 occurs with the (GTP-bound) tense conformation of FtsZ Δ NIDR, as these residues are too far apart in the relaxed conformation to crosslink. Given as P203S restores the defect in FtsZ Δ NIDR, likely the NIDR is functionally active with tense form of FtsZ. It also seems unlikely that any conformational change as suggested in lines 267-270 (eg, to relaxed form of FtsZ with GTP hydrolysis) would improve FtsZ NIDR interactions, because the FtsZ NIDR will be more distant from residues it interacts with (particularly Q56 & F40) when filaments are in the relaxed conformation.

Response: Thank you for these insightful views and we agree with you in principle. We modified our descriptions as follows: “In light of these observations, we hypothesize that the N-IDR may additionally modulate FtsZ GTPase activity in promoting FtsZ protofilament disassembly. FtsZ protofilament formation, which is triggered by GTP binding⁴⁷, simultaneously allows the N-IDR to engage with the shared surface around residue K51. This in turn weakens interactions between the two surfaces respectively containing K51 and P203. As the surface containing P203 is located in the T7 loop critical for FtsZ GTPase activity⁴⁸, such a conformational readjustment may promote FtsZ GTPase activity. Subsequent GTP hydrolysis^{49,50} would generate a conformational change disrupting longitudinal interactions and making FtsZ protofilaments more susceptible to disassembly. In support of this hypothesis, we observed that N-IDR removal decreased GTPase activity (Figs. 4e and 4f) and also impaired FtsZ protofilament disassembly (Figs. 4a, 4g, and 4h). The fact that *FtsZ* ^{Δ NIDR-P203S} GTPase activity remained low but largely functional in supporting cell proliferation indicated that disrupting longitudinal interactions appeared to be more important than modulating the GTPase activity for N-IDR to act as a *cis* disassembly element.” (lines 384–398 in the revised manuscript).

44. Lines 278-281: This is an interesting idea, and there is prior evidence for the G105S (ftsZ84) allele having a disassembly defect (PMID: 11029443). However, I am pretty sure that FtsZ84 protein has not been shown to assemble in vitro (at any temperature), so this point may not be valid for that reason.

Response: Thank you for this comment. Although the FtsZ84 protein was not shown to assemble under standard *in vitro* conditions, it did form functional Z-rings in cells (PMID: 11029443 and PMID: 11854462). We freely admit that it is difficult to reproduce what occurs in living cells under *in vitro* conditions. Having said that, our explanation for FtsZ84 remains largely valid. Nevertheless, our description on that mutant is speculative and we decided to take your advice by removing the description: “Similarly, the contradictory observation that the FtsZ^{G105S} variant exhibited a dramatically reduced GTPase activity, but remained functional¹⁹, could be explained by

the fact that the substitution simultaneously disrupted the longitudinal interface where residue 105 was located.”

45. Line 284: Earlier comments about FtsN recruitment and cell wall biogenesis also apply here.

Response: Please refer to responses to comments #24 and #47.

46. Lines 285-287: this sentence is confusing; “far earlier before” does not make sense. The sentences following this also do not make sense.

Response: Please refer to responses to comments #24 and #47.

47. Lines 282-291: there is already considerable evidence for certain proteins, including FtsZ, as limiting factors for division. Too little FtsZ: no division. Too much: minicells. FtsZ:FtsA ratios are important too. So, it is inaccurate to say that division relying on quantitative measures is a new idea; same with FtsZ disassembly as being important (FtsZ84 with its low GTPase activity has defects in disassembly, similar to the Δ N-IDR mutants). The main novelty of the present work is that it identifies an interesting function of the extreme N terminus of FtsZ that was not previously appreciated, and that this function may be to compete with longitudinal interactions in order to tip the balance more towards polymer disassembly. I suggest removing this paragraph and replacing it with a brief summary of the main findings of the paper without the needless speculation.

Response: Thank you for this invaluable suggestion. We agree that our hypothesis was perhaps too speculative and off topic. To address this, we replaced the last paragraph as follows: “In summary, the extreme N terminus of FtsZ acts as a *cis* disassembly element, which disrupts longitudinal assembly by contacting a peripheral site and providing a stumbling block (forced interaction that does not generate an interaction force) near the interface, tipping the balance more toward polymer disassembly. This in turn enables Z-ring formation only at midcell (rather than elsewhere) upon modulation by certain *trans*-acting factors (e.g., the *E. coli* MinC protein).” (lines 416–421 in the revised manuscript).

48. Are there conserved features in the FtsZ N-IDR that might suggest this mechanism plays a role in other FtsZ proteins? A greater exploration of this across diverse bacterial species would increase the relevance and impact of these findings beyond just the *E. coli* cell division field.

Response: Thank you for this valuable suggestion. We analyzed the FtsZ N-IDR from *Bacillus subtilis* and *Staphylococcus aureus* and observed similar effects to *E. coli* FtsZ, (Supplementary Fig. 6). We added the following paragraph in the Results section to describe these data:

“The FtsZ N-IDR in other bacteria also functions as a *cis* disassembly element

We next examined whether the FtsZ N-IDR in other bacteria also acted similarly. To this end, we removed corresponding FtsZ sequences from *Bacillus subtilis* and *Staphylococcus aureus* (both are Gram positive bacteria that are distantly related to *E.*

coli and whose FtsZ has a low amino acids sequence identity with *E. coli* FtsZ (Supplementary Fig. 6a)) and assessed assembly properties under *in vitro* conditions. These deletions significantly increased assembly levels similar to *E. coli* FtsZ as shown by 90° angle light scattering (Supplementary Fig. 6b) and sedimentation analyses (Supplementary Fig. 6c and 6d).” (lines 339–347 in the revised manuscript).

Minor comments

1. Line 14: “has to be both steady and constantly changing”

Although the meaning will be clear to experts in the field, authors should more explicitly explain the meaning for the sake of other readers

Response: Thank you for your suggestion. We modified the description as follows: “Bacterial cell division hinges on the Z-ring, an architecture built from the dynamical assembly and disassembly of FtsZ proteins. This delicate balance ensures not only apparent stability, but also continuous remodeling, both of which are required for Z-ring functioning.” (lines 15–18 in the revised manuscript).

2. Lines 18-21: “a previously unobserved ...”

The text following the semicolon is a sentence fragment that lacks a subject; it needs to be rewritten.

Response: We are sorry for this grammatical issue. We modified the text as follows: “Here, by identifying all amino acid residues participating in FtsZ self-assembly in *Escherichia coli*, we show that the extreme N-terminal intrinsically disordered region (N-IDR) of FtsZ acts as a *cis* disassembly element that contacts and disrupts the longitudinal interface, tipping the balance more toward polymer disassembly. This previously unappreciated structural characteristic is indispensable for promoting Z-ring architecture condensation at midcell (rather than elsewhere) upon modulation by certain *trans*-acting factors (such as the *E. coli* MinC protein).” (lines 19–25 in the revised manuscript).

3. Line 27: Sentence missing a period & has unnecessary comma.

Response: We are sorry for this grammatical issue. The sentence has been modified as follows: “Cell division is a key process in both eukaryotes and prokaryotes. Prokaryotic cells divide via a binary fission process involving DNA replication and cytokinesis^{1,2}.” (lines 29–30 in the revised manuscript).

4. Line 87: delete “a”

Response: We are sorry for this grammatical issue. The sentence has been modified as follows: “To further clarify whether these interactions participated in longitudinal self-assembly, we performed photocrosslinking studies (under *in vitro* conditions) using purified Bpa variants of FtsZ at low concentrations, such that variants would assemble into single individual protofilaments²⁷.” (lines 106–109 in the revised manuscript).

5. Line 275: replace “resumption” with “rescue” or “restoration”

Response: Thank you for this advice. We replaced “resumption” with “restoration” in the text.

6. Figure 3F: A different label for the lower row of images would be helpful. The sideways ‘-’ looks like an ‘I’ here and can be confusing.

Response: Thank you for this advice. Fig. 3f has been modified.

7. Fig. 4H: y axis “proportion” is misspelled

Response: We are sorry for this issue. Fig. 4h has been corrected.

Reviewer #2 (Remarks to the Author):

This study explores the formation of the bacterial Z-ring through the self-assembly of the FtsZ protein in *Escherichia coli*. Using protein photocrosslinking, mass spectrometry, and high-resolution microscopy, the researchers identified the role of specific residues of the N-terminal intrinsically disordered region (N-IDR) in FtsZ longitudinal contacts in *E. coli*. Disrupting the N-IDR impairs cell division, underscoring its importance in Z-ring formation. IDR regions have been classically ignored because they cannot be crystallized, so this study follows a necessary new trend of further characterize their unexplored biological functions. Thus, this research article would be interesting in the field of bacterial cell division,

providing significant advancements in our knowledge on the FtsZ ring formation. That said, there are some issues that must be addressed before publication:
- In general, the text needs strong edition. While the experiments were explained in a logical order, and the story could be followed, some parts of the work were difficult to understand.

Response: We extensively revised our manuscript and clarified many unclear descriptions. We believe our manuscript is much more readable in the revised version.

For instance, which specific residues were eliminated in FtsZ Δ NIDR variant?

Response: We modified the description as follows: “To this end, we disrupted this interaction by removing the N-IDR (i.e., residues 1–10, but retained the first Met for protein expression purposes) and constructed an FtsZ ^{Δ NIDR} variant.” (lines 128–131 in the revised manuscript).

What is the difference between Supplementary Figures 1A and 1B? Is just a repetition restricting the experiment to a set of positions, or is experimentally different? Authors could include the explanation.

Response: We apologize for this confusion. We clarified this as follows: “For this, we verified photocrosslinking results involving the N-IDR and N-domain residues in the *ftsZ-Avi* strain (the FtsZ-Avi protein is expressed from genomic DNA)²¹, and analyzed crosslinked products using regular (rather than high-throughput) sodium dodecyl sulfate-polyacrylamide gel electrophoresis (SDS-PAGE).” (lines 82–85 in the revised manuscript).

- Bpa photocrosslinking has been shown to underperform when applied to disordered regions (Zhang et al., 2011, Nat Chem Biol). Could the authors comment on the motivation of the election of Bpa instead of other available options, for instance, DiZPK, also developed by the authors?

Response: We used DiZPK at initial study stages. However, we had to rely on our collaborator to obtain large DiZPK quantities, which went beyond synthesis capacity. As a result, we used Bpa, which is commercially available at affordable prices.

- To support the conclusions of in vivo photocrosslinking (specific amino acid interactions) the authors should validate these interactions with an independent approach, compatible with intrinsically disordered domains, like HDX, NMR, BLI, SPR... Additionally, Molecular Dynamics simulations could help to understand the movements and contacts of the N-IDR, and the data obtained can be used to plot frequency contact maps.

Response: Thank you for your insightful suggestions. Although these biophysical techniques are powerful for exploring relatively stable protein systems, it would be challenging to use them to study the N-IDR structure whose interactions occur in protofilaments during highly dynamic assembly-disassembly processes. We examined N-IDR interactions using cryo-electron microscopy; however, data from our collaborators failed to reconstruct a structure with sufficient resolution to see full N-

IDR engagement. Other than this, we are unsure what other *in vitro* physical methods can be used to examine such systems.

But, acting on your suggestion, we conducted molecular dynamic (MD) simulation studies. In particular, we performed MD simulations on a GTP-bound *E. coli* FtsZ dimer, focusing on interactions between the N-IDR and the longitudinal interaction surface. We admit that these simulations provided valuable data in supporting the N-IDR role in modulating filament assembly-disassembly processes. We added these data (Fig. 5i) and the following paragraph to describe these important observations.

“To learn more about N-IDR actions in FtsZ protofilaments, we performed molecular dynamic (MD) simulation studies. In particular, we performed five independent 150 ns MD simulations based on a FtsZ dimer to examine how N-IDR contacted its neighboring subunit. The 150 ns mean smallest distance frequency plot (Fig. 5i, top row) and average structures from the final 50 ns for each simulation (Fig. 5i, middle row) revealed the following. 1) Starting from an initial extended conformation, the N-IDR transitioned to a folded state that contacted various surfaces on the monomer itself and on a neighboring FtsZ subunit. While exhibiting consistent trends, each simulation displayed distinct N-IDR conformations and contacting locations. This suggested that the N-IDR was somehow flexible and existed in multiple possible states. 2) The N-IDR interacted with several residues within the N-domain in a neighboring FtsZ subunit. Notably, residue pair contacts between the N-IDR and N-domain were clearly observed in simulation analysis (Fig. 5i, top row), which agreed with our *in vivo* photocrosslinking analysis and mass spectrometry data (summarized in Fig. 1f).

Further mean smallest distance analysis between the N-domain of one FtsZ subunit and the N-IDR or T7 loop of a neighboring subunit during the final 50 ns of each simulation revealed the following contact pattern relationships. The contact between the N-IDR and N-domain appeared to be inversely related to the interaction between the N-domain and the T7 loop. This was indicated by Sim1 and Sim3 data (Fig. 5i, bottom row): while contact between the N-IDR and the N-domain was weak and the interaction between the N-domain and T7 loop was strong in Sim1, but the opposite was observed in Sim3. This observation agreed with *in vivo* crosslinking results (Fig. 5c).” (lines 315–336 in the revised manuscript).

The legend for Fig. 5i: “i Five heat maps depicting the mean smallest distance between the N-IDR of one FtsZ subunit and the N-domain of its neighbouring subunit, each representing an independent 150 ns molecular dynamics simulation (top row). Five average conformations of the FtsZ dimer structure based on the last 50 ns of each simulation were displayed (middle row). Five heat maps depicting the mean smallest distance between the N-IDR (residues 1-10) or the T7 loop (residues 198-211) of one FtsZ subunit, and the N-domain (residues 42-54) of its neighbouring subunit during the final 50 ns of each simulation were displayed (bottom row).” (lines 1021–1028 in the revised manuscript).

- Was the experiment in Fig.1G performed with the other Bpa positions indicated to participate in longitudinal interactions: residues 10, 20, 24, 56, etc?

In my opinion, this experiment was very clever, and it would be nice to see if all those amino acids behave in the same manner as 8 and 51.

Response: Thank you for this constructive comment. We agree it would have been ideal to examine them all. The problem was that these are very time-consuming and expensive experiments, as we would have had to purify a large amounts of Bpa-containing FtsZ proteins.

Line 94 - The authors claim that "We strongly emphasize that longitudinal self-assembly interactions mediated by the N-IDR and N-domain were revealed here for the first time, and were almost completely missed in previous studies (21–24), evidently due to N-IDR invisibility in resolved crystal structures."

However, they are missing some previous studies that support the role of the N-IDR in protofilament formation: Corrales-Guerrero et al., 2018 (Front Microbiol) already showed that the deletion of N-IDR in multicellular cyanobacteria leads to straight filaments that are more stable by ultracentrifugation and present a reduced GTPase activity. Also, Silber et al., 2020 (mBio) showed that the N-IDR is more protected from ClpP degradation when *Bacillus subtilis* FtsZ is forming a protofilament, and proposed that the reason might be the interaction within subunits. Fujita et al, 2023 (Nat Comms) resolved *Klebsiella pneumoniae* FtsZ by cryo-EM, showing that the N-IDR region participates both in longitudinal and lateral interactions. The authors should change the mentions to "the first time" or "previously unobserved" across the text.

Response: Here, we emphasized that our observations were for the “first time” and “previously unobserved”, in the sense that the direct interaction between N-IDR and N-domain in living cells was unveiled for the first time.

Despite this, we used more moderate words and modified the descriptions as follows:

“This previously unappreciated structural characteristic is indispensable for promoting Z-ring architecture condensation at midcell (rather than elsewhere) upon modulation by certain *trans*-acting factors (such as the *E. coli* MinC protein).” (lines 22–25 in the revised manuscript).

“Of note, interactions involving the N-IDR were largely missed in previous studies^{28–31}. Furthermore, we noted that the N-IDR interacted with residues (e.g., F40, Q56, and S62) peripheral to those previously defined at the longitudinal interface³⁰. Therefore, this posed a series of questions; did this interaction have a role in the longitudinal assembly of FtsZ, Z-ring formation, and cell division?” (lines 121–125 in the revised manuscript).

- Related with the previous comment: To broaden the conclusions, the authors could test if the hypothesis of the competing interactions is applicable to other bacteria.

Response: Reviewer #1 also raised this issue (see comment #48). We performed additional experiments and analyzed the FtsZ N-IDR from *Bacillus subtilis* and *Staphylococcus aureus* and observed similar effects as for *E. coli* FtsZ (Supplementary

Fig. 6). We added the following paragraph to the Results section to describe these data:
“The FtsZ N-IDR in other bacteria also functions as a cis disassembly element

We next examined whether the FtsZ *N-IDR* in other bacteria also acted similarly. To this end, we removed corresponding FtsZ sequences from *Bacillus subtilis* and *Staphylococcus aureus* (both are Gram positive bacteria that are distantly related to *E. coli* and whose FtsZ has a low amino acids sequence identity with *E. coli* FtsZ (Supplementary Fig. 6a)) and assessed assembly properties under *in vitro* conditions. These deletions significantly increased assembly levels similar to *E. coli* FtsZ as shown by 90° angle light scattering (Supplementary Fig. 6b) and sedimentation analyses (Supplementary Fig. 6c and 6d).” (lines 339–347 in the revised manuscript).

Line 131 - In the text is written: "To this end, we adopted a screening approach by replacing residues in the FtsZ N-IDR with each of the other 19 natural amino acids", but only mutations from D10 are shown in Supplementary Figure 4. Were the mutations in other positions not producing cell growth defects? The differences in phenotype between dNIDR and D10F shown in Fig. 2D would point to other residues having a role too. I don't think is necessary to make replacements by every single amino acid, but if it is done already, it would be helpful to show it and discuss it in the text.

Response: Reviewer #1 also raised this issue (comment #23).

We actually selected residues T8 and D10 for mutational analysis, given that they showed the most pronounced photocrosslinking bands (Fig. 1c and Supplementary Fig. 1a). However, when T8 was replaced with the other 19 amino acids, none appeared to interfere with LY928- Δ *ftsZ* strain proliferation. The functional complementation results from these variants were added to Supplementary Fig. 4a. The following sentences were also provided.

“These two residues were selected for mutational analysis as they showed the most pronounced photocrosslinking bands (Fig. 1c and Supplementary Fig. 1a). Interestingly, when T8 was replaced by the other 19 amino acids, none appeared to interfere with LY928- Δ *ftsZ* strain proliferation. However, among the 19 D10 variants, approximately half showed significant defects in cell proliferation (Fig. 2a, right panel for FtsZ^{D10F} and Supplementary Fig. 4a for the rest), and three examined variants (FtsZ^{D10F}, FtsZ^{D10Q}, and FtsZ^{D10Y}) showed Z-ring architecture defects (Supplementary Fig. 4b).” (lines 168–174 in the revised manuscript).

Based on what we observed later in our study, these results may be explained as follows. T8 variant results suggested that N-IDR function did not rely on highly specific interactions (Supplementary Fig. 4a). D10 variant results were likely due to its location at a site where certain replacements disrupted proper N-IDR positioning. Of note, functional complementation assays only showed dramatic defects in cell proliferation, and thus the failure to see T8 variant defects (as well as certain D10 variants) could not completely rule out that some of them may have caused certain Z-ring formation defects.

Line 208 - The authors claim that "the polymers formed by FtsZ variants exhibited more resistance to MinC protein depolymerization effects". If I

understood correctly, they expect that this is an indirect effect derived from the fact that the polymers are stabilized in the variants FtsZ Δ NIDR and FtsZD10F, and MinC binds mainly to Helix 8 and 10, which are only available in the last subunit of the polymer. However, Machado et al., 2024 (biorxiv) NMR experiments on *Bacillus subtilis* FtsZ show that, apart from residues already described in the C-terminal region, FtsZ residues L11, A12 and A26 presented significant differences in the chemical shifts (Figures 3C,E from the preprint) in the presence of MinC.

For that reason, the results shown in Figures 3F,G and 4G,H of this work could be explained by both direct and indirect facts, or the addition of both. The authors must clarify if the N-IDR region interacts with MinC in their conditions in order to properly explain the results.

Response: Reviewer #1 also raised a similar issue relating to the interaction between MinC and FtsZ protofilaments (comment #37). We modified the description as follows: “Additionally, while showing similar response mechanisms to wild-type FtsZ, mutant FtsZ assemblies buffered against MinC-mediated disassembly, apparently due to their higher extent of assembly. This was indicated by the fact that the level of assembly was decreased for all these three forms upon the addition of MinC, but to a different degree (Figs. 4g and 4h).” (lines 268–272 in the revised manuscript).

We did not generate any data hinting at a direct interaction between MinC and N-IDR based on *in vivo* photocrosslinking analyses when Bpa was introduced into this region (Fig. 1c). In view of this, the NMR data suggest that the chemical shifts of residues L11, A12, and A26 may have resulted from indirect effects.

- The authors should demonstrate that the interaction between S203 and K51 is debilitated again in FtsZ Δ NIDR P203S variant in comparison to FtsZ Δ NIDR.

Response: This was one interaction that should have been examined, but the problem was how. The most feasible approach would have been by *in vivo* photocrosslinking analysis, but the expected result would be a weakened band, which is a negative result in nature. Furthermore, this experiment was barely feasible as we would have had to degrade the FtsZ-pdt protein in different cell types to the exact same degree. Thus, it would have been difficult to generate convincing results to address the reviewer’s concerns.

- Figure 5G - What is the effect (by ultracentrifugation) of P203S mutation in the wild type background? The authors need to show that the effect of the mutation is directly related to the interaction with K51 and not only with being in the polymerization surface. The combination of D10F P203S would also shed light into the role of the N-IDR.

Response: Reviewer #1 also raised a related issue (comment #41). To address your concerns, we generated a P203S mutation in the wild-type background and showed our results in Fig. 5e and Supplementary Fig. 5c. We added the following sentences to describe these data: “We further observed that the P203S mutation barely showed any

detectable effects in terms of FtsZ functioning, neither in cell proliferation (Fig. 5e) nor in Z-ring formation (Supplementary Fig. 5c). This was in line with the fact that residue K51, that was in close contact with residue E3, had a weak contact with P203, and thus replacing it with Ser, would exert little effect on wild-type FtsZ assembly (Fig. 5c, lanes 2 and 4), but became unnecessarily strong in entities assembled by FtsZ^{ΔNIDR}.” (lines 304–309 in the revised manuscript).

We agree that the combination of D10F and P203S could shed light on N-IDR roles, but we felt it was not an absolutely essential experiment, given our observation that the P203S mutation alone showed barely any detectable effects on FtsZ functioning.

- Figure 5I - If the hypothesis is correct, titration of a peptide containing N-IDR to full-length FtsZ (and to FtsZ Δ NIDR) could destabilize filaments, and promote disassembly. This could be assayed by RALS or sedimentation assays. This is only a suggestion, but could reinforce the model.

Response: This is an interesting point. But these interactions do not appear to be very specific based on our mass spectrometry analysis (Fig. 1f) and molecular dynamic simulations (Fig. 5i). Consequently, it might be difficult to generate significant effects if such titration assays were performed.

- It has been demonstrated that FtsZ suffers a conformational change, which is related to the monomeric/polymeric states (Matsui et al., 2014, J Biol Chem; Wagstaff et al., 2017, MBio). This conformational change affects the N-terminal domain, so it could also affect N-IDR movements and contacts. In fact, Fujita et al, 2023 (Nat Comms, already mentioned) analyzed FtsZ conformational changes by cryo-EM, and observed the N-IDR region ordered as an alpha-helix in one of the conformations, participating both in longitudinal and lateral interactions. The authors could comment on that in the discussion.

Response: Thank you for these suggestions.

We cited Matsui et al., 2014 (Ref. 48) in the following description: “This in turn weakens interactions between the two surfaces respectively containing K51 and P203. As the surface containing P203 is located in the T7 loop critical for FtsZ GTPase activity⁴⁸, such a conformational readjustment may promote FtsZ GTPase activity.” (lines 387–390 in the revised manuscript).

We cited Wagstaff et al., 2017 (Ref. 23) in the following description: “In light of our observations, we propose an interaction-modulating mechanism to explain how the N-IDR acts as a *cis* element to facilitate effective FtsZ protofilament disassembly at cell poles, and Z-ring architecture condensation at midcell (schematically represented in Fig. 5j). Specifically, in the initial FtsZ protofilament that is likely in its T (tense) state^{23,30}, the N-IDR is in a disassembly-non-facilitating conformation. Following the formation of protofilament, the N-IDR switches to a disassembly-facilitating conformation and thus contacts the shared surface (containing residue K51) and partially disrupts the longitudinal interface, transforming the FtsZ

protofilament to a state conducive to disassembly, roughly corresponding to its R (relax) state^{23,30}.” (lines 366–374 in the revised manuscript).

We cited Fujita et al, 2023 (Ref. 43) in the following description: “...cryo-EM studies indicated that in protofilaments formed by FtsZ bound to a specific monobody, the N-IDR existed as an α -helix that partially contacted the monobody as well as the N-domain of the neighboring subunit⁴³.” (lines 360–362 in the revised manuscript).

Minor:

- Fig.5A: Please change color of MinC in the scheme so it is not the same as the one for FtsZ (weakened interaction).

Response: Thank you for this suggestion; the color has been changed.

- Line 605 - Methods: I understand that FtsZ concentration was 15 micromolar, not millimolar.

Response: We apologize for this typo and have corrected it to “15 μ M”.

- Why FtsZ Bpa variants were expressed/induced using two different manners? If there is a reason why variants 2-230 couldn't be induced with tetracycline like the others, this should be stated in Materials and Methods.

Response: We apologize for this confusion. We added the following sentences to clarify this: “Another pTac plasmid containing the P_{T-23105} promoter was used to generate FtsZ variants where residues 231–383 were individually replaced by Bpa, such that variant protein expression only occurred in the presence of TetR and anhydrotetracycline. This stringent control strategy prevented the production of potentially toxic truncated FtsZ forms (which might be produced when the introduced TAG codon was read as a stop codon rather than one encoding Bpa) that would cause lethal phenotypes before inducer addition.

The P_{T-23105} promoter was modified from the BBa_J23105 promoter (selected from the Anderson promoter collection (parts.igem.org/Promoters/Catalog/E.coli/Constitutive)). The middle sequence

between -10 and -35 sequences (bold) in the P₂₃₁₀₅ promoter was then replaced with the *tetO* operator sequence (underlined) and the λ t1 transcriptional terminator sequence (italics) was added upstream of the P₂₃₁₀₅ promoter. This allowed the promoter
(**CGCAAAAACCCCGCTTCGGCGGGGTTTTTCGCTAAGGGATTTTGGTTTTA**
CTCCCTATCAGTGATAGATACTAT) to function constitutively and expresses FtsZ (or variants) at levels comparable with endogenous FtsZ in a strain lacking TetR (a transcription repressor binding the *tetO* sequence), but function in a regulatable manner in a strain expressing TetR (from pYR5C-tetR plasmid).” (lines 612–628 in the revised manuscript).

Reviewer #3 (Remarks to the Author):

In this study by Yin, et al., the authors show using photocrosslinking and tandem mass spec. that residues (T8 and D10) in the FtsZ N-terminal IDR crosslink to residues in the polymerization domain. To investigate this further, they characterize two mutants, D10F and deltaIDR for in vivo Z-ring assembly and in vitro for GTP hydrolysis and polymerization. Both mutants, D10F and dIDR form filament-like structures in a cell and recruit other division proteins, including FtsA and FtsN, but fail to coalesce into condensed rings. Interestingly, the FtsZ mutant proteins appear less dynamic in vivo and in vitro for GTP hydrolysis, and they have a higher tendency to bundle. The results are very interesting, but the authors make several claims that seem to be over-interpreted, or could be verified through additional analysis. For example, many FtsZ mutants with modifications to the filament interface have slow GTP hydrolysis rates and show similar structures in vivo (i.e., Q47K or D212C in Redick, et al., 2005). While this does suggest that N-terminal residues may engage the protofilament interface, the supposition that the N-terminus is intrinsically a destabilizing cis element seems premature and not well supported.

Response: We designated the N-IDR as a *cis* disassembly element based mainly on our observation that the FtsZ ^{Δ NIDR} mutant was defective due to unnecessarily strong assembly (Fig. 4), and that the FtsZ ^{Δ NIDR-P203S} double mutant restored FtsZ function by decreasing assembly levels to that of the FtsZ^{WT} (Fig. 5). This was an unexpected observation as when we observed the abnormal entities formed by these mutants by live cell imaging (Fig. 3), we intuitively (but mistakenly) considered that they were due to a partial disruption of FtsZ subunit assembly.

In view of this, the results reported by Redick, et al., 2005 shared certain similarities with our data, namely, what they observed was due to an enhanced rather than weakened assembly (although Redick, et al., designed mutations to disrupt the interface). In particular, the FtsZ Q47K mutant, where the N-IDR interaction was very likely disrupted, exhibited a defective phenotype which was highly similar to our mutant proteins.

In our revised manuscript, upon your suggestion and that of the other reviewers, we added more experimental data to support our hypothesis that the N-IDR acted as a *cis* disassembly element. These included the following. Firstly, we showed that FtsZ N-IDR's from other bacterial species appeared to have the same roles (Supplementary Fig. 6). Secondly, in molecular simulation dynamics analysis, the N-IDR showed a conformation potential to act as such a *cis* disassembly element (Fig. 5i). Thirdly, the GTPase activity of the FtsZ^{ΔNIDR-P203} double mutant, a functional revertant of FtsZ^{ΔNIDR}, remained as low as FtsZ^{ΔNIDR} (Supplementary Fig. 5d), further indicating that the N-IDR functioned to destabilize the protofilament.

Moreover, the analysis of MinC failing to disrupt FtsZ-dIDR polymers could also be due to poor recognition by MinC, which may result from the mutation itself, or from the slower dynamics.

Response: The fact that our data did not hint at a direct interaction between MinC and N-IDR, based on *in vivo* photocrosslinking when Bpa was introduced to this region (Fig. 1c), does not support the possibility that the failure of MinC to disrupt FtsZ^{ΔNIDR} polymers was due to poor FtsZ recognition by MinC. Rather, it was due to a slower dynamicity of assembled protofilaments as a result of an absent N-IDR as a disassembly element.

Finally, it is not clear that the IDR itself plays a role, but that residues 8 and 10 may be important irrespective of the rest of the IDR. In fact, in the predicted structure of FtsZ (via AlphaFold), residue 10 is practically part of the polymerization domain and not the IDR.

Response: We provided additional experimental and simulation data showing that the N-IDR did not appear to function as a highly specific interaction element, but rather a stumbling block.

In particular, in addition to D10 (which tolerated replacement by approximately five other residues, including A, E, G, H, and M, Supplementary Fig. 4a), we also replaced T8, which is another residue that showed the most pronounced photocrosslinking bands (Fig. 1c and Supplementary Fig. 1a), with the other 19 amino acids, but none appeared to interfere with LY928- Δ *ftsZ* strain proliferation. This was indicated in functional complementation results from these variants (newly added Supplementary Fig. 4a). This is described in the following newly added sentences: “These two residues were selected for mutational analysis as they showed the most pronounced photocrosslinking bands (Fig. 1c and Supplementary Fig. 1a). Interestingly, when T8 was replaced by the other 19 amino acids, none appeared to interfere with LY928- Δ *ftsZ* strain proliferation. However, among the 19 D10 variants, approximately half showed significant defects in cell proliferation (Fig. 2a, right panel for FtsZ^{D10F} and Supplementary Fig. 4a for the rest), and three examined variants (FtsZ^{D10F}, FtsZ^{D10Q}, and FtsZ^{D10Y}) showed Z-ring architecture defects (Supplementary Fig. 4b).” (lines 168–174 in the revised manuscript).

Furthermore, our MD simulation studies (newly added Fig. 5i) suggested that when

starting from an initial extended conformation, the N-IDR transitioned to a folded state that contacted various surfaces on the monomer itself, and on a neighboring FtsZ subunit. While exhibiting consistent trends, each simulation displayed distinct N-IDR conformations (including residue 10) and contacting locations. This suggested that the N-IDR was somehow flexible and existed in multiple possible states. This is described in the following newly added sentences: “To learn more about N-IDR actions in FtsZ protofilaments, we performed molecular dynamic (MD) simulation studies. In particular, we performed five independent 150 ns MD simulations based on a FtsZ dimer to examine how N-IDR contacted its neighboring subunit. The 150 ns mean smallest distance frequency plot (Fig. 5i, top row) and average structures from the final 50 ns for each simulation (Fig. 5i, middle row) revealed the following. 1) Starting from an initial extended conformation, the N-IDR transitioned to a folded state that contacted various surfaces on the monomer itself and on a neighboring FtsZ subunit. While exhibiting consistent trends, each simulation displayed distinct N-IDR conformations and contacting locations. This suggested that the N-IDR was somehow flexible and existed in multiple possible states. 2) The N-IDR interacted with several residues within the N-domain in a neighboring FtsZ subunit. Notably, residue pair contacts between the N-IDR and N-domain were clearly observed in simulation analysis (Fig. 5i, top row), which agreed with our *in vivo* photocrosslinking analysis and mass spectrometry data (summarized in Fig. 1f).

Further mean smallest distance analysis between the N-domain of one FtsZ subunit and the N-IDR or T7 loop of a neighboring subunit during the final 50 ns of each simulation revealed the following contact pattern relationships. The contact between the N-IDR and N-domain appeared to be inversely related to the interaction between the N-domain and the T7 loop. This was indicated by Sim1 and Sim3 data (Fig. 5i, bottom row): while contact between the N-IDR and the N-domain was weak and the interaction between the N-domain and T7 loop was strong in Sim1, but the opposite was observed in Sim3. This observation agreed with *in vivo* crosslinking results (Fig. 5c).” (lines 315–336 in the revised manuscript).

In addition to these comments, the authors are asked to address the following:

1. How conserved are the N-terminal 1-9 residues?

Response: N-IDR sequences are not highly conserved as indicated by the following description to the revised manuscript: “Further sequence alignment analysis of FtsZ proteins from different bacteria showed that they all possessed N-IDR regions, with various lengths and low amino acid sequence similarities²⁶.” (lines 78–80 in the revised manuscript).

This is also indicated by newly added data (Supplementary Fig. 6a). See below.

a

	N-IDR
Escherichia coli	-MFEPMELTNDAVIKVIGVGGGGNAVEHMMVRERIEGVEFFAVNTDAQALRKTA
Bacillus subtilis	MLEFETNIDGLASIKVIGVGGGGNAVNRMINEVQGVEYIAVNTDAQALNLSK
Staphylococcus aureus	MLEFEQGFNHLATLKVIGVGGGGNAVNRMIDHGMNNVEFIAINTDGQALNLSK

3. Lines 40-42 – This is an oversimplification of FtsZ behavior, because while polymers assemble with GTP, polymers have been observed to contain GDP after hydrolysis by prior to disassembly (i.e., curved polymers). Also, which variant is referenced here, as many variants are described in the literature with altered hydrolysis activities.

Response: Thank you for pointing this out. We revised the descriptions as follows: “*In vitro* studies have suggested that FtsZ assembly occurs upon guanosine triphosphate (GTP) binding, and disassembly is facilitated by GTP hydrolysis^{11,17,18}. However, the importance of GTPase activity for FtsZ functioning remains unclear. For example, the FtsZ84 variant (FtsZ^{G105S}), which has approximately 10% of the GTPase activity of wild-type FtsZ and significant defects in FtsZ treadmilling, and also slower recovery in fluorescence-recovery-after-photobleaching (FRAP), was shown to support cell division¹⁹.” (lines 46–52 in the revised manuscript).”

2. Lines 78-84. What is the distance in angstroms spanned by the N-terminal IDR assuming an extended random coil? What is the IDR residue cutoff? Residues 10, and also 8, would be really close to the polymerization domain and not really part of the IDR. Could the D10F substitution artificially create a more stable polymer by stabilizing the interface?

Response: The distance spanned by the N-IDR (residues 1–10) as an extended random coil was approximately 30 Å, which allowed it to effectively contact the N-domain of neighboring subunits as indicated by our MD simulation analysis (Fig. 5i).

We consider residue 10 as part of the N-IDR because its removal (FtsZ^{ΔNIDR}) significantly strengthened rather than weakened protofilaments (Fig. 4).

Although we cannot absolutely exclude the possibility that the D10F substitution artificially created a more stable polymer by stabilizing the interface, it would be difficult to understand that replacing D10 by 13 other residues, all showing similar phenotypic defects (Supplementary Fig. 4a), could create more stable polymers by stabilizing the interface. More importantly, the N-IDR was mainly defined on the properties of FtsZ^{ΔNIDR} in which residue 1-10 was removed.

3. Fig. 1G – Where does A87 map on the structure in 1F?

Response: We indicated residue A87 position in the structure in Fig. 1f (shown below) and added a description: “Residue A87 (its position is shown in Fig. 1f) was assumed to participate in lateral interactions based on the fact that it was located between R85 and R89, both of which were reported to mediate lateral interactions with FtsZ protofilaments²¹.” (lines 115–118 in the revised manuscript).

4. If GDP is used *in vitro* instead of GTP as in Fig. 1G, are dimer interactions also disrupted. If not, then the GDP interface is different from the GTP interface.

Response: This would be an interesting experiment. When GDP is added instead of GTP to our *in vitro* assembly system, we expect that crosslinking patterns (including dimers) will largely remain the same. In our model, the difference at the interface, when GTP or GDP is bound, would not cause large crosslinking pattern changes, namely, the GTP interface does not differ from the GDP interface in a large degree.

5. Lines 78-93. “Single individual protofilaments (line 88)”. This should be corroborated with TEM images. However, Fig. 4D clearly shows that minimally FtsZ-dIDR bundles.

Response: We apologize for this confusion. Indeed, *in vitro* crosslinking studies were performed using Bpa variants of FtsZ^{WT} which existed as single individual protofilaments by TEM (see the left panel in Fig. 4d). Of note, both were examined using approximately the same protein concentration (~1 μ M). To clarify this, we modified the legend in Fig. 1g as follows:

“**g Immunoblotting results (left) show photocrosslinked products formed with indicated purified Bpa variants (~1 μ M), probed with polyclonal antibodies against FtsZ. Of note, FtsZ^{WT} assembled into single individual protofilaments at 1 μ M (as indicated by TEM analysis, left panel Fig. 4d);” (lines 940–944 in the revised manuscript).**

6. Lines 99-104. It is not clear how many residues were removed from deltaN-IDR (nine or 10?). It is assumed that an initiator methionine was inserted, resulting in the sequence ‘M-DAVIKV..’

Also, why are only these two mutants D10F and dIDR characterized? What about T8 substitutions or shorter N-terminal deletions? Does mutation of residue 9 show any phenotypic aberrations? What about the phenotypes and *in vitro* characterization of K51, S62, F40, and Q56? These are important mutations to analyze.

Response: We apologize for this confusion. We actually deleted residues 1–10 but retained the first residue Met for protein expression purposes. We modified this description as follows: “To this end, we disrupted this interaction by removing the N-IDR (i.e., residues 1–10, but retained the first Met for protein expression purposes) and constructed an FtsZ^{ΔNIDR} variant.” (lines 128–131 in the revised manuscript).

We tried to generate shorter N-IDR mutants. However, mutant protein expression levels were significantly reduced, with levels far lower than wild-type levels, which prevented us performing further functional complementation analyses.

We also analyzed T8, the results of which are now added to Supplementary Fig. 4a and described as follows: “These two residues were selected for mutational analysis as they showed the most pronounced photocrosslinking bands (Fig. 1c and Supplementary Fig. 1a). Interestingly, when T8 was replaced by the other 19 amino acids, none appeared to interfere with LY928-Δ*ftsZ* strain proliferation. However, among the 19 D10 variants, approximately half showed significant defects in cell proliferation (Fig. 2a, right panel for FtsZ^{D10F} and Supplementary Fig. 4a for the rest), and three examined variants (FtsZ^{D10F}, FtsZ^{D10Q}, and FtsZ^{D10Y}) showed Z-ring architecture defects (Supplementary Fig. 4b).” (lines 168–174 in the revised manuscript).

We did not perform similar amino acid replacement studies on residue 9 due to its low crosslinking product levels. However, based on our observations on T8 variants, only certain variants may exhibit phenotypic aberrations.

We did not mutate residues in the N-domain, including K51, S62, F40, and Q56 residues because such replacements may have caused conformational alterations or even misfolding. This would not have generated clearcut results.

7. Line 100. Does the N-domain here refer to the polymerization domain? Again, residue numbers in the text would help. It is counterintuitive that the N-terminal IDR is not in the N-terminal domain.

Response: To the best of our knowledge, the N-domain has been conventionally defined as covering residues 11–176 (Fig. 1b), being only part of the polymerization domain. This N-domain does not include the N-IDR sequence.

To clarify this, we modified the descriptions as follows: “Remarkably, residue positions showing crosslinking products were located across the whole polypeptide chain, not only in the previously defined N-domain (residues 11–176), C-domain (residues 201–316)^{23,24} and the C-terminal sequence (residues 317–383), but also in the extreme N-terminus (residues 1–10), which was previously under-characterized (Fig. 1b).” (lines 66–70 in the revised manuscript).

8. Line 105-110. This result implies that the IDR is important for division- the suggestion that this defect tracks with the IDR-N-domain interaction is an overinterpretation.

Response: We agree with your comment and modified this description as follows: “These observations implied that the N-IDR plays an indispensable role for FtsZ functioning.” (lines 142–143 in the revised manuscript).

9. Line 112 – Please clarify “entities”.

Response: We apologize for this confusion. We replaced “entities” with “Z-ring” in line 112 and added a simple definition of entities as subcellular assemblies. See below.

“To achieve this, we first sought to directly visualize the Z-ring formed by the FtsZ^{ΔNIDR} variant in live-cell imaging ... Live-cell imaging revealed that the FtsZ^{ΔNIDR} variant generated multiple highly distorted fluorescent FtsZ entities (subcellular assemblies) dispersed throughout the cell (Fig. 2b, middle panel) ...” (lines 145 and 153 in the revised manuscript).

10. Line 184 – What is FtsZ^{WT}29. Is this a reference?

Response: Yes, 29 is a reference. We apologize for this typo and have corrected it.

“To this end, we examined whether the entities assembled by FtsZ variants could be effectively disassembled upon MinC overexpression in cells, similar to FtsZ^{WT} 36.” (lines 225–227 in the revised manuscript)

11. The experiments with MinC overexpression are difficult to interpret. Failure of MinC to disrupt dimers could simply be due to reduced MinC activity/binding, such as if the N-terminal residues of FtsZ overlap the MinC N-domain or C-domain binding sites. This should be tested directly in vitro.

Response: As already outlined, we generated no data that hinted at a direct interaction between MinC and N-IDR, based on *in vivo* photocrosslinking analysis when Bpa was introduced to this region (Fig. 1c). We performed *in vitro* studies to test MinC on FtsZ mutant assembly.

We modified our descriptions as follows to clarify this: “Additionally, while showing similar response mechanisms to wild-type FtsZ, mutant FtsZ assemblies buffered against MinC-mediated disassembly, apparently due to their higher extent of assembly. This was indicated by the fact that the level of assembly was decreased for all these three forms upon the addition of MinC, but to a different degree (Figs. 4g and 4h).” (lines 268–272 in the revised manuscript).

12. If the model for MinC function is for the N-domain to bind to an open polymer interface to prevent additional FtsZ subunits from attaching, then an FtsZ protein with reduced dynamics should be more resistant to destabilization by MinC. FtsZ(G105S) is known to have slower dynamics because it has a mutation near the GTP binding site, is it also more resistant to MinC than wild type, similar to FtsZ-dIDR?

Response: As far as we know, two models exist that explain MinC effects, one of which is what you refer to. Another model proposes that MinC/MinD fragments FtsZ filaments. Our observations are explained by both models. Regarding the N-IDR role, we describe this in the discussion section of the newly revised manuscript. “In summary, the extreme N terminus of FtsZ acts as a *cis* disassembly element, which disrupts longitudinal assembly by contacting a peripheral site and providing a stumbling block

(forced interaction that does not generate an interaction force) near the interface, tipping the balance more toward polymer disassembly. This in turn enables Z-ring formation only at midcell (rather than elsewhere) upon modulation by certain *trans*-acting factors (e.g., the *E. coli* MinC protein).” (lines 416–421 in the revised manuscript).

It follows that reduced dynamics do not necessarily result in MinC resistance. What is important is the K_{eq} of the assembly process (Fig. 3e). Reduced dynamics only indicate a decrease in K_{off} , but if the K_{on} is also reduced to a similar degree, then the K_{eq} will be around normal levels, allowing FtsZ to be functional. For the FtsZ (G105S) mutant, both K_{off} and K_{on} are reduced, maintaining the K_{eq} in the normal range. However, for FtsZ^{ΔNIDR}, the K_{off} is reduced, but the K_{on} is not reduced leading to an increased K_{eq} . As a result, unlike FtsZ^{ΔNIDR}, the FtsZ (G105S) mutant should not have increased resistance to MinC, thus it forms a normal Z-ring.

This explains why the FtsZ (G105S) mutant could not assemble under *in vitro* conditions as Reviewer #1 noted (comment #44): “However, I am pretty sure that FtsZ84 protein has not been shown to assemble *in vitro* (at any temperature).”

13. Line 190-194. This is an overstatement.

Response: To address this, we replaced our assertive claims with more measured statements: “Collectively, these *in vivo* data appeared to indicate that the N-IDR had important roles endowing Z-ring architecture with a proper high-level dynamicity, and also an appropriate extent of assembly, allowing MinC to disassemble FtsZ protofilaments at sites beyond midcell.” (lines 246–249 in the revised manuscript).

14. Line 204. Light scatter is non-quantitative and does not distinguish between a few bundles versus longer filaments versus many short filaments. Caution should be taken in comparing and interpreting difference in amplitudes of scatter. The TEM data favors that FtsZ-dIDR has a higher tendency to bundle laterally, and this is what may cause the increase in scatter.

Response: We agree that light scattering is a qualitative assay, but as an initial rough examination, it helped us investigate any differences in FtsZ variant assembly. We modified the description as follows: “Firstly, FtsZ variants exhibited significant assembly differences when roughly monitored by 90° angle light scattering analyses (Fig. 4a).” (lines 254–256 in the revised manuscript).

The increased bundling by FtsZ^{ΔNIDR} could be explained due to enhanced longitudinal interactions that promoted lateral interactions of the protofilaments. This is because longitudinal FtsZ assembly is essential and a prerequisite for lateral interactions (PMID: 29889022).

15. Line 207. Clarify or restate “maintenance times”. This language is strange.

Response: Thank you for pointing this out. The sentence has been modified as follows: “Secondly, variants with an altered N-IDR showed prolonged assembly duration times (Fig. 4a) and reduced GTPase activity (Figs. 4e and 4f), indicating a defect in FtsZ disassembly.” (lines 266–268 in the revised manuscript).

16. Fig. 4B – Why is there such a low percentage of wt FtsZ in the GTP pellet (also in 5F)?

Response: We performed this to examine the differences between wild-type and mutants, and was done using a protein concentration that marginally allowed protein assembly to occur.

17. Fig. 4C – Proportion is misspelled.

Response: We apologize for this typo. Figure 4C has been corrected.

18. Fig. 5A. Proportion is misspelled. Architecturing is insufficiently descriptive. Please correct/clarify.

Response: We apologize for this typo. Figure 5A has been corrected (see above).

We replaced "architecturing" with "assembling" throughout the manuscript (including the title) to ensure clarity and accuracy.

19. A recent BioRxIV paper further suggests that ZapA, which bundles FtsZ filaments, also engages the FtsZ N-terminal region (Fujita, et al., BioRxIV, 2024). This and the idea that other FtsZ-interacting proteins could engage the N-terminus should be discussed further.

Response: We appreciate this new information and added it to our discussion: “Interestingly, a recent cryo-EM study revealed that ZapA, a protein that binds FtsZ straight protofilaments, sequestered the N-IDR away from the longitudinal interaction surface of FtsZ⁵¹. This *in vitro* observation suggested that the N-IDR may act as a target for proteins like ZapA to stabilize FtsZ protofilaments. It follows that when the N-IDR is released from ZapA, FtsZ protofilament disassembly is facilitated. Furthermore, a recent study showed that during natural interbacterial antagonism, *Yersinia pseudotuberculosis* secreted a protein effector, CccR, which covalently modified residue D8 with an adenosine monophosphate (AMP) group within the FtsZ N-IDR in *Enterobacteriaceae*, inducing cell filamentation⁵². Together, these observations implied that the N-IDR acted as an effective and general target for bacteria to modulate their cell proliferation or that of other species.” (lines 405–415 in the revised manuscript).

Reviewer #4 (Remarks to the Author):

FtsZ is an exceptionally well-studied and essential bacterial cell division protein. In vitro FtsZ assembles head-to-tail to form single stranded polymers that interact laterally to form larger bundled structures. To date research has largely overlooked the short, intrinsically disordered N-terminal domain. Using previously a developed high throughput technique to evaluate the interactions of individual protein residues, Yin and Liu et al. present data supporting a role for the N-terminal IDR as a modulator of the longitudinal FtsZ-FtsZ assembly interface. Using UV crosslinking and mass spectrometry they identify residues on FtsZ’s longitudinal interface that appear to interact with the N-terminal IDR. In vivo mutational analysis and microscopy, as well as in vitro biochemical assays further support the author’s assertion that the identified residues play an important role in FtsZ assembly, although it is not clear that this is limited exclusively to the longitudinal interface. Genetic experiments indicating that a P203 substitution that weakens its interaction with residue K51) reduces the self-assembly of FtsZ Δ NIDR were particularly appreciated. Finally, the authors utilize MinC and Lon protease to suggest that the N-terminal IDR plays a role in weakening the longitudinal interface. Combined these data suggest a mechanistically detailed role for the N-terminal IDR in facilitating Z ring localization and disassembly.

While the mass spec experiments support a role for the N-terminal IDR as a mediator of longitudinal interactions between FtsZ polymers, data indicating a critical role for this domain in FtsZ assembly and function in vivo is less compelling lessening the overall strength of the manuscript.

Major comments

Western blot methodology: were any steps taken to ensure sample uniformity between strains, such as total protein stains of the blot or normalizing samples by optical density/protein concentration? If so, please include this in the methodology.

Response: Thank you for pointing this out. We added the following paragraph to

the Methods section. “For high-throughput SDS-PAGE, loaded protein samples were not normalized. Cells were cultured in four 96-well plates (due to large sample numbers and low protein quantities). For regular SDS-PAGE (except those shown in Fig.1c and Supplementary Fig. 1b), protein samples were normalized by measuring the optical density of the cells.” (lines 664–667 in the revised manuscript).

577-578: The methodology for the FRAP experiments is highly unusual. “To reduce interference by FtsZ treadmilling, we used overnight-cultured cells with relatively stable Z-ring architectures.” How many hours specifically were cells grown? How long were cells in stationary phase? What percentage of cells had Z rings? We would expect few cells to be forming Z rings and dividing in stationary phase cultures. This calls into question how well these experiments can represent active division.

Response: We apologize for this confusion. The term “overnight-cultured” was improperly used because the cells were cultured for 10 h. Thus, we modified this as follows: “To reduce interference by FtsZ treadmilling, we used cells that were cultured for 10 h and had reached late logarithmic phase, a timepoint where most wild-type cells formed well-maintained Z-ring architectures.” (lines 808–810 in the revised manuscript).

Authors only mention using bright field in the context of Sup figure 3A, while other figures clearly include brightfield images as well. More care should be taken to specify which experiments included brightfield and when these images were taken during the experiment both in the methodology and figure legends. To this end, it’s unclear if brightfield images were taken before and after fluorescence imaging, as is often standard. This is especially important for frap experiments, as the photobleaching is a harsh cellular perturbation.

Response: We apologize for not clearly describing these procedures. We added descriptions to the Methods section and appropriate figure legends as follows.

In the Methods: “Brightfield microscope images were taken before fluorescence imaging to record cell shapes.” (lines 793–794 in the revised manuscript).

In the Fig. 2 legend: “c Timelapse live-cell fluorescence micrographs merged with brightfield micrographs showing cells expressing indicated FtsZ forms (each fused to mNeonGreen) which were taken every 10 min and show three-dimensional (3D) FtsZ entities at correlated time points...”. (lines 952–955 in the revised manuscript).

“Overall cell shapes were drawn according to brightfield micrographs and are indicated by dotted white lines (b, e, f, g).” (lines 963–964 in the revised manuscript).

In the Fig. 3 legend: “c Live-cell images show the directional motion of indicated FtsZ entities, and kymographs show entities in two cells recorded every 2 s (total = 100 s). Tracked FtsZ condensates are indicated by white arrows. Overall cell shapes were drawn according to brightfield micrographs and are indicated by dotted white lines; scale bar = 0.5 μm .” (lines 971–974 in the revised manuscript).

In the Fig. 5 legend: “**h** Timelapse images (brightfield, fluorescence, and merged) showing *ftsZ-ΔNIDR-P203S-mneongreen* cells; scale bar = 2 μm.” (lines 1019–1021 in the revised manuscript).

In the Supplementary Fig. 3 legend: “**a** Brightfield micrographs showing residual cells expressing FtsZ^{ΔNIDR}; the red arrow indicates dead cell debris; scale bar = 5 μm.”

In the Supplementary Fig. 4 legend: “**b** Live-cell imaging micrographs (merged brightfield and fluorescence images) showing cells expressing indicated D10 variants of FtsZ (each fused to mNeonGreen); scale bar = 1 μm.”

In the Supplementary Fig. 5 legend: “**c** Live-cell imaging micrographs (merged brightfield and fluorescence images) showing cells expressing the indicated forms of FtsZ (each fused to mNeonGreen); scale bar = 2 μm.”

4A. ΔNIDR appears to assemble far higher than expected for something that isn't bundling. Additionally, the TEMs in 4D may show some bundling as well. Taken together, this casts some doubt on the validity of 1G, where we can no longer be sure that dimers, trimers, etc. are only the result of lateral interactions. This is especially a concern because the “low” concentration of FtsZ used in 1G is not detailed.

Response: We apologize for this confusion. Increased bundling with FtsZ^{ΔNIDR} could be explained as a consequence of enhanced longitudinal interactions that promoted lateral interactions of protofilaments. This is because longitudinal FtsZ assembly is essential and a prerequisite for lateral interactions (PMID: 29889022).

Indeed, *in vitro* crosslinking studies (Fig. 1g) were performed using Bpa variants of FtsZ^{WT} which existed as single individual protofilaments by TEM (the left panel in Fig. 4d). Of note, both were examined using approximately the same protein concentration (~1 μM). To clarify this, we modified the Fig. 1g legend as follows: “**g** Immunoblotting results (left) show photocrosslinked products formed with indicated purified Bpa variants (~1 μM), probed with polyclonal antibodies against FtsZ. Of note, FtsZ^{WT} assembled into single individual protofilaments at 1 μM (as indicated by TEM analysis, left panel Fig. 4d) ...” (lines 940–944 in the revised manuscript).

Line 605: 15mM seems like an unusually high concentration, particularly compared to the other assays. Is this a typo? The typical concentration for in vitro assays of FtsZ is 5μM. Additionally, 5-minute time points is very infrequent- WT is already disassembling by the end of the first 10 minutes.

Response: We apologize for this typo which has been corrected to “15 μM”. We used a concentration of 15 μM rather than 5 μM for our analysis because at 5 μM, FtsZ^{ΔNIDR} assembly would have lasted for at least 220 min, which was too long for manual recording.

Due to technical constraints, we were unable to monitor filament formation at intervals shorter than 5 min. We also had to work with three samples at the same time and perform measurements one by one on a single piece of equipment. Our setup required removing samples from a temperature-controlled incubator for light scattering measurements. Such operations limited our ability to perform more frequent recordings

on assembly dynamics. While we regret that we did not record initial filament formation stages in a more precise manner, our data demonstrated distinct differences between samples.

It would benefit the authors to cite Silber et al. 2020, as their findings may corroborate the assertion that the N-terminal IDR plays a role in FtsZ disassembly. Corrales-Guerrero 2018 also implicates the N-terminal IDR in FtsZ polymerization in cyanobacteria. Additionally, Vaughan et al. 2006 serves as a valuable reference for the conservation of the N-terminal IDR in FtsZ across species.

Response: Thank you for these valuable suggestions. We cited Silber et al. as reference #44, Corrales-Guerrero et al. as reference #42, and Vaughan et al. as reference #26 in Discussion and Results sections.

“It is noteworthy that our revelation showing the N-IDR role in FtsZ assembly was supported by previous observations. For example, it was shown that the FtsZ N-IDR deletion in cyanobacteria reduced GTPase activity and the formation of more stable protofilaments⁴²; while cryo-EM studies indicated that in protofilaments formed by FtsZ bound to a specific monobody, the N-IDR existed as an α -helix that partially contacted the monobody as well as the N-domain of the neighboring subunit⁴³. Furthermore, it was shown that the N-IDR was more protected from ClpP protease degradation when the *B. subtilis* FtsZ formed a protofilament, hinting at possible participation in certain interactions⁴⁴.” (lines 357–365 in the revised manuscript).

“Further sequence alignment analysis of FtsZ proteins from different bacteria showed that they all possessed N-IDR regions, with various lengths and low amino acid sequence similarities²⁶.” (lines 78–80 in the revised manuscript).

Other comments:

Generally, architecture is not a verb although it is used as such throughout this manuscript.

Assemble, organize, orchestrate, are more precise terms.

Response: Thank you for this suggestion. We replaced all instances of "architecturing" with "assembling" throughout the manuscript, including the title, to ensure clarity and accuracy.

Line 33/34 more extensive citations are recommended.

Response: Thank you for your suggestion. We supplemented the following statement with more references: “Super-resolution fluorescence microscopy studies have shown that Z-ring architecture exhibits high heterogeneity in cells^{4,6-8}.” (lines 36–37 in the revised manuscript).

References:

“6. Strauss, M. P. *et al.* 3D-SIM super resolution microscopy reveals a bead-like

arrangement for FtsZ and the division machinery: implications for triggering cytokinesis. *PLoS Biol.* **10**, e1001389 (2012).

7. Buss, J. *et al.* *In vivo* organization of the FtsZ-ring by ZapA and ZapB revealed by quantitative super-resolution microscopy. *Mol. Microbiol.* **89**, 1099–1120 (2013).
8. Rowlett, V. W. & Margolin, W. 3D-SIM Super-resolution of FtsZ and its membrane tethers in *Escherichia coli* cells. *Biophys. J.* **107**, L17–L20 (2014).” (lines 434–440 in the revised manuscript).

Line 33 fluorescence microscopy (not fluorescent)

Response: Thank you for pointing this out. We made the correction as follows: “Super-resolution fluorescence microscopy studies have shown that Z-ring architecture exhibits high heterogeneity in cells^{4,6-8}.” (lines 36–37 in the revised manuscript).

Line 156: This sentence is grammatically confusing. Do the authors mean that the two mutants can support normal divisome assembly and division? I am confused by the word “architecture” here.

Response: We apologize for this confusion; we modified the descriptions as follows: “Surprisingly, we observed that these defective Z-ring architectures, that were mainly assembled by FtsZ^{D10F} and FtsZ^{ΔNIDR}, still recruited FtsA (Fig. 2e) and also FtsN (Fig. 2f). FtsA is a key membrane anchoring protein for FtsZ^{33,34}, and FtsN is the last essential protein recruited to the divisome^{5,35}.” (lines 199–202 in the revised manuscript).

Figure 1C. the right-side blot is numbered differently on the top than the bottom- appears to be a typo?

Response: We apologize for this confusion. The top numbers indicate residue positions. We modified Fig. 1c as follows.

Sup 1C. It would be interesting to see what these other proteins that certain residues appear to be crosslinking.

Response: Indeed. We identified proteins that crosslinked to some of these residue positions and identified new proteins (not related to cell division). Since we have not yet characterized these proteins, they are not described in the manuscript.

2A could use a little more detail in the interpretation of the results, or at least more explanation of the experimental set up in the figure legend. The experiment itself does make sense on examination though.

Response: Thank you for pointing this out. We modified both the figure and the legend as follows: “**a** Complementation results for FtsZ^{ΔNIDR} and FtsZ^{D10F} supporting *ΔftsZ* strain growth (contains the rescue plasmid pJSB100) where FtsZ^{ΔNIDR}, FtsZ^{D10F} or FtsZ^{WT} (positive control) were constitutively expressed from a pTac plasmid.” (lines 947–950 in the revised manuscript).

3F. Fluorescence is difficult to see, please include images of individual channels, not just the merges.

Response: Thank you for this suggestion. We decided to only show fluorescence images as recommended by Reviewer#1. The modified figure is shown here.

Line 156: This sentence is written in a confusing way. Please simplify.

Response: Thank you for pointing this out. We modified this as follows: “Surprisingly, we observed that these defective Z-ring architectures, that were mainly assembled by FtsZ^{D10F} and FtsZ^{ΔNIDR}, still recruited FtsA (Fig. 2e) and also FtsN (Fig. 2f). FtsA is a key membrane anchoring protein for FtsZ^{33,34}, and FtsN is the last essential protein recruited to the divisome^{5,35}.” (lines 199–202 in the revised manuscript).

Line 590: The “kymograph analysis” method section might be better named “SIM-TIRF microscopy and kymograph analysis”

Response: Thank you for your suggestion. We modified this subtitle as follows: “SIM-TIRF microscopy and kymograph analysis”. (line 822).

2E-F. To make the figures more colorblind accessible, we recommend using green and magenta instead of green and red.

Response: Thank you for this suggestion. The colors in Figs. 2e–g have been altered as follows.

Protein purification methods do not specify if a dialysis step occurred after eluting off the Hi-trap column.

Response: We apologize for this and have modified the statement as follows: “Before use, proteins were desalted using 5 mL Hi-Trap desalting columns (Cytiva) equilibrated with polymerization buffer.” (lines 736–737 in the revised manuscript).

Sedimentation methodology lists 3 concentrations for assays, but doesn’t label which experiment uses which concentration.

Response: We apologize for this confusion. We listed these concentrations in the figure legends of particular experiments and modified the methods section and figure legends as follows: “For EcFtsZ, the purified protein was diluted to the indicated concentrations in polymerization buffer in a final reaction volume of 400 μ L.” (lines 844–845 in the revised manuscript).

Fig. 4 legend: “b Coomassie brilliant blue staining shows the sedimentation analysis of polymerized FtsZ forms, with a starting monomeric concentration of 1 μ M and 2 mM of GTP; T, total; S, supernatant; P, pellet.” (lines 988–990 in the revised manuscript).

“g Sedimentation analysis of indicated FtsZ forms with a starting monomeric concentration of 5 μ M in the presence of 5 μ M MinC.” (lines 996–998 in the revised manuscript).

Fig. 5 legend: “f Coomassie brilliant blue staining shows sedimentation fractions containing polymerized FtsZ forms with a starting monomeric concentration of 3 μ M and 2 mM of GTP; T, total; S, supernatant; P, pellet.” (lines 1015–1018 in the revised manuscript).

4A and E. It is not immediately clear if there are replicates of this data. Please add clarifying language.

Response: These data represent one set of experiments with no replicates. We modified descriptions in the legend as follows:

“a Representative light scattering curves show time-dependent polymerization processes for indicated FtsZ forms, each with a starting monomeric concentration of 15 μ M, initiated by 1 mM GTP, and monitored by 90° angle light scattering.” (lines 985–988 in the revised manuscript).

“e These curves reflect time-dependent relative Pi concentrations released from polymerization samples in (a).” (lines 993–994 in the revised manuscript).

5H. Fluorescence is difficult to see, please include images of individual channels, not just the merges.

Response: Thank you for this suggestion. We added images of all channels as shown below.

Line 290: While I agree that ring condensation isn't necessarily a prerequisite for divisome recruitment, this seems like a dramatic flattening of a process we still don't fully understand.

Response: We appreciate these positive comments. Reviewer 1 raised similar concerns. We replaced these descriptions with the following more focused summary: “In summary, the extreme N terminus of FtsZ acts as a *cis* disassembly element, which disrupts longitudinal assembly by contacting a peripheral site and providing a stumbling block (forced interaction that does not generate an interaction force) near the interface, tipping the balance more toward polymer disassembly. This in turn

enables Z-ring formation only at midcell (rather than elsewhere) upon modulation by certain *trans*-acting factors (e.g., the *E. coli* MinC protein).” (lines 416–421 in the revised manuscript).

Reviewer #5 (Remarks to the Author):

Email: chanzy@pku.edu.cn

Point-to-point response for manuscript NCOMMS-24-40635A

REVIEWER COMMENTS

Reviewer #1:

I have thoroughly reviewed the revised manuscript, the comments posed by the other reviewers, and the responses of the authors to all the comments. Overall I feel that the authors have sufficiently addressed the concerns posed. The revised manuscript is now much stronger than the version originally submitted and should be accepted for publication.

I do have a few minor courtesy comments for the authors to consider before publication. I do not believe these require any further response from the authors:

1. Supplementary Fig 1a: residue “2” is labeled but is not colored according to the five categories added in revision.

Response: Thank you for raising this point. We labeled it this way to indicate that we performed Bpa substitutions starting at residue 2. To avoid confusion, we removed this labeling from Supplementary Fig. 1a (see below).

a

2. Lines 358-360: as written it appears that FtsZ N-IDR deletion in cyanobacteria reduced the formation of more stable protofilaments, which is the opposite as reported in ref 42. This sentence should be modified as follows (or equivalent) “For example, it was shown that the FtsZ N-IDR deletion in cyanobacteria reduced GTPase activity and ****promoted**** the formation of more stable protofilaments”

Response: Thank you for pointing out this error. We corrected this sentence (in the Discussion part) as follows: “*For example, it was shown that the FtsZ N-IDR*

deletion in cyanobacteria reduced GTPase activity and promoted the formation of more stable protofilaments;” (lines 366–368 in the revised manuscript).

3. Line 360: “while” suggests the following point will contrast with the previous, but instead it builds upon it. I suggest replacing it with “additionally,”

Response: Thank you for pointing this out. We replaced the word “while” with “additionally”. We modified this sentence (in the Discussion part) as follows: “*additionally, cryo-EM studies indicated that in protofilaments formed by FtsZ bound to a specific monobody, the N-IDR existed as an α -helix that partially contacted the monobody as well as the N-domain of the neighbouring subunit.*” (lines 368–370 in the revised manuscript).

Reviewer #2:

The revised version of the manuscript shows a significant improvement. The authors have addressed some of the reviewer's concerns, providing a better characterization of the IDR role.

However, I am concerned about the MD experiments depicted in Figure 5:

The MD analysis indeed fits the mass spectrometry results, and suggests the competition between the N-terminal domain and the T7-loop for the N-IDR region. That said, the authors claimed that the “N-IDR transitioned to a folded state” and indicated that the N-IDR was flexible and mobile during the dynamics to finally interact with different residues. However, I am missing some data supporting these statements (perhaps in supplementary material) about the behavior of such regions during the full dynamics (for instance parameters such as RMSF, RG, RMSD..) including secondary structure analysis, and also the contact maps of the full protein from which Fig.5i was extracted.

Response: Thank you for bringing this to our attention. The required adjustments have been made as per your recommendations

Upon performing molecular simulation analysis, we noted that the N-IDR finally reached a relatively stable extended state, which might be insufficient to be viewed as a folded state. We thus modified the statement (in the Results part) as follows: “*1) Starting from an initial extended conformation, the N-IDR transitioned to a state that contacted various surfaces on the monomer itself and on a neighbouring FtsZ subunit.*” (lines 327–329 in the revised manuscript)

We included RMSF, RG, and RMSD data in the revised manuscript (newly added Supplementary Fig. 6) and also shown here for your reference.

And added the following descriptions (in the Results part): *“To assess the adequacy of simulation timescales, we analyzed root mean square deviation (RMSD) and radius of gyration (Rg) values of the N-IDR over five independent 150 ns trajectories. As shown (Supplementary Fig. 6), both metrics plateaued after ~100 ns with fluctuations (standard deviation) within 0.03 nm (RMSD) and 0.02 nm (Rg), indicating that conformational equilibrium was reached within the simulation timescale. Consistently elevated values in both root mean square fluctuation (RMSF) profiles and Rg distributions across all replicates (except Sim4) indicated that the N-IDR retained high structural plasticity, and preferentially adopted extended conformations (Supplementary Fig. 6).”* (lines 317–325 in the revised manuscript); and the following

figure legend (in the Supplementary Information): **“Supplementary Figure 6. Molecular dynamics simulations of the N-terminal intrinsically disordered region (N-IDR), N-domain, and T7-loop in the FtsZ dimer interface. Five independent 150 ns molecular dynamics simulations (Sim1–Sim5) were conducted at 300 K using GROMACS with the following parameters: protein FtsZ, AMBER ff99SB-ILDN force field; ligand GTP, AMBER parameters; solvent, SPC/E water model; counterions, sodium ions for system neutralization. Trajectory analysis was performed using frames sampled at 10 ps intervals. Figure panels show: (Left) Temporal evolution of root-mean-square deviation (RMSD) from initial structures, quantifying global conformational stability; (Middle) Per-residue root-mean-square fluctuation (RMSF) mapping local flexibility across the FtsZ dimer interface; (Right) Radius of gyration (Rg) trajectories tracking overall compactness changes. Color coding is consistent throughout: N-IDR (blue), T7-loop (orange), and N-domain (green).”**

We considered it unnecessary to include these secondary structure evolution data in the manuscript due to the following reasons: (1) the N-IDR remained largely extended without forming such a regular secondary structure as alpha-helix or beta-strands and (2) the T7 loop and N-domain preserved their original secondary structure composition with minimal fluctuations. Secondary structure evolution data are shown below for your reference.

As you requested, we included full protein contact maps (shown below for your reference) from which Fig.5i was extracted in submitted Source Data.

Initial
Mean smallest distance

Sim1
Mean smallest distance

Sim2
Mean smallest distance

Sim3
Mean smallest distance

Sim4
Mean smallest distance

Sim5
Mean smallest distance

Also, according to Materials and Methods the first set of contact maps was calculated for the full dynamics (150ns) but the second only for the last 50 ns? I would assume that is always better to analyze only the last 50 ns in all cases, after initial stabilization of the system. And, if it was calculated using the full 150 ns how was separated for the initial situation, and how was the latter analyzed?

Response: We sincerely appreciate this valuable suggestion. We replaced contact maps (top panel in Fig. 5i) by showing the results of the final 50 ns of the simulation (rather than the whole 150 ns). We freely admit that these 50 ns maps demonstrate stronger contact frequencies between the N-IDR and N-domain than those in 150 ns maps, thereby providing a more accurate representation of the state. A modified Fig. 5i is shown below for your reference.

We modified the sentence (in the Results part) as follows: *“The final 50 ns mean smallest distance frequency plot (Fig. 5i, top row) and average structures from the final 50 ns for each simulation (Fig. 5i, middle row) revealed the following.”* (lines 325–327 in the revised manuscript)

In our molecular dynamics (MD) simulations, the initial state refers to the system conformation upon adding water molecules and ions, but prior to reaching energy minimization and equilibrium.

We modified the Fig. 5i legend accordingly: *“Five heat maps depicting the mean smallest distance between the N-IDR of one FtsZ subunit and the N-domain of its neighbouring subunit, each representing the last 50 ns from an independent 150 ns molecular dynamics simulation (top row). Five average conformations of the FtsZ*

dimer structure based on the last 50 ns of each simulation are displayed (middle row). Five heat maps depicting the mean smallest distance between the N-IDR (residues 1–10) or the T7 loop (residues 198–211) of one FtsZ subunit, and the N-domain (residues 42–54) of its neighbouring subunit during the last 50 ns of each simulation are displayed (bottom row). For comparison, the initial conformation (prior to energy minimization and subsequent steps) is also presented.” (lines 1066–1074 in the revised manuscript)

I have another question regarding the initial structure, which was obtained from a model using *S. aureus* 3VOA crystal structure. I understand that they selected this structure because it is in the open conformation. However, 3VOA was crystallized in a GDP-bound form. While this may not change the result of the MD, I would like to see an independent MD simulation using a model created after an *E. coli* structure. 6UNX corresponds to *E. coli* L178E-GTP bound, it is in the open conformation, and shows a similar conformation to 3VOA, however there are subtle differences in the subunit interface. This structure could be used as a model for the *E. coli* WT protein.

Response: We would like to clarify this as follows. The FtsZ subunit in 6UNX was actually in the closed or R conformation, rather than the open or T conformation, as the authors described (PMID: 32039891): “*structures were determined of the E. coli FtsZ mutant FtsZ (L178E) with GDP and GTP bound to 1.35 and 1.40 Å resolution, respectively. The E. coli FtsZ(L178E) structures both crystallized as straight filaments with subunits in the R conformation...*”.

There are obvious differences at the longitudinal interface between 6UNX and 3VOA, particularly in terms of contacts between the N-domain and the T7-loop (structural alignments are provided below for your reference).

Furthermore, when we used the *E. coli* FtsZ(L178E) structure in the R conformation (6UNX) for preliminary simulation tests, the N-IDR barely contacted the N-domain in the simulation, only contacting a small surface and with low frequency. Additionally, as emphasized by Reviewer #1, the FtsZ structure in the T conformation (3VOA) was more relevant to our photocrosslinking and mass spectrometry results for FtsZ-K51Bpa. Therefore, we used FtsZ in the T conformation (3VOA) for our simulations.

We sincerely appreciate your comments, which led us to carefully reexamine our MD simulation workflow and input files. This thorough review revealed an important oversight in our previous simulations: GTP molecules were missing hydrogen atoms and had incorrect charge assignments in ITP files.

To address this, we corrected GTP parameters by properly protonating molecules and assigning accurate charges, then performed five new independent simulations with a revised setup. We thus revised protocol descriptions (in the Methods part) as follows: *“Topology and force field parameters for the GTP molecule were generated using the ACPYPE tool⁶⁶. The FtsZ dimer was positioned at the center of a rectangular periodic boundary box filled with water molecules. Sodium ions were added to neutralize negative charges in the system. Simulation system setup is summarized (Supplementary Table 6). FtsZ protein and sodium ions were modeled using the AMBER ff99SB-ILDN force field⁶⁷, whereas GTP parameters were derived from the standard AMBER force field⁶⁸, and water molecules were defined using the SPC/E model⁶⁹. All residues maintained their canonical protonation states at pH 7.0, as parameterized in the AMBER ff99SB-ILDN force field. No explicit protonation state adjustments were required for the simulated system.*

The simulation methodology followed conventional practices for all-atom MD of protein-nucleotide systems. AMBER ff99SB-ILDN force field and AMBER parameters were adopted for their balance of accuracy and computational efficiency in modeling folded and disordered protein regions. The SPC/E water model was selected for its compatibility with the AMBER force field and its realistic dielectric properties. While model limitations (e.g., fixed-charge approximations) are acknowledged, this setup enabled the probing of ms-scale conformational changes that were inaccessible to

polarizable models. These choices provided sufficient accuracy to resolve residue-level interactions and conformational changes.

The system was energy-minimized using the steepest descent algorithm (50,000 steps) with a convergence threshold of 1000 kJ mol⁻¹ nm⁻¹ for the maximum force. A 0.1 ns equilibration at 300 K was performed under the NVT ensemble with positional restraints applied to the protein. Temperature coupling was achieved using the V-rescale thermostat algorithm. The system was further equilibrated for 0.1 ns under the NPT ensemble (1 bar, 300 K) using the Parrinello-Rahman barostat method, with positional restraints maintained. The system underwent 150 ns of unrestrained MD simulation under constant temperature and pressure (NPT) conditions, with a 2 fs integration interval. Hydrogen bonds were constrained using the LINCS algorithm⁷⁰, and non-bonded interactions were computed using a 1.0 nm cutoff for both Coulombic and van der Waals terms, with long-range electrostatics handled using the Particle-Mesh Ewald method⁷¹. Trajectories were saved every 10 ps for analysis. Verlet cutoff scheme and periodic boundary conditions were applied throughout. Randomly assigned velocities were initialized from a Maxwell–Boltzmann distribution at 300 K.” (lines 900–932 in the revised manuscript). All related data and figures were updated accordingly, including Fig. 5i and Supplementary Fig. 6 (as well as the secondary structure evolution and the full contact map shown above).

Importantly, while these corrections improved the technical accuracy of our simulations, they did not alter our original conclusions. The new results consistently showed that the N-IDR preferentially localized near the FtsZ dimer interface and maintained dynamic contacts with the N-domain, confirming our initial finding that the N-IDR has a strong tendency to interact with the N-domain in T-state FtsZ dimers. We modified our descriptions (in the Results part) accordingly as follows: *“Simulation results indicated that, in the initial state, the N-IDR and N-domain had no direct contact, while the T7 loop maintained close direct contact with the N-domain. However, in the last 50 ns of simulations, when the system approached stabilization, the N-IDR contacted the N-domain to varying degrees and at different sites, whereas contacts between the T7 loop and N-domain weakened to different extents, particularly for Sim1 and Sim3.”* (lines 338–343 in the revised manuscript).

We are grateful for your insightful comments, which not only helped us identify and fix this technical issue, but also provided an opportunity to further validate the robustness of our conclusions. The agreement between the original and corrected results strengthens our conclusion regarding N-IDR behavior.

Also, regarding new results shown in Supplementary Figure 5d, the authors could consider if the low GTPase activity presented by FtsZ dNIDR-P203S could be due to

low polymerization capacity (compared to WT) caused by the mutation of P203S. In my opinion, the experiments *in vivo* performed by the authors shed some light to the function of P203, but they could be complemented by *in vitro* assays, like TEM or sedimentation, to discern if the polymerization capacity is impaired. Low impairment could not be discarded even when the growth of the *E. coli* strain is not compromised.

Response: We agree with you and performed additional *in vitro* sedimentation assays to compare FtsZ^{P203S} polymerization efficiency with that of wild-type FtsZ. Our results demonstrated that FtsZ^{P203S} exhibited comparable sedimentation patterns to FtsZ^{WT} (Supplementary Figs. 5d and 5e, see below). These data indicated that the P203S substitution did not significantly impair the protein's intrinsic polymerization capability *in vitro*.

We modified the descriptions (in the Results part) as follows: *“Of note, we also observed that the P203S substitution exhibited only marginal effects on FtsZ^{WT} properties, under either in vivo (Fig. 5e and Supplementary Fig. 5c) or in vitro (Supplementary Figs. 5d and 5e) conditions. This was in line with the fact that residue K51, that was in close contact with residue E3, had a weak contact with P203, and thus replacing it with Ser, would exert little effect on wild-type FtsZ assembly (Fig. 5c, lanes 2 and 4), but became unnecessarily strong in entities assembled by FtsZ^{ANIDR}.”* (lines 303–308 in the revised manuscript); and added the figure legend accordingly (in the Supplementary Fig.5): *“d Coomassie brilliant blue staining revealed the sedimentation profile of FtsZ polymers formed from 3 μM monomeric protein in the presence of 2 mM GTP (T: total lysate; S: supernatant; P: pellet). e Quantification of FtsZ polymerization (Fig. 5d) shows no statistically significant differences in sedimented fractions (n = 3 biological replicates; data represented as the mean ± SEM; Student's t-test, n.s. = no significance).”*

d

e

In addition, I have a few minor comments:

For Light scattering experiments of BsFtsZ and SaFtsZ: In Materials and methods is written 10 μM but 15 μM in the legend figure.

Response: We apologize for this error. The actual concentration was 15 μM . We have corrected this (in the Methods part). “*The purified form of particular E. coli FtsZ (EcFtsZ)/BsFtsZ/SaFtsZ was added at a final concentration of 15 μM in respective polymerization buffers, to a fluorometer cuvette with a 1 cm path length.*” (lines 851–854 in the revised manuscript)

Reviewer #3:

In this study by Yin, et al., the authors show that residues (T8 and D10) in the FtsZ N-terminal IDR crosslink to residues in the polymerization domain. To investigate this further, they characterize two mutants, D10F and deltaIDR for in vivo Z-ring assembly and in vitro for GTP hydrolysis and polymerization. Both mutants, D10F and dIDR form filament-like structures in a cell and recruit other division proteins, including FtsA and FtsN, but fail to coalesce into condensed rings. Interestingly, the FtsZ mutant proteins appear less dynamic in vivo and in vitro for GTP hydrolysis, and they have a higher tendency to bundle. The results are very interesting and suggest that N-terminal residues engage the protofilament interface and act as a destabilizing element. The revised version of the manuscript now includes molecular dynamic simulations that are consistent with their hypothesis and additional clarification of residues implicated in the interaction. The authors have addressed all of my concerns associated with the original version of the manuscript. Overall the work is noteworthy for understanding structural implications for FtsZ stability and is of general interest.

Reviewer #4:

We appreciate the authors’ clarifications of the manuscript, especially regarding the methodology. However, now that the experimental methodology has been clarified, several critical experiments do not conform to standards in the field, making comparison with previous work difficult if not impossible.

Specifically:

“Response: We apologize for this confusion. The term “overnight-cultured” was improperly used because the cells were cultured for 10 h. Thus, we modified this as follows: “To reduce interference by FtsZ treadmilling, we used cells that were cultured for 10 h and had reached late logarithmic phase, a timepoint where most wild-type cells formed well-maintained Z-ring architectures.” (lines 808–810 in the revised manuscript).”

Unless the authors can provide evidence otherwise, if these cells were cultured for 10 hours at 37°C in LB, they are almost certainly in stationary phase with division

occurring only in a limited population. For these types of imaging experiments, the authors should employ established protocols. Yang et al. 2017 (doi: 10.1126/science.aak9995) provides exemplary methodology that is accepted by the field.

Response: We thank you for raising this concern. Although the protocol described by Yang et al. (2017) is an established one, we found it incompatible with our experimental system as detailed below.

First, using M9 medium containing 0.4% glucose, we hardly detected mNeonGreen-labeled FtsZ proteins by either fluorescent microscopy or immunoblotting (panels a and b below). This was apparently due to glucose inhibiting rhamnose-inducible promoter activity in expressing mNeonGreen-labeled FtsZ proteins.

Second, mutant FtsZ variants (especially FtsZ^{ΔNIDR}) exhibit lethal phenotypes when expressed to certain levels (Fig. 2), thus, we used our strictly controlled Rha-induction system (which was incorporated into genomic DNA) prevent premature cell death. We demonstrated that a complete absence of FtsZ-mNeonGreen fusion protein under non-inducing (-Rha) conditions (PMID: 30675381, Fig. S3b). Therefore, it would be difficult, if not impossible, for us to employ any other systems.

In terms of the time point when cells were appropriate for examination, we have presented growth curves (panel c, left section) to illustrate this. While cells cultured for 10 h in LB medium, as used in our previous protocol, remained in the late log phase, to further address your concerns, we performed FRAP experiments using cells cultured for 6.5 h, the earliest time point at which obvious fluorescence and significant phenotypes were observed. At 6.5 h of culturing, approximately 87.5% of wild-type cells contained midcell Z-rings, indicating that most cells were in a dividing state (panel c, right section). We modified the sentence as follows: “*We performed FRAP experiments using cells cultured for 6.5 h, the earliest time point at which obvious fluorescence and significant phenotypes were observed, also a time point where most wild-type cells formed well-maintained Z-ring architectures.*” (lines 823–825 in the Methods part of the revised manuscript).

a**b****c**
Under this new experimental setup, FRAP results showed that fluorescence recovery half-times for FtsZ^{WT}, FtsZ^{D10F}, and FtsZ^{ΔNIDR} were 4.48 s, 7.54 s, and 13.39 s, respectively, showing similar relative to those in our previous version of this manuscript. We replaced data in Figs. 3a and 3b with these updated data (See below for your reference). We modified the descriptions accordingly: “the recovery half-times for FtsZ^{WT}, FtsZ^{D10F}, and FtsZ^{ΔNIDR} are 4.48, 7.54, and 13.39 s respectively.” (lines 1015–1016 in legend of Fig.3 in the revised manuscript); “While entities assembled by FtsZ^{WT} exhibited a fluorescence recovery half-time of 4.48 s after photobleaching, those for FtsZ^{D10F} and FtsZ^{ΔNIDR} took 7.54 and 13.39 s, respectively (Figs. 3a and 3b).” (lines 210–212 in the Results part of the revised manuscript).

“Response: Due to technical constraints, we were unable to monitor filament formation at intervals shorter than 5 min. We also had to work with three samples at the same time and perform measurements one by one on a single piece of equipment. Our setup required removing samples from a temperature-controlled incubator for light scattering measurements. Such operations limited our ability to perform more frequent recordings on assembly dynamics. While we regret that we did not record initial filament formation stages in a more precise manner, our data demonstrated distinct differences between samples.”

1. 15 μM FtsZ is not a physiologically relevant concentration and is 3x higher than concentrations typically used in these types of experiments. GTP is depleted more quickly at high FtsZ concentration, leading to accumulation of GDP which itself inhibits FtsZ polymerization. This experiment should be repeated at more physiologically relevant FtsZ concentrations similar to those used in prior studies of FtsZ assembly.

Response: We actually performed these experiments at both 5 μM and 15 μM concentrations (as shown below) and observed similar difference patterns for wild-type and variant FtsZ proteins. We selected 15 μM for all analyses mainly because this concentration allowed us to obtain highly reproducible data.

Fig. 4a 15 μ M FtsZ

2. It is unclear why the authors chose to incubate cuvettes at 30°C, rather than 25°C, which is standard.

Response: We noted that although many experiments were indeed performed at 25°C, many others were performed at 30°C; Mukherjee, et al. *J. Bacteriol* 1999 (PMID: 9922245), Ting, et al. *Cell* 2018 (PMID: 30343895), Bernhardt, et al. *Molecular Cell* 2005 (PMID: 15916962), and Shinn, et al. *PNAS* 2022 (PMID: 36215496). We performed the *in vitro* study at 30°C because it was closer to 37°C, a temperature at which we performed our *in vivo* studies.

3. The authors should use a spectrophotometer capable of continuous reads such that data points are close together (e.g. 10 seconds or less between reads). Closer time points will allow the authors to capture assembly dynamics in full and permit capture of an entire time course for a single sample, instead of collecting data for three samples at individual time points. The expectation is that there will still be substantial difference in light scattering between WT and mutants, and this approach will allow results to be compared to previously published work. (Using a spectrofluorimeter to take continuous reads is considerably easier for the operator too.)

Response: Indeed, spectrophotometers capable of continuous reading are conventionally used in similar studies.

We performed our experiments based mainly on the following three considerations. First, our experimental design enabled us to examine the three different FtsZ variants under identical conditions in a parallel manner, maintaining rigorous comparability. Second, this experimental design allowed us to simultaneously measure both GTPase enzymatic activity and light scattering parameters, which are much more difficult to

achieve by continuous reading. Third, although we did not employ continuous spectrophotometry, the kinetic curves obtained provided a resolution high enough to reveal the differences of the assembly/disassembly dynamics among the wild-type and the two mutant FtsZ proteins.

4. If the authors are more concerned about capturing disassembly dynamics than assembly dynamics, it's recommended they use their existing data to estimate an appropriate incubation period before taking readings.

Response: Given the inherent challenge of precisely delineating the transition threshold between assembly and disassembly states, which would be different for the three FtsZ variants, we argue that continuous monitoring of the complete dynamic continuum - as opposed to analyzing isolated phases - is more appropriate. Furthermore, an exclusive focus on disassembly may lead to an oversight of critical transitional intermediates, potentially compromising experimental reproducibility in subsequent analyses. Moreover, a synchronized comparison of the three FtsZ variants' dynamic signatures was methodologically crucial for our systematic analysis.

5. The authors should include in the main figure or the supplement data on baseline values before addition scattering begins. Some FtsZ mutants can have unusually high baselines compared to WT, which may be especially important for mutants which are impaired for disassembly.

Response: The average scattering light baseline values for FtsZ^{WT}, FtsZ^{D10F}, and FtsZ^{ΔNIDR} were 102.29, 89.516, and 91.352, respectively (see below). These values indicated that none of the mutants exhibited unusually high baselines compared to the WT.

Baseline

Fig. 4a

Please note that these baseline values were already incorporated in previously submitted Source Data. We can add these in supplementary data if you strongly suggest so.

Reviewer #5:

Reviewer #2 (Remarks to the Author):

I would like to thank the authors for taking into account all suggestions and implementing them in the manuscript. By addressing the concerns of all reviewers, the manuscript convincingly establishes the impact of the N-terminal region of FtsZ in maintaining longitudinal interactions and increasing dynamicity to the Z-ring, and it is, in my opinion, suitable for publication.

Reviewer #4 (Remarks to the Author):

Respectfully, regarding the methodology, we remain unsatisfied with the authors' explanations. If experiments are not done using standard approaches, it is difficult if not impossible to compare results between studies. In the case of FtsZ, as outlined in our previous review, the methodology used by the overwhelming majority of labs in the field is straightforward and produces reliable, repeatable results. If the authors are unable to adapt standard methods to their needs, they should provide an explanation for their decision to use different approaches in the main text of the manuscript. E.g. In the case of data obtained at 15uM FtsZ please indicate that you also attempted these experiments at the more physiological concentration of 5uM but were unable to obtain reliable results.

Response: We modified the sentence as follows: *“Although initial experiments were conducted with FtsZ at a physiologically relevant concentration of 5 μ M, we ultimately adopted a 15 μ M concentration to ensure experimental reproducibility and operational feasibility in achieving our specific research objectives. It followed that the purified form of particular E. coli FtsZ (EcFtsZ)/BsFtsZ/SaFtsZ was added at a final concentration of 15 μ M in respective polymerization buffers, to a fluorometer cuvette with a 1 cm path length.”* (lines 686–691 in the Methods section of the revised manuscript).

With regard to point 5, baseline data should be included in supplementary data not source data.

Response: The baseline datasets from Figure 4a have now been included as Supplementary Figure 4.

Reviewer #5 (Remarks to the Author):
